# Live-cell three-dimensional single-molecule tracking reveals modulation of enhancer dynamics by NuRD

S. Basu [1,2,3,17], O. Shukron [4,17], D. Hall [1,2], P. Parutto [4], A. Ponjavic [5,12,13], D. Shah[1], W. Boucher[1], D. Lando [1], W. Zhang[1], N. Reynolds[2], L. H. Sober[1], A. Jartseva[1], R. Ragheb [2], X. Ma[1], J. Cramard [2], R. Floyd[2,14], J. Balmer[1], T. A. Drury[1], A. R. Carr[5], L.-M. Needham [5], A. Aubert[6], G. Communie[6], K. Gor[2,3,15], M. Steindel[2,3], L. Morey [7,16], E. Blanco [7], T. Bartke [8], L. Di Croce [7,9], I. Berger [10], C. Schaffitzel [10], S. F. Lee [5], T. J. Stevens [11], D. Klenerman [5] ✉, B. D. Hendrich [1,2] ✉, D. Holcman [4] ✉ & E. D. Laue [1,2] ✉

To understand how the nucleosome remodeling and deacetylase (NuRD) complex regulates enhancers and enhancer–promoter interactions, we have developed an approach to segment and extract key biophysical parameters from live-cell three-dimensional single-molecule trajectories. Unexpectedly, this has revealed that NuRD binds to chromatin for minutes, decompacts chromatin structure and increases enhancer dynamics. We also uncovered a rare fast-diffusing state of enhancers and found that NuRD restricts the time spent in this state. Hi-C and Cut&Run experiments revealed that NuRD modulates enhancer–promoter interactions in active chromatin, allowing them to contact each other over longer distances. Furthermore, NuRD leads to a marked redistribution of CTCF and, in particular, cohesin. We propose that NuRD promotes a decondensed chromatin environment, where enhancers and promoters can contact each other over longer distances, and where the resetting of enhancer–promoter interactions brought about by the fast decondensed chromatin motions is reduced, leading to more stable, long-lived enhancer–promoter relationships.

Three-dimensional (3D) genome organization and chromatin dynamics are thought to be crucial for the spatiotemporal control of gene expression. However, little is known about the multiscale dynamics of enhancers and promoters and how this relates to genome organization. In particular, whether chromatin regulators modulate these dynamics remains unclear. To probe the dynamics and organization of regulatory elements at a single-cell level, two complementary methods can be used: live-cell imaging[1–5] and single nucleus chromosome conformation capture experiments[6–12] (such as Hi-C) which reveals snapshots of the structure of the dynamic 3D genome in different individual fixed cells[6–17].

The NuRD complex is a highly conserved 1 MDa multisubunit protein complex that binds to all active enhancers[18]. NuRD combines two key enzymatic activities: nucleosome remodeling via its helicase-containing ATPase, predominantly CHD4 in mouse embryonic stem (mES) cells; and lysine deacetylation via its HDAC1/2 subunits[17–23]. These activities are thought to be present in two subcomplexes: HDAC1/2 associates, along with the histone chaperones RBBP4/7, with the core scaffold proteins MTA1/2/3 to form a stable subcomplex with deacetylase activity[24] and the nucleosome remodeler CHD4 interacts with chromatin by itself and also forms a

second subcomplex with GATAD2A/B and DOC1 (CDK2AP1)[24–26]. The methyl-CpG DNA binding domain proteins MBD2/3 interact directly with both the deacetylase subcomplex and GATAD2A/B[24,27–29], and thus has a critical role in linking the CHD4 remodeler and HDAC subcomplexes together to assemble the intact holo-NuRD complex (Fig. 1a).

Assembly of the intact NuRD complex is critical for controlling cell fate transitions. Knockout of *Mbd3*, which disrupts intact NuRD complex assembly, leads to only moderate up- or downregulation (fine-tuning) of transcription levels, but this modulation prevents mES cell lineage commitment[30–33]. Nucleosome remodeling by NuRD regulates transcription factor and RNA polymerase II binding at active enhancers[18], but whether this impacts enhancer dynamics or enhancer–promoter interactions has remained unclear. Here, to understand whether enhancer dynamics are regulated by this crucial chromatin remodeler, we combine live-cell single-molecule tracking with Hi-C experiments. Specifically, to explore NuRD function, we exploit the ability to unlink the chromatin remodeling and deacetylase subunits of the intact complex by deleting *Mbd3*.

## Results

### An algorithm to segment single-molecule trajectories

To understand how the NuRD complex alters chromatin structure and dynamics, we first set out to understand how its assembly influences chromatin binding. We carried out live-cell 3D single-molecule tracking of NuRD complex subunits in wild-type and *Mbd3*-knockout (mES cells[31]. This strategy, which exploits the fact that MBD2 (and thus the MBD2-linked holo-NuRD complex) is expressed at low levels in mES cells and cannot rescue the *Mbd3* deletion[34,35], allowed us to specifically perturb NuRD complex function. We generated knock-in mES cell lines expressing the endogenous *Chd4*, *Mbd3* and *Mta2* genes fused with C-terminal HaloTags, and confirmed that the tags did not prevent NuRD complex assembly (although subtle changes in subunit expression were observed) (Extended Data Fig. 1). We used a double-helix point spread function microscope[25] to record 3D tracks of single NuRD-HaloTag-JF$_{549}$ complexes as they moved through a 4-µm slice of the nucleus (Fig. 1b) at two distinct temporal regimes: 20 ms and 500 ms (Extended Data Fig. 2). Recording at a 20 ms time resolution allows the detection of both freely diffusing and chromatin-bound proteins[25], and can thus be used to extract the chromatin binding kinetics of NuRD complexes. In contrast, at a 500 ms time resolution, 'motion blurring' substantially reduces the detection of freely diffusing molecules[1], allowing us to focus on the slower subdiffusive chromatin-bound NuRD. Videos showing examples of a succession of images tracking both static and moving CHD4 molecules (recorded using either 20 or 500 ms exposures) can be found in Supplementary Videos 1–8.

To extract biophysical parameters, we developed a machine learning method (a Gaussian mixture model (GMM)) to segment the single-molecule tracks into different classes of subtrajectory (confined and unconfined) by studying their behavior over a sliding window of 11 consecutive images (Fig. 1c). From each subtrajectory, we estimated not just the apparent diffusion coefficient, $D_{app}$ (as previously used for classifying subtrajectories[36,37]) but also the anomalous exponent $\alpha$, the localization length Lc, and the drift magnitude norm$\|V\|$ (ref. 38). The $\alpha$ value (mean squared displacement $\propto$ time$^{\alpha}$), is particularly informative. Diffusing proteins are characterized by an $\alpha$ close to 1 whereas chromatin-bound (confined) proteins exhibit a lower $\alpha$ (refs. 3,38,39), which represents the condensation state[38,40]. The Lc of chromatin-bound proteins is also informative as it reflects the spatial scale that the molecules explore within the nucleus. Finally, by computing the magnitude of the drift vector $V_i$ in three dimensions, we can characterize the total displacement of a molecule during the sliding window. Further details of the approach and of the simulations we carried out to test the algorithm can be found in the Supplementary Data and Methods.

Analysis of the 20 ms exposure tracks of single CHD4 molecules using our approach revealed a fast unconfined state that was freely diffusing with an $\alpha$ of 0.94 ± 0.12 and a $D_{app}$ of 1.3 ± 0.3 µm² s⁻¹ (matching previous observations[24,25]), as well as a confined chromatin-bound state characterized by subdiffusive motion with an $\alpha$ of 0.51 ± 0.02 and a $D_{app}$ of 0.43 ± 0.03 µm² s⁻¹ (Fig. 1c). Similar results were obtained when segmenting the trajectories of two other NuRD complex components, MBD3 and MTA2 (Extended Data Fig. 3c). To demonstrate that our approach can reliably determine differences in the $\alpha$ for diffusing and chromatin-bound molecules, we imaged freely diffusing HaloTag protein. We found that only a small proportion of HaloTag molecules bind to chromatin (as observed[41,42]), and that many molecules have an $\alpha$ of around 1, which is significantly higher than observed for fixed dye molecules ($\alpha$ of 0.62 ± 0.01) (Extended Data Fig. 3c). We conclude that we can use 20 ms trajectories to distinguish unconfined freely diffusing molecules from confined chromatin-bound proteins. We note, however, that the $D_{app}$ of chromatin-bound NuRD molecules can only just be distinguished from those that are stationary when imaging using 20 ms exposures: immobile dye molecules have a median localization error of 60 nm and a $D_{app}$ of 0.3 ± 0.2 µm² s⁻¹, which is quite similar to the 0.43 ± 0.03 µm² s⁻¹ determined for chromatin-bound molecules (Extended Data Fig. 3c). We also note that shorter exposures are needed for the detection of faster moving smaller molecules (for example, transcription factors).

### HDAC subcomplex requires CHD4 for chromatin binding

Having developed an approach to segment the 20 ms exposure trajectories of the NuRD complex into chromatin-bound and freely diffusing molecules (Fig. 2a,b), we investigated how removal of MBD3, which disrupts the interaction between the HDAC- and CHD4-containing NuRD subcomplexes[18,24], affects chromatin binding. To explore whether the two subcomplexes are preassembled before binding to chromatin, we imaged both the CHD4 remodeler and the HDAC-containing subcomplexes. Imaging the HDAC-containing subcomplex using tagged MTA2 revealed a 1.7-fold increase in $D_{app}$ for freely diffusing MTA2 in the absence of MBD3 (Fig. 2c and Extended Data Fig. 3c), demonstrating that the deacetylase subcomplex is normally associated with CHD4 in intact NuRD (Fig. 1a). Single-molecule tracking of CHD4, however, revealed only a 1.05-fold increase in the $D_{app}$ of freely diffusing CHD4 in the absence of MBD3 (Fig. 2c and Extended Data Fig. 3c). A larger increase might have been expected from the disassembly of the holo-NuRD complex but mES cells also contain CHD4 that is not present in NuRD, both on its own and in the ChAHP complex with ADNP and HP1β,γ[43]. As a control, we also imaged-tagged MBD3 and showed that both freely diffusing MBD3 and MTA2 molecules have similar diffusion coefficients, consistent with MBD3 linking the two subcomplexes together in intact NuRD[18,24,25] (Fig. 1a). Finally, we showed that MBD3 does indeed interact with CHD4 via GATAD2A in vitro using purified GATAD2A in pulldown reconstitution experiments (Extended Data Fig. 3b), and through the knockdown of *Gatad2a* and *Gatad2b*, which slightly increased the diffusion coefficient of CHD4 (1.05-fold) (Extended Data Fig. 3d,e). We conclude that the CHD4 and HDAC subcomplexes in NuRD are normally preassembled before binding to chromatin.

We then examined how the NuRD complex interacts with chromatin by comparing the percentage of freely diffusing versus chromatin-bound CHD4 and MTA2 molecules in the presence and absence of MBD3. We observed a 1.1-fold decrease in the percentage of CHD4 molecules bound to chromatin in the absence of MBD3, but there was a much more significant (2.4-fold) decrease in the percentage of chromatin-bound MTA2 molecules upon MBD3 depletion (Fig. 2c). This suggested that CHD4, rather than the deacetylase subcomplex, is primarily responsible for the association of NuRD with chromatin. This finding was supported by in vitro experiments that showed that, in comparison with CHD4, the deacetylase subunit by itself does not bind strongly to nucleosomes (Extended Data Fig. 4a). We conclude that NuRD normally exists as an intact complex in mES cells and that

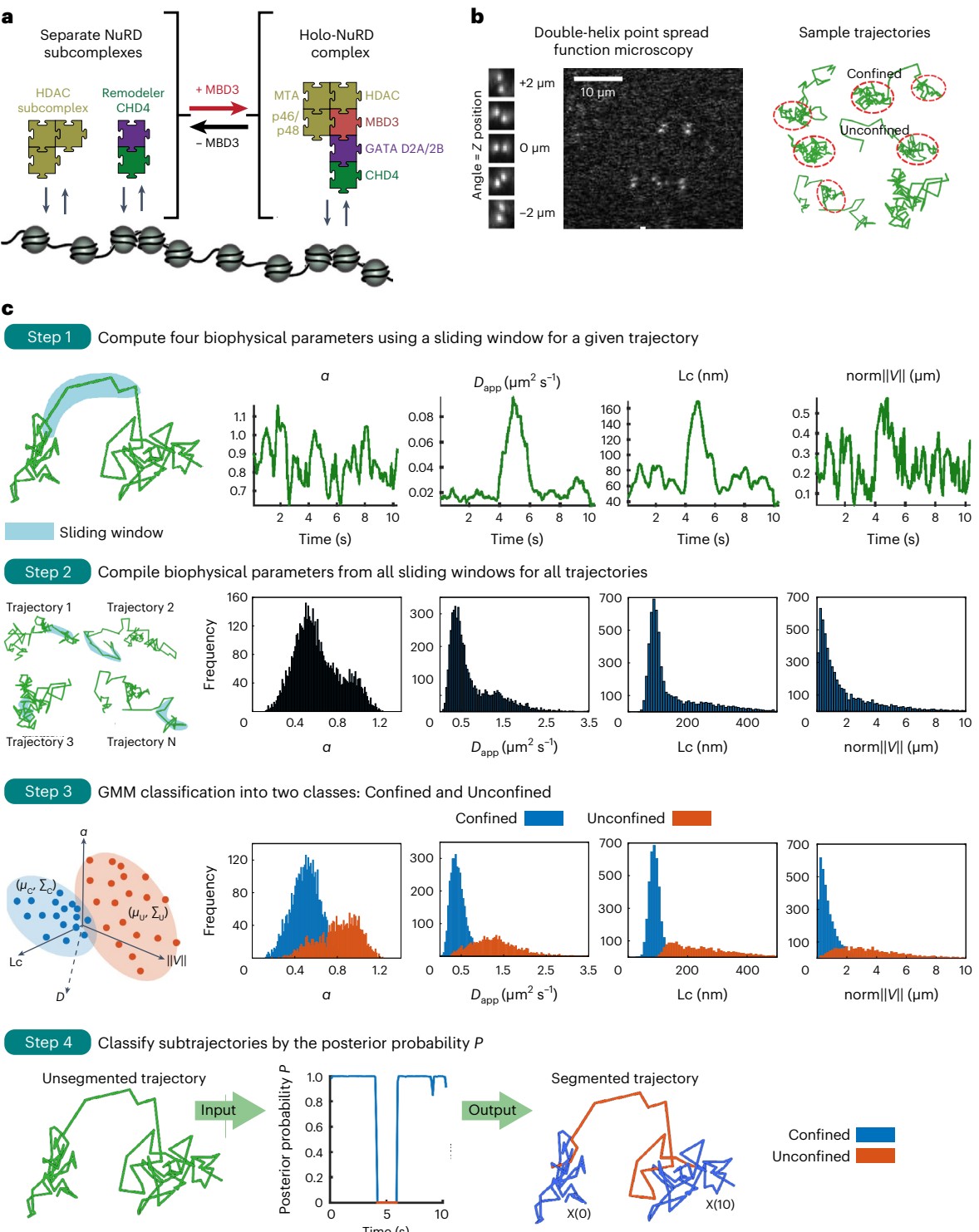

**Fig. 1 | Live-cell imaging to study NuRD complex binding kinetics and function. a**, Schematic representation of the NuRD complex interacting with chromatin in the presence and absence of MBD3. **b**, Left, single JF$_{549}$-HaloTagged molecules in the NuRD complex were tracked in 3D using a double-helix point spread function microscope; two puncta are recorded for each fluorophore with their midpoint providing the lateral $x$, $y$ position and the angle between them representing the axial position in $z$ relative to the nominal focal plane (see Extended Data Fig. 2 and Supplementary Videos 1–8 for examples of the raw data). Right, examples of extracted single particle trajectories from 20 ms exposure imaging of CHD4 show periods of unconfined and confined diffusion. **c**, The approach used for segmentation of the single-molecule tracks; the data shown are from the 20 ms exposures of CHD4-HaloTag-JF$_{549}$. Step 1, Left, a single-molecule trajectory showing an example sliding window (blue). Right, four biophysical parameters are calculated for a sliding window that is moved through

the trajectory: $\alpha$, $D_{app}$, Lc and the norm$\|V\|$ of the mean velocity, were all estimated from a sliding window of 11 consecutive images. Step 2, Left, several trajectories with example sliding windows (blue). Right, Histograms of the values of the four biophysical parameters extracted in Step 1 from all the sliding windows computed for all the recorded trajectories. Step 3, Left, Based on the values of the four biophysical parameters (producing a four-dimensional feature space) each point in each trajectory is classified as either confined (C) or unconfined (U) using a Gaussian mixture model (GMM). The histograms from Step 2 can then be separated into confined (blue) and unconfined (orange) populations. Step 4, the posterior probability $P$ of the GMM (Step 3) is computed on the four parameters for each sliding window $X_i$ where the index of the trajectory is represented by $i = 1,\ldots,N$ ($X_i(k\Delta t) \in C$ with $P(k\Delta t) > 1 - P(k\Delta t)$ (blue); otherwise $X_i(k\Delta t) \in U$ (orange)). The result is a segmented trajectory where each timepoint is assigned as confined or unconfined (see Methods for more details).

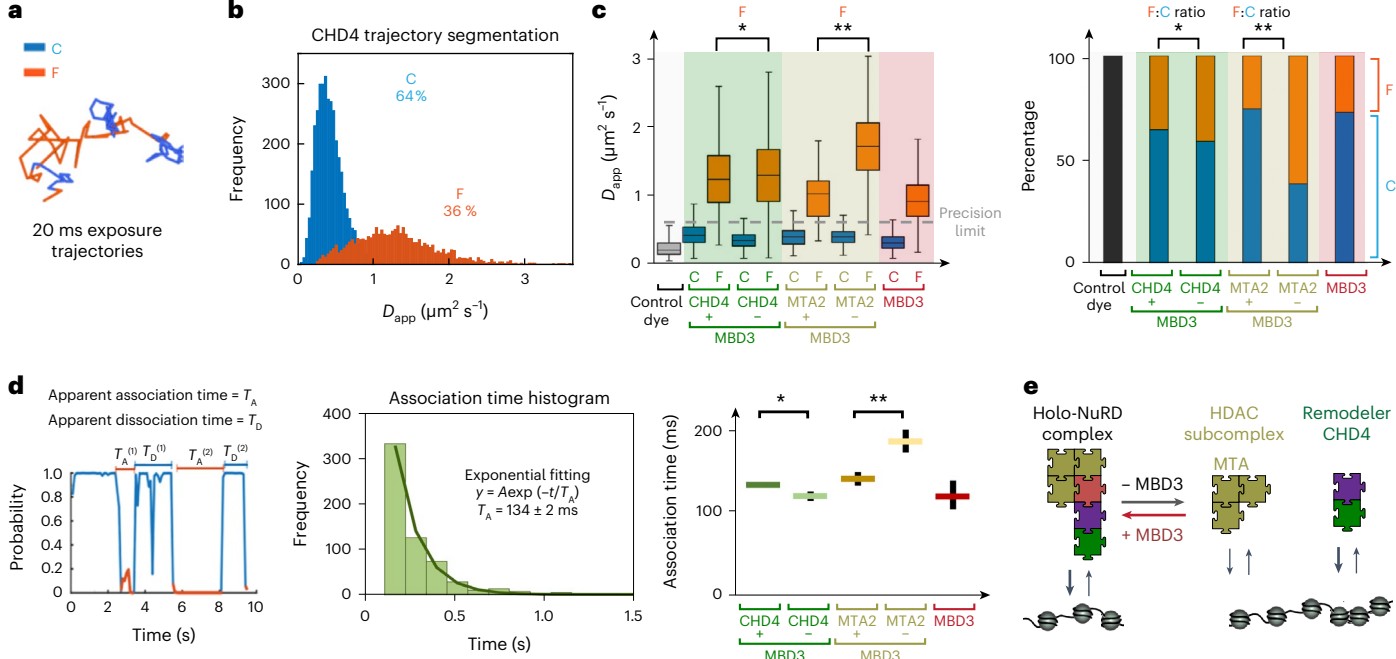

**Fig. 2 | Live-cell single-molecule tracking reveals that the NuRD complex assembles before it binds to chromatin. a**, Segmentation of an example 20 ms trajectory of CHD4 into chromatin-bound (C) (blue) and freely diffusing (F) states (orange). **b**, Percentage of molecules and distribution of $D_{app}$ for chromatin-bound and freely diffusing CHD4 molecules. **c**, Left, boxplot of $D_{app}$ for CHD4 and MTA2 molecules in the presence and absence of MBD3; *$P$ = 0.009, **$P$ = 1×10$^{-25}$ (two-sided Kolmogorov–Smirnov test). Center line, median; box limits, upper and lower quartiles; whiskers, 95% confidence interval. Data for MBD3 are shown as a control, and the gray dotted line indicates the upper bound (at the 95% confidence interval) of the $D_{app}$ determined for stationary JF$_{549}$ dye molecules. Right, percentage of CHD4 and MTA2 molecules in the presence and absence of MBD3 (from Gaussian fitting, *$P$ = 1×10$^{-6}$, **$P$ < 1×10$^{-27}$ (two-sided Fisher's exact test)). The numbers of cells per trajectory used in the analysis were: 30/5,557 (CHD4), 25/2,337 (CHD4−MBD3), 10/336 (MTA2), 10/652

(MTA2−MBD3) and 30/2,224 (MBD3). **d**, Left, a plot of the confinement probability allows determination of the association $T_A$ and dissociation $T_D$ times (defined, respectively, as the time a trajectory spends between periods of confined or unconfined motion). Middle, a single exponential curve of rate lambda = 1/$T_A$ was then fitted to the distribution of association times. Right, the association times extracted for CHD4 and MTA2 were then compared with those in the absence of MBD3, and with those for the MBD3 control. The number of association times used in the analysis were: 1,436 (CHD4), 668 (CHD4−MBD3), 62 (MTA2), 361 (MTA2−MBD3) and 407 (MBD3). Data are presented as mean values. Error bars show 95% confidence intervals, *$P$ = 1×10$^{-47}$ for CHD4 versus CHD4−MBD3 and **$P$ = 1×10$^{-25}$ for MTA2 versus MTA2−MBD3 (two-way analysis of variance (ANOVA)). **e**, Schematic representation of a model in which MBD3-dependent assembly of the NuRD complex increases the association rate of the deacetylase subcomplex.

the removal of MBD3 disrupts chromatin binding of the deacetylase complex but not the CHD4 remodeler.

To further investigate the chromatin binding kinetics of the CHD4 remodeler and the MTA2 deacetylase subcomplex in the presence and absence of MBD3, we next determined association times from the time spent freely diffusing between confined chromatin-bound states. The distribution of association times was well approximated by a single exponential, suggesting a Poissonian process. Consistent with our finding that CHD4 is primarily responsible for recruitment of NuRD to chromatin, we found no increase in the association time of CHD4 upon removal of MBD3 (Fig. 2d). (The decrease in the observed association time is consistent with faster diffusion of the smaller CHD4 subcomplex resulting in more frequent collisions with chromatin.) However, we did find a significant (1.3-fold) increase in the association time of MTA2 upon MBD3 depletion (Fig. 2d), consistent with CHD4 recruiting the deacetylase subcomplex to chromatin.

We also attempted to determine dissociation times from the time spent bound to chromatin between unconfined freely diffusing states. Although no changes in dissociation time were observed (Extended Data Fig. 4c), we reasoned that our trajectories would be truncated by photobleaching. We therefore took advantage of 'motion blurring' when recording 500 ms trajectories to detect only chromatin-bound proteins[1,44], and combined this with time-lapse imaging using different intervals between exposures. To our surprise, this showed that the dissociation times were much longer than we had expected (greater than

100 s for MBD3; Extended Data Fig. 4e), such that it proved impossible to track individual molecules for long enough to determine reliable dissociation rates. We conclude that, once bound to a target site, intact NuRD binds for unexpectedly long times.

**Intact NuRD modulates chromatin movement at enhancers**

We next studied the dynamics of chromatin-bound NuRD by tracking these slower-moving molecules at a time resolution of 500 ms (Fig. 3a and Extended Data Fig. 5a). Analysis of trajectories that lasted more than 5 s using the GMM (Fig. 3b) revealed two states of chromatin-bound CHD4 (slow and fast moving) with a $D_{app}$ of 0.006 ± 0.002 and 0.018 ± 0.006 μm$^2$ s$^{-1}$ (Fig. 3c and Extended Data Fig. 5b). The slow-moving chromatin-bound NuRD molecules could still only just be distinguished from those that are stationary even when imaging using 500 ms exposures; we found that immobile dye molecules had a median localization error of 34 nm and a $D_{app}$ of 0.004 ± 0.003 μm$^2$ s$^{-1}$, which is again similar to the 0.006 ± 0.002 determined for slow-moving chromatin-bound CHD4 (Extended Data Fig. 5a).

We then compared the dynamics of chromatin-bound CHD4 molecules in *Mbd3*-knockout and wild-type cells. Surprisingly, we found that $\alpha$, Lc and $D_{app}$ of the fast-moving chromatin-bound CHD4 molecules were all higher in wild-type cells (Fig. 3d). The increase in $\alpha$ in wild-type cells unexpectedly suggests that, in the presence of NuRD, chromatin is less condensed, whereas the increased $D_{app}$ and Lc show that chromatin-bound CHD4 molecules diffuse more rapidly and explore

a larger nuclear volume. We had expected to find that recruitment of the deacetylase by CHD4 would lead to less acetylated chromatin and greater condensation[45–49] and that the chromatin-bound CHD4 molecules in wild-type cells would thus explore a smaller nuclear volume.

When we visualized trajectories of the fast-moving chromatin-bound CHD4, we observed a proportion of molecules exhibiting periods of motion in a defined direction, characterized by a high $\alpha$ (>1.0) and high drift (for example, the trajectory in Fig. 3a). This suggested that there may be two types of fast-moving chromatin-bound CHD4. Indeed, when looking at the shape of the distribution of $\alpha$ values extracted from sliding windows of these trajectories, we observed two different populations of fast-moving molecules (Fig. 3c) and used Gaussian fitting to characterize their distributions (Fig. 3b). The two stages of our analysis thus revealed a single slow state (S) with $\alpha_s$ of $0.59 \pm 0.01$ (67% of subtrajectories) and two fast substates (F1 and F2) with different $\alpha$ values: $\alpha_{F1}$ of $0.60 \pm 0.01$ (26%) and $\alpha_{F2}$ of $0.89 \pm 0.02$ (7%) (Fig. 3c and Extended Data Fig. 5c,d). Molecules in the fast F1 state have the same distribution of $\alpha$ as those in the slow state and they therefore explore the same chromatin environment. However, they diffuse faster and have a larger Lc and thus move further within the nucleus (Fig. 3c). Molecules in the fast F2 state, however, have a higher $\alpha$ and they explore an even larger area of the nucleus (higher Lc) than those in both the slow and the fast F1 states (for example, the trajectory in Fig. 3a). Moreover, they have high drift, indicative of movement in a defined direction; this is also consistent with the higher $\alpha$.

Having observed both condensed (low $\alpha$) and decondensed (high $\alpha$) motion for chromatin-bound CHD4, we carried out a similar analysis in *Mbd3*-knockout cells. Although chromatin is less condensed in the presence of intact NuRD (see above), we observed a significant decrease in the proportion of CHD4 molecules in the fast decondensed F2 state (7.4% in wild-type cells versus 18% in *Mbd3*-knockout cells; Fig. 3e and Extended Data Fig. 5c,d). As a control, we also compared the dynamics of chromatin-bound MBD3 with that of CHD4 and found that it too exhibited one slow and two fast states. Both chromatin-bound MBD3 and CHD4 molecules exhibited motion in the fast F1 and F2 states in around 22–26% and 7–8% of trajectories, respectively, confirming that these states are a property of the intact NuRD complex and not just of CHD4 (Fig. 3d,e). Importantly, visualization of individual trajectories identified molecules that switch between the three states: S, F1 and F2 (Fig. 3a and Extended Data Fig. 5e). Thus, they are unlikely to represent either CHD4 forming different complexes or NuRD complex molecules bound in different regions of the nucleus.

The fast F1 and F2 states of chromatin-bound NuRD could result from movement on DNA due to chromatin remodeling or, bearing in mind the long dissociation times we determined for CHD4 (see above), from movement of NuRD-bound enhancers. To distinguish between these possibilities, we targeted sites near active enhancers with dCas9-GFP, either by transfecting a previously studied CARGO vector expressing 36 different gRNAs targeting a *Tbx3* enhancer[3] or by transfecting a single gRNA that targets DNA repeats near the

*Nanog* gene (Extended Data Fig. 6a,b). Targeting nearby the *Nanog* enhancer using our single gRNA was confirmed by colocalization of GFP-tagged dCas9 (detected by immunofluorescence) with DNA fluorescence in situ hybridization (FISH) probes (Extended Data Fig. 6c). We carried out these experiments in cells expressing an ER–MBD3–ER (estrogen receptor–MBD3–estrogen receptor) fusion protein in which the nuclear localization of MBD3 (and thus assembly of the intact NuRD complex) is tamoxifen-inducible[18]. In this system, the intact NuRD complex assembles and remodels chromatin/transcription factor binding following induction; after 24 h, the transcription factor landscape has been reset and transcriptional changes have occurred. This allowed us to study the chromatin environment created by intact NuRD assembly shortly after it had become established (allowing us to distinguish direct from downstream effects). We also imaged cells showing bright undivided foci to exclude data from cells in the S or G2 phases of the cell cycle, which exhibit blurred foci or doublets (Extended Data Fig. 7a). Because of the background fluorescence from freely diffusing dCas9-GFP we had to track the enhancer loci in a single two-dimensional (2D) plane. Although this meant that we could not directly compare the parameters obtained in the 2D (active enhancer) and 3D (NuRD single-molecule) tracking experiments, classification of the subtrajectories once again revealed a slow and a fast-moving chromatin state (Extended Data Fig. 7b), including two subpopulations of fast-moving chromatin (Extended Data Fig. 7c). The longer enhancer locus trajectories (cf. chromatin-bound CHD4) allowed better characterization of the proportions of the different slow and fast states (for example, see Fig. 4a and Supplementary Videos 9–12). As when tracking CHD4 single molecules, addition of MBD3 (and thus the assembly of intact NuRD) significantly increased the $D_{app}$ of both enhancers in the fast-diffusing F1 and F2 states (Fig. 4b). Moreover, in the presence of intact NuRD, we again observed a decreased proportion of subtrajectories in the fast decondensed F2 state for both the *Tbx3* and *Nanog* enhancers (Fig. 4c and Extended Data Fig. 7d).

Previous work has suggested that enhancer dynamics are related to transcription[3] and we wondered whether the changes we observe (± intact NuRD) result from altered levels of gene expression. We therefore tracked chromatin-bound CHD4 molecules after adding DRB—a small molecule inhibitor of transcriptional elongation[50]. Premature termination by DRB led to some reduction in the proportion of bound CHD4 molecules exhibiting the fast F1 motion (from 26% to 19%), but there was no reduction in the proportion of molecules in the fast F2 state or change in the chromatin environment (that is, in $\alpha$) in the presence of a block on transcriptional elongation (Extended Data Fig. 5c). Finally, we tracked MBD3 molecules while blocking HDAC1/2 deacetylase activity with FK228 (ref. 51). Once again, however, there was no significant change in the proportion of molecules in the fast decondensed F2 state (Extended Data Fig. 5c). We conclude that the changes in enhancer dynamics ± intact NuRD are not due to altered transcription elongation or deacetylation activity.

**Fig. 3 | Assembly of the NuRD complex decondenses chromatin. a**, Left, example trajectory of a chromatin-bound CHD4 molecule showing periods of both slow (dark blue) and fast (light blue) subdiffusive motion. Two fast substates (F1 and F2) are observed, with the F2 state showing movement in a defined direction. Right, four biophysical parameters calculated along this trajectory with the fast F2 subtrajectories showing a higher $\alpha$, increased Lc and increased drift. **b**, Schematic illustrating the analysis of the 500 ms exposure trajectories of chromatin-bound NuRD complex subunits. **c**, Histograms of the values of the four biophysical parameters extracted from all the sliding windows computed for all the recorded trajectories to distinguish slow-moving/immobile (dark blue) and fast-moving (light blue) chromatin-bound molecules (Stage 1 of the analysis in **b**). Gaussian fitting to the distribution of $\alpha$ values (Stage 2 of the analysis in **b**) identified two values of $\alpha$ for molecules in the fast-moving chromatin state (light blue) (Extended Data Fig. 5). **d**, Comparison of biophysical

parameters for the CHD4 remodeler in the presence and absence of MBD3, and for MBD3 itself. Left, $\alpha$ values resulting from Gaussian fitting (data presented as mean values, error bars show 95% confidence intervals, *$P = 1 \times 10^{-4}$, two-sided Kolmogorov–Smirnov test). Boxplots of (middle) the lengths of confinement and (right) $D_{app}$ values (*$P = 1 \times 10^{-4}$ and *$P = 1 \times 10^{-32}$, respectively, two-sided Kolmogorov–Smirnov test). In **c** and **d**, the gray dotted lines indicate the upper bounds of the different biophysical parameters (at the 95% confidence interval) determined for stationary JF$_{549}$ dye molecules. The numbers of cells per trajectory used in the analysis were: 30/3,059 (CHD4), 15/2,111 (CHD4–MBD3) and 30/1,816 (MBD3). Center line, median; box limits, upper and lower quartiles; whiskers, 95% confidence interval. **e**, Left, percentage of molecules in the slow or fast chromatin-bound states (*$P < 0.01$, two-way ANOVA). Right, schematic representation of the three states of chromatin-bound NuRD.

## NuRD increases intermediate-range enhancer–promoter contacts

To understand whether the alteration of enhancer dynamics in NuRD affects genome architecture and enhancer–promoter interactions, we next carried out in-nucleus Hi-C experiments. We obtained high-quality contact maps for both wild-type and *Mbd3*-knockout ES cells after combining our wild-type data with previously published[52] (and consistent) datasets (Extended Data Fig. 8a). As previously observed[6,8–17,53], the Hi-C contact maps showed that the genome is segregated into: (1) A and B compartments (regions containing a higher or lower density of genes, respectively); (2) megabase-scale topologically associating domains (TADs), which have a higher frequency of intradomain

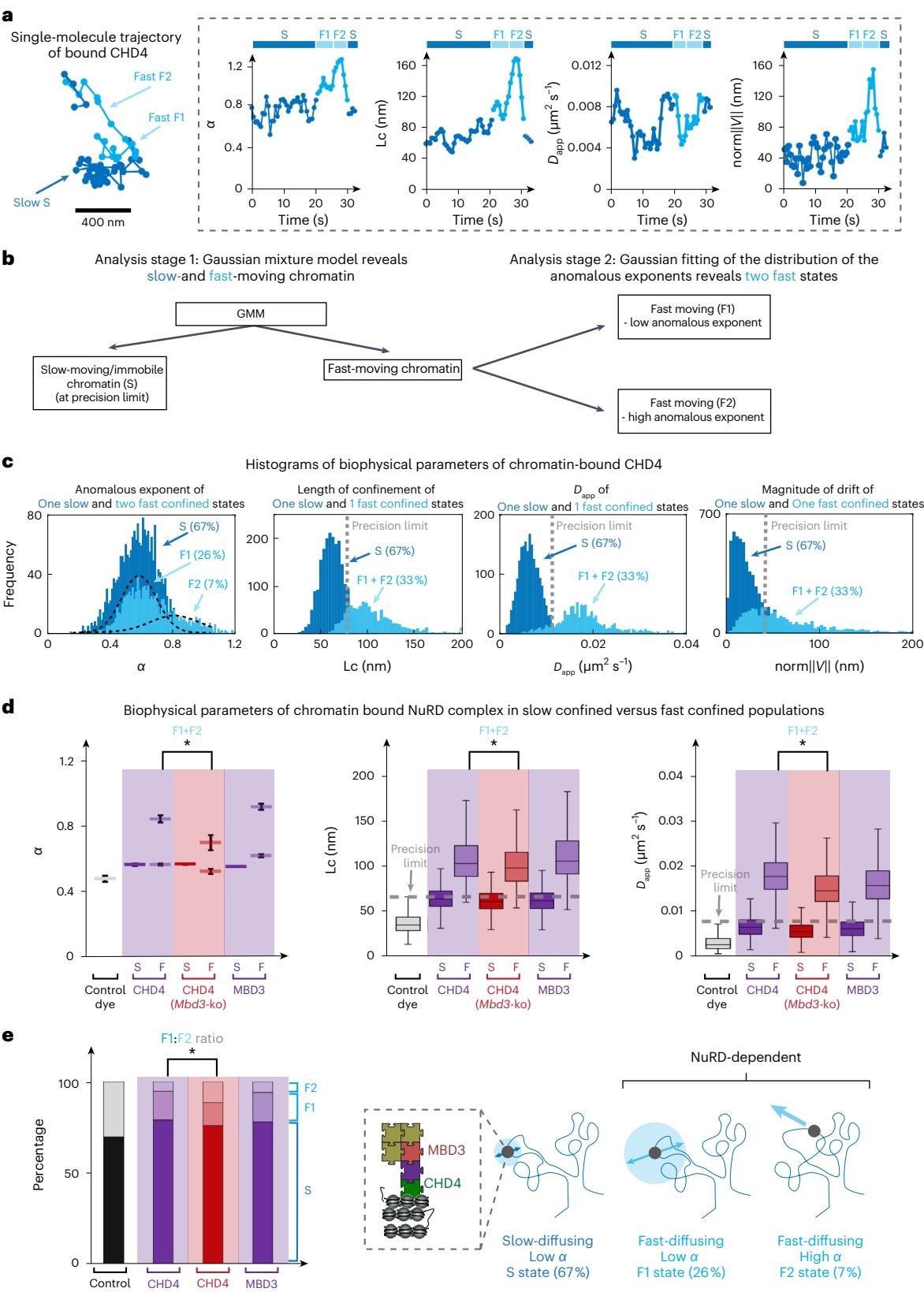

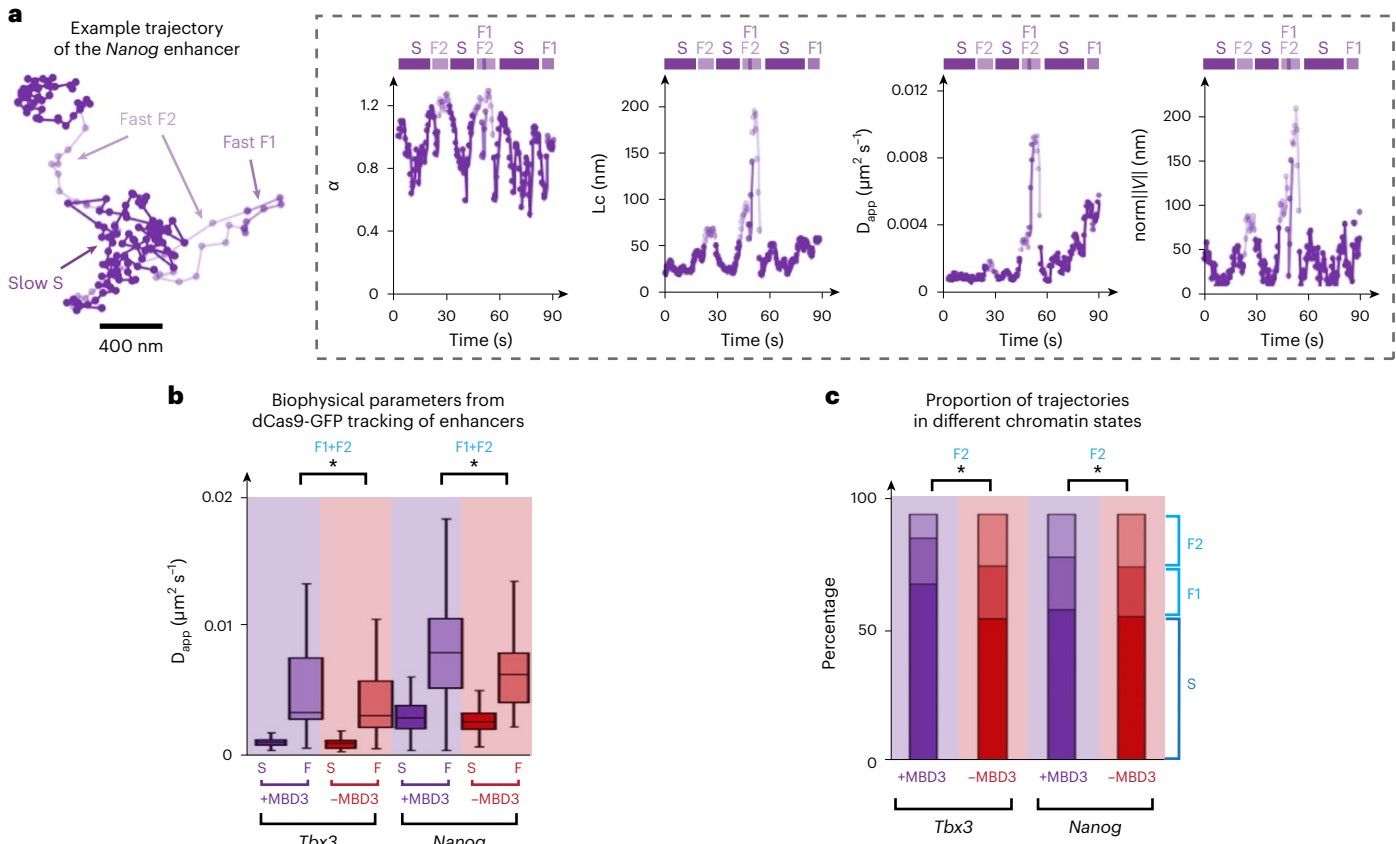

**Fig. 4 | Assembly of the intact NuRD complex modulates the movement of active enhancers. a**, Left, example trajectory of the *Nanog* enhancer segmented to show periods of slow and fast subdiffusive motion (dark and light blue, respectively). Right, $\alpha$, Lc, $D_{app}$ and norm$\|V\|$ of the locus extracted from the trajectory shown. **b**, Boxplots of $D_{app}$ calculated for 2D trajectories of loci near the *Tbx3* and *Nanog* enhancers, in the presence and absence of MBD3 (*$P = 0.045$ and *$P = 1 \times 10^{-4}$ for *Tbx3* and *Nanog*, respectively, two-sided Kolmogorov–Smirnov

test). **c**, Gaussian fitting to the distribution of the $\alpha$ values identifies a single slow and two faster states for enhancer loci (Extended Data Fig. 7). The percentage of subtrajectories of enhancer loci exhibiting slow (S) and fast chromatin motions with either a low (F1) or high (F2) $\alpha$ is shown (*$P < 0.01$, two-way ANOVA). The numbers of trajectories used in the analysis were 237 (*Tbx3* + MBD3) and 287 (*Tbx3*-MBD3); 546 (*Nanog* + MBD3) and 229 (*Nanog*-MBD3). Center line, median; box limits, upper and lower quartiles; whiskers, 95% confidence interval.

chromatin interactions; and (3) loops mediated, for example, via CTCF/cohesin binding, where specific genomic regions contact each other more frequently.

Comparison of the *Mbd3*-knockout and wild-type Hi-C data showed that NuRD leads to an increase of ~30% in the probability of intermediate-range contacts on the scale of TADs (500 kb to 3 Mb) (Fig. 5a and Extended Data Fig. 8b). The genome-wide increase in mean contact length per region binned (Extended Data Fig. 8c) was most noticeable for regions containing NuRD-regulated genes that are within the A compartment in both *Mbd3*-knockout and wild-type cells (KO-A and WT-A, respectively) (Extended Data Fig. 8d). Indeed, 77% and 17% of NuRD-regulated genes[54], respectively, were found in the A and B compartments, with only small proportions (~3%) moving from A to B or vice versa. In addition, NuRD downregulated genes are significantly enriched in the A compartment ($P < 1 \times 10^{-10}$; odds ratio, 1.16), and significantly depleted in the B compartment ($P < 1 \times 10^{-10}$; odds ratio, 0.43) (Extended Data Fig. 8e,f). Chromatin immunoprecipitation followed by high-throughput sequencing (ChIP–seq) and CUT&RUN experiments in the absence of fixation show that NuRD binds predominantly to enhancers with little enrichment at promoters[18], suggesting that the MBD3 ChIP–seq signal observed at promoters results from association with NuRD-bound enhancers. We therefore further categorized promoters according to whether or not they bind MBD3, and found that putative 'NuRD downregulated, NuRD enhancer contacting' genes are also significantly enriched in the A compartment ($P < 1 \times 10^{-10}$;

odds ratio, 1.27) and significantly depleted in the B compartment ($P < 1 \times 10^{-10}$; odds ratio, 0.41) (Extended Data Fig. 8e,f). Thus, analysis of the Hi-C and ChIP–seq data showed that NuRD-regulated genes are predominantly in the A compartment where they may be downregulated through contact with NuRD-bound enhancers.

Comparison of the Hi-C contact maps for *Mbd3*-knockout and wild-type cells showed that NuRD weakens the boundaries between A/B compartments and TADs (Fig. 5b and Extended Data Fig. 8g), with promoters suspected (see above) of contacting NuRD-bound enhancers having an increased number of cross-compartment contacts (Extended Data Fig. 8h). We also found that NuRD facilitates a significant genome-wide decrease in the insulation between TADs (Extended Data Fig. 8i). Although NuRD-regulated genes do not move between compartments (Extended Data Fig. 8j), these results suggest that the NuRD-mediated increase in enhancer dynamics may facilitate interactions across A/B compartment and TAD boundaries[55,56].

The blurring of TAD and A/B compartment boundaries suggested that NuRD might alter CTCF/cohesin binding[56,57]. We therefore carried out CUT&RUN experiments to study chromatin binding of CTCF and cohesin, which have a key role in loop and TAD formation[13,58–63]. We found that NuRD leads to a redistribution of both CTCF and, more particularly, SMC3 (a subunit of the cohesin complex) (Extended Data Fig. 9a–c,e). Moreover, a significant proportion of NuRD-regulated genes are found near to CTCF/cohesin binding sites; this was most noticeable for genes that are upregulated and whose promoters (we

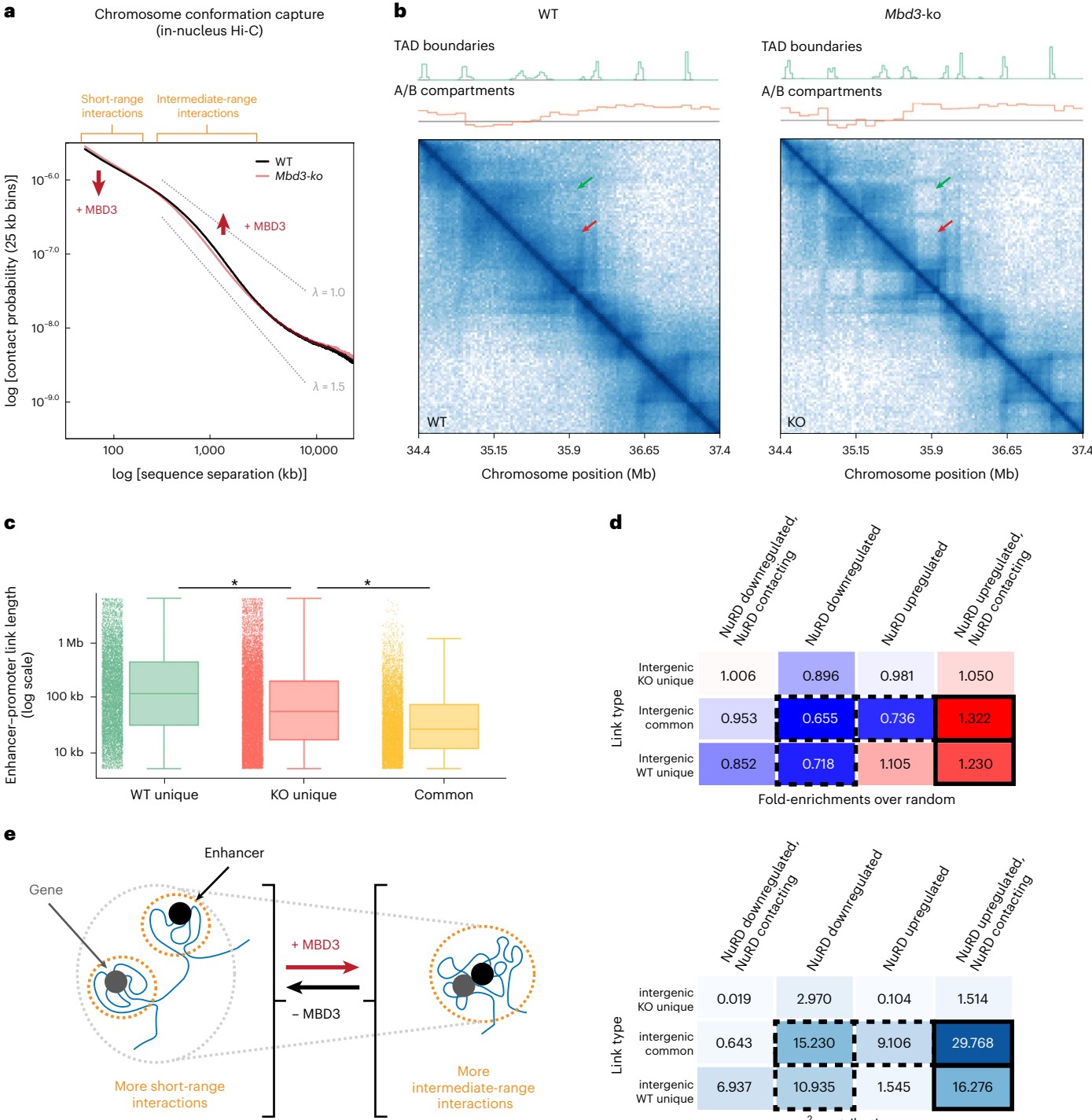

**Fig. 5 | Assembly of the NuRD complex increases Mb-range genome and enhancer–promoter interactions. a**, log–log plots of contact probability as a function of genomic sequence separation (averaged across the genome), derived from in-nucleus Hi-C experiments of *Mbd3*-knockout (KO) (red) and wild-type (WT) (black) ES cells, shows a significant (~30%) increase in intermediate-range (~1 Mb) contacts in WT cells ($P = 1 \times 10^{-18}$, two-sided Mann–Whitney *U*-test; see also Extended Data Fig. 8b). **b**, Part of the Hi-C contact maps for Chromosome 1; the density of contacts is indicated by the color intensity. TAD boundaries are weakened in wild-type cells, resulting in an increased density of contacts between adjacent TADs, both within and between A/B compartments (red and green arrows); see Extended Data Fig. 8g–i for genome-wide comparisons. **c**, Boxplots showing intrachromosomal enhancer–promoter link lengths, determined using a modified version of the activity-by-contact algorithm[66], present in both KO and WT ES cells (orange), in only KO cells (red) or in only WT cells (green). The number of WT unique links = 7,941; common links = 8,546; KO unique

links = 12,932; *$P = 1 \times 10^{-10}$, Bayesian version of *t*-test. Center line, median; box limits, upper and lower quartiles; whiskers, 95% confidence interval; bars on the left of each plot show all the data. **d**, Fold-enrichment (upper panel) and (lower panel) chi-squared test (lower) when intergenic enhancer–promoter interactions found in both WT and KO cells, as well as those found uniquely in either KO or WT, are correlated with genes that are up- or downregulated in the presence of intact NuRD. Enriched and depleted types of interaction are colored red and blue, respectively, and significant changes are highlighted using solid and dashed black boxes (see Extended Data Fig. 10c for an example of changes in enhancer–promoter contacts). Interactions where NuRD-bound enhancers can be seen to contact the promoter, which are either found in both KO and WT or uniquely in WT, are enriched at upregulated genes. In contrast, there is a depletion in interactions between intergenic enhancers and promoters of downregulated genes. **e**, Schematic interpretation of the results of the Hi-C experiments.

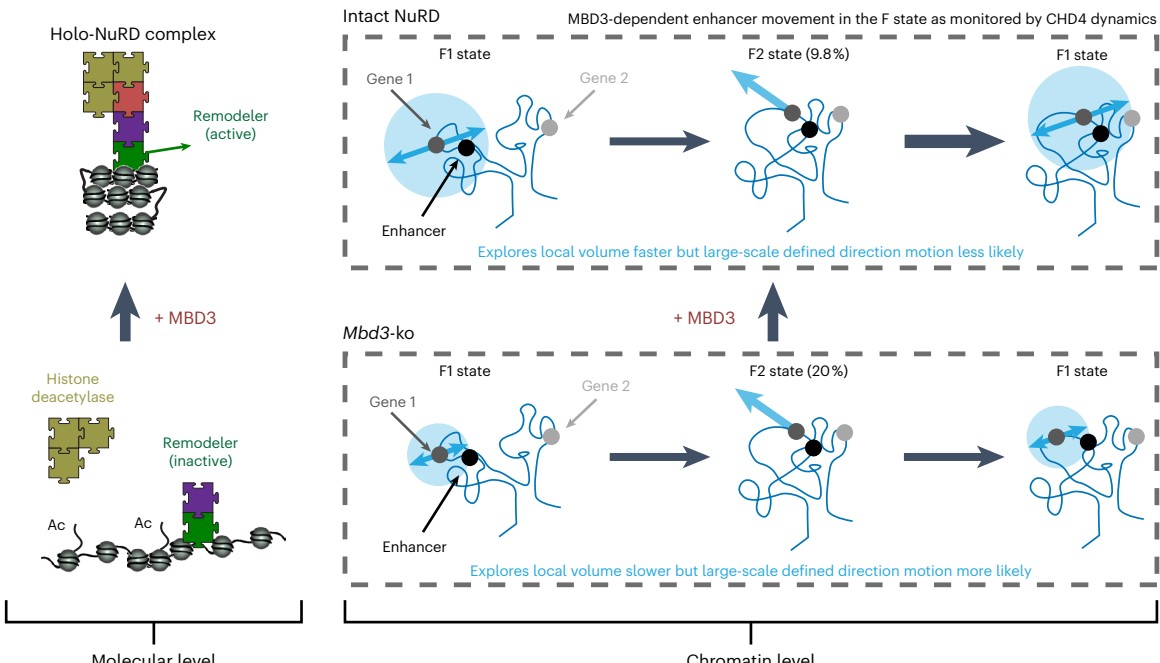

**Fig. 6 | Schematic model of NuRD function.** Left, a molecular level view of how assembly of the NuRD complex affects chromatin remodeling[18]. Right, a chromatin level view of how assembly of the NuRD complex increases the volume explored by an enhancer while at the same time reducing its likelihood of entering the fast F2 state in which the movement of decondensed chromatin might reset enhancer–promoter proximity.

suspect) contact NuRD-bound enhancers, but it was also true for such genes that are downregulated (Extended Data Fig. 9d). We also found that genes that are either up- or downregulated by NuRD tend to be colocated in the linear genome sequence, for example, the *Slc6a12*, *Prmt8* and *Htra1* genes (Extended Data Fig. 9e). This clustering of similarly regulated genes suggests that NuRD alters genome structure/dynamics and affects the expression of groups of nearby genes in a similar way.

Because changes in TADs and CTCF-cohesin loops are thought to influence enhancer–promoter proximity[60,64,65], we used a modified version of the activity-by-contact algorithm[66] to study changes in enhancer–promoter interactions in *Mbd3*-knockout versus wild-type cells (Extended Data Fig. 10). After defining active enhancers and promoters using H3K27ac and H3K4me3 ChIP–seq profiles[18] (Methods), this revealed that enhancer–promoter interactions that occur only in the presence of the intact NuRD complex tend to link together genomic regions separated by longer distances than those that occur in its absence (Fig. 5c). In addition, when we considered enhancer–promoter interactions and changes in transcription[54], we found more contacts between intergenic NuRD-bound enhancers and promoters that are upregulated by NuRD, and fewer contacts between intergenic enhancers and promoters that are downregulated by NuRD (Fig. 5d). We also carried out two-color enhancer–promoter DNA-FISH studies of three key pluripotency genes (*Bmp4*, *Sox2* and *Tbx3*), and showed that the presence of NuRD led to a significant increase in the average distance between their enhancers and promoters (Extended Data Fig. 6d,e). This confirmed (using a different approach to the Hi-C experiments) that NuRD not only modulates chromatin decompaction and enhancer dynamics, but also alters enhancer–promoter interactions.

## Discussion

In this work, we have developed a computational approach to analyze 3D trajectories of single molecules and segment them according to their diffusion behavior. Key advantages of our computational approach are that it allows us to (1) measure four different biophysical parameters to explore nuclear dynamics and (2) remove regions of the trajectories where the molecules are essentially not moving–allowing us to estimate biophysical parameters that define the behavior of moving molecules.

Using 20 ms exposure imaging, we were able to distinguish molecules that are freely diffusing from those that were chromatin bound. By comparing wild-type mES cells with a mutant cell line where we had knocked out *Mbd3*–a gene encoding a protein subunit that links the chromatin remodeling and histone deacetylase activities of NuRD together–we were able to show that the NuRD complex diffuses as an intact entity within the nucleus. We also showed that it associates mainly with chromatin through the CHD4 (chromatin remodeling) subcomplex that serves to recruit the histone deacetylase subcomplex to chromatin. In principle, we can use the approach to study chromatin association/disassociation kinetics. However, for NuRD, although we were able to study the association kinetics at short timescales, we rather unexpectedly found that it disassociates from chromatin too slowly for us to measure the disassociation kinetics (with residence times on the order of a minute or more). Nevertheless, we anticipate that this approach will allow studies of the binding kinetics of proteins that do not interact with chromatin so tightly (for example, many transcription factors), and will become increasingly useful as the field develops more photostable fluorophores.

Using 500 ms exposure imaging, we then studied the movement of chromatin-bound NuRD molecules, where motion blurring prevents the imaging of NuRD molecules that are freely diffusing. We were able to observe two states of chromatin-bound molecules–a slow and a fast state–where the fast state being further divided into two substates: a condensed fast (F1) and decondensed fast (F2) state. Comparison of the movement of chromatin-bound CHD4 molecules in wild-type cells versus the *Mbd3*-knockout surprisingly suggested that chromatin is less condensed in the presence of NuRD, with chromatin-bound molecules diffusing more rapidly and exploring a larger nuclear volume. Moreover, in wild-type cells, we found that fewer of the chromatin-bound NuRD molecules exhibit movement in a defined direction, suggesting that the intact NuRD complex spends less time in this fast decondensed state. We confirmed that these motions are a property of NuRD-bound chromatin (as opposed to, for example, the movement

of NuRD molecules along DNA) by using a dead Cas9-GFP fusion protein to label the genome near specific enhancers and demonstrating that we could observe the same dynamics.

Finally, we used in-nucleus Hi-C and CUT&RUN experiments to ask whether altered enhancer dynamics, mediated by the intact NuRD complex, might affect genome architecture and enhancer–promoter interactions. We showed that the NuRD complex does indeed alter chromosome architecture by increasing the probability of intermediate-range contacts at the scale of TADs, leading to a blurring of the boundaries between A/B compartments and between TADs. We were also able to show that enhancer–promoter interactions that occur only in the presence of intact NuRD tend to link together genomic regions separated by longer distances. Moreover, we found that the NuRD complex leads to a marked redistribution of CTCF and, in particular, cohesin, with a significant proportion of newly formed CTCF/cohesin binding sites being found near to NuRD-regulated genes. We speculate that the NuRD complex promotes an environment with increased chromatin mixing where enhancers and promoters can contact each other over longer distances and where the resetting of enhancer–promoter interactions brought about by the fast decondensed F2 motions is reduced, leading to more stable, long-lived interactions (Fig. 6). This could provide an explanation for the observed increase in transcriptional noise, or low-level inappropriate transcription, observed in both human and mouse ES cells lacking functional NuRD[67,68].

## Online content

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

[1]Department of Biochemistry, University of Cambridge, Cambridge, UK. [2]Wellcome-MRC Cambridge Stem Cell Institute, Jeffrey Cheah Biomedical Centre, Cambridge, UK. [3]Department of Physiology, Development and Neuroscience, University of Cambridge, Cambridge, UK. [4]Department of Applied Mathematics and Computational Biology, Ecole Normale Supérieure, Paris, France. [5]Yusuf Hamied Department of Chemistry, University of Cambridge, Cambridge, UK. [6]The European Molecular Biology Laboratory EMBL, Grenoble, France. [7]Centre for Genomic Regulation (CRG), The Barcelona Institute of Science and Technology, Barcelona, Spain. [8]Helmholtz Zentrum München, German Research Center for Environmental Health, Institute of Functional Epigenetics, Neuherberg, Germany. [9]Institució Catalana de Recerca i Estudis Avançats (ICREA), Barcelona, Spain. [10]School of Biochemistry, University of Bristol, Bristol, UK. [11]MRC Laboratory of Molecular Biology, Cambridge Biomedical Campus, Cambridge, UK. [12]Present address: School of Physics and Astronomy, University of Leeds, Leeds, UK. [13]Present address: School of Food Science and Nutrition, University of Leeds, Leeds, UK. [14]Present address: Centre for Biodiversity Genomics, University of Guelph, Guelph, Ontario, Canada. [15]Present address: The European Molecular Biology Laboratory, Heidelberg, Germany. [16]Present address: Sylvester Comprehensive Cancer Center, Department of Human Genetics, University of Miami Miller School of Medicine, Biomedical Research Building, Miami, FL, USA. [17]These authors contributed equally: S. Basu, O. Shukron. ✉e-mail: dk10012@cam.ac.uk; bdh24@cam.ac.uk; holcman.david@gmail.com; e.d.laue@bioc.cam.ac.uk

## Methods

### In-nucleus chromosome conformation capture

**Data acquisition.** In-nucleus Hi-C[17] was carried out on E14 wild-type and 7g9 *Mbd3*-knockout (see Kaji et al.[31] for construction and characterization) ES cells; 50 bp paired-end sequencing was carried out on a HiSeq4000 instrument. Our own Hi-C analysis of *Mbd3*-knockout ES cells cultured in 2i consisted of three experiments: SLX-18035, SLX-7676 and SLX-19611 (which was sequenced twice). Alongside these four datasets, we also collected wild-type 2i data (SLX-7672) and compared this with recently published Hi-C data for ES cells cultured in 2i conditions (https://www.ebi.ac.uk/arrayexpress/experiments/E-MTAB-6591/) (Extended Data Fig. 8a). We used four replicates of this published data (ERR239137, ERR239139, ERR239141 and ERR239143; Supplementary Table 1).

**Preprocessing.** For each experimental condition and replicate, FASTQ files were first processed and aligned against the mouse GRCm38.p6 reference genome using the nuc_processing package (https://github.com/tjs23/nuc_processing). The number of unique contacts observed varied between replicates and conditions but, overall, a high number of reads were obtained in both conditions. The resulting raw contacts in NCC format were converted into hic and cooler format contact matrices using the juicer Pre (https://github.com/aidenlab/juicer/wiki/Pre) and HiCExplorer hicConvertFormat tools for downstream analysis (https://hicexplorer.readthedocs.io/en/latest/content/tools/hicConvertFormat.html)[69–71]. To assess the correlation between replicates within and between the two experimental conditions, we made use of the HiCExplorer hicPlotdistvscounts tool. Clustering of replicates based on contact distance distributions identified clusters for the two experimental conditions showing good agreement between replicates. We therefore merged the obtained contacts resulting in 705,063,441 and 327,180,884 contacts for the wild-type and *Mbd3*-knockout conditions, respectively. Owing to the difference in read coverage, we downsampled the merged wild-type dataset to have the same number of contacts as our merged *Mbd3*-knockout dataset. Finally, we performed Knight-Ruiz normalization of the merged datasets to ensure balanced matrices.

**Compartment analysis.** To identify A/B chromatin compartments within our merged datasets, we made use of the recently developed CscoreTool (https://github.com/scoutzxb/CscoreTool)[72]. This tool assigns each genomic window $i$ a score $\in [-1,1]$ with 1 assigning a probability 1 that window $i$ is in the A compartment while −1 assigns a probability 1 that window $i$ is in the B compartment. Importantly, unlike eigenvector analysis, c-scores of different samples can be compared directly since they represent probabilities. Binary compartments can also be assigned based on the sign of the c-score.

**TAD analysis.** TAD calling was performed using Lavaburst, a recently published tool[57] that uses an insulation-style metric called the TAD separation score to identify the degree of separation between the up and downstream region at each Hi-C matrix bin. Local minima of the TAD separation score are considered as putative TAD boundaries and assigned $q$-values for calculation of a false discovery rate. For this analysis, we used parameters gamma = 10 and beta = 50,000. For both datasets, we identified TADs using a binning resolution of 25 kb.

**Enhancer–promoter links.** We wanted to investigate whether transcriptional misregulation in *Mbd3*-knockout ES cells was significantly associated with a disruption of cis-regulatory interactions. To identify putative promoter-regulatory element (RE) interactions, we made use of a recent study[66] that profiled RE–promoter interactions via thousands of separate CRISPR deletions. In particular, Fulco et al.[66] found that a relatively simple activity-by-contact (ABC) model could be used to identify, with reasonable precision and recall, functional RE–promoter interactions. Crucially, the ABC model produced interactions

with higher accuracy than those identified using either linear distance along the genome or Hi-C contacts alone.

The ABC model considers the genome in 5 kb bins. For each promoter $p$ and regulatory element $r$ we define:

$$\mathrm{ABC}_{p,r} = \frac{A_r \times C_{p,r}}{\sum_{s \in N(p)} A_s \times C_{p,s}}$$

where $N(p)$ is the set of all regulatory elements within 5 Mb of $p$, $C_{p,r}$ is the contact frequency between $p$ and $r$, and $A_r$ is the activity of $r$. For regions of the genome with poor read coverage, $C_{p,r}$ is estimated assuming a power-law decay of contact frequencies:

$$C_{p,r} \propto d_{p,r}^{-\gamma}$$

where $d_{p,r}$ is the genomic distance between the promoter and regulatory element, and $\gamma$ is inferred from the Hi-C contact maps. Fulco et al.[66] define the activity of RE $r$ as the geometric mean of read counts of DNase I hypersensitive (DHS) and H3K27ac ChIP–seq at $r$. We did not have DHS tracks for our *Mbd3*-knockout ES cells and therefore implemented our own version of the ABC model where we used the read counts in just the H3K27ac ChIP–seq data to score the activity of each regulatory element. Specifically, we identified putative regulatory elements using H3K4me3 and H3K27ac ChIP–seq data from wild-type cells and knockout cells where MBD3 had been depleted[18] as follows:

(1) Promoter regions were assigned as those regions ±1kb of a transcription start site and overlapping with an H3K4me3 ChIP–seq peak.
(2) H3K27ac ChIP–seq peaks from that condition were considered as an initial putative list of condition-specific REs.
(3) H3K27ac peaks closer than 500 bp were merged.
(4) Peaks with total length <500 bp were discarded.
(5) Peaks overlapping with promoter regions were discarded.
(6) The union of the nonpromoter peaks and promoter regions was then treated as a master list of putative REs that were each assigned a condition-specific score based on the mean H3K27ac peak strength across that RE.

In particular, REs could be assigned within a single condition as being either 'intergenic' or 'promoter' associated. Since REs were defined per condition, wild-type-unique, knockout-unique and common overlapping peaks were assigned unique IDs for downstream analysis. The relevant code to perform this analysis and subsequent calculation of ABC scores can be found at https://github.com/dhall1995/Acitivity-By-Contact_Enhancer-Promoter_Link_Prediction.

Using calculated ABC scores, wild-type and knockout links were assigned unique link IDs based on the promoter and enhancer pair in question as well as the promoter genomic position in the case of a gene with multiple possible promoter regions. The top 10% of all identified links (wild-type and knockout) when ranked by ABC score were then selected as 'strong' links. In our data, this corresponded to an ABC threshold of ~0.12. Based on this threshold, unique link IDs could be assigned as wild-type-unique, knockout-unique or common depending on the conditions in which we observed that link. These thresholds were chosen to maximize the precision of identified links while retaining a large number of links to analyze although we acknowledge that maximizing precision comes at the expense of recall using the ABC model. Finally, after identification of wild-type-unique, knockout-unique and common links, enrichment analysis was performed by associating each link with a promoter and performing $\chi^2$ analysis using the statsmodels Python module.

### CUT&RUN

CUT&RUN experiments were carried out according to Meers et al.[73] using a *Drosophila* genomic DNA spike-in and 'input' controls consisting

of samples processed in parallel with the CUT&RUN samples but with untethered MNase. Western blots of nuclear lysates were also carried out[67] to measure the relative levels of CTCF and SMC3 in wild-type and *Mbd3*-knockout ES cells (Extended Data Fig. 1a). For the antibodies used see Supplementary Table 2.

We carried out 50 bp paired-end sequencing on a Novaseq instrument, with three biological replicates per sample obtaining 8–16 million mapped reads per replicate, respectively, whilst the input samples had 8–23 million mapped reads per replicate, respectively. All CUT&RUN data was trimmed using trim_galore (https://www.bioinformatics.babraham.ac.uk/projects/trim_galore/) and then aligned using Bowtie2 (ref. 74) with standard parameters to the GRCm38.p6 *Mus musculus* reference genome (https://www.ncbi.nlm.nih.gov/assembly/GCF_000001635.26/). Heatmaps of CUT&RUN enrichment were made using Deeptools v.2.5.0 (ref. 75). CUT&RUN bigwig tracks were calculated using BedTools and bedGraphToBigWig with standard parameters. The coverage was calculated with computeMatrix reference-point with options–binSize 10. The heatmap of standardized signal was then plotted using plotHeatmap. Peaks were called using MACS2 (ref. 76) so as to give a false discovery rate of 1% and above fivefold enrichment. All Venn diagrams were plotted using the matplotlib-venn library in Python v.3.6.

### mES cell line generation

mES cell lines were cultured in 2iL conditions[77] (50% DMEM/F-12 medium (Gibco catalog no. 21041025) and 50% Neurobasal Medium (Gibco catalog no. 12348017)) supplemented with 1× N2 to a final concentration of 2.5 µg ml$^{-1}$ insulin (provided in-house by the Cambridge Stem Cell Institute), 0.5× B-27 Supplement (Gibco catalog no. 17504044), 1× minimum essential medium nonessential amino acids supplement (Sigma-Aldrich, catalog no. M7145), 2 mM L-glutamine (Life Tech, catalog no. 25030024), and 0.1 mM 2-mercaptoethanol (Life Tech, catalog 21985023), 2i inhibitors (1 µM PD0325901, 3 µM CHIR99021) and 10 ng ml$^{-1}$ mouse leukemia inhibitory factor (mLIF provided by the Biochemistry Department, University of Cambridge). Cells were passaged every 2 days by washing in PBS (Sigma-Aldrich, catalog no. D8537), adding Trypsin-EDTA 0.25% (Life Tech, catalog no. 25200072) to detach the cells, and then washing in medium before replating in fresh medium. To help the cells attach to the surface, plates were incubated for 15 min at room temperature in PBS containing 0.1% gelatin (Sigma-Aldrich, catalog no. G1890). All cell lines were screened routinely for mycoplasma contamination at least twice yearly and tested negative.

ESC cells expressing CHD4 tagged at the C terminus with HaloTag were generated in the presence and absence of MBD3 (refs. 17,24,25). Briefly, this was achieved by CRISPR–Cas9 based knock-in of a cassette containing mEos3.2-HaloTag-Flag and a puromycin selection gene into one *CHD4* allele. The puromycin cassette was then removed using Dre recombinase to generate the *CHD4* allele with a C-terminal HaloTag fusion. Since knockout of *CHD4* is lethal, we used cell viability assays and the ability to immunoprecipitate NuRD component proteins (Extended Data Fig. 1c) to verify that the function of the tagged CHD4 was not impaired. We similarly generated knock-in ES cells in an E14Tg2a (XY) background expressing MBD3 tagged at the C terminus with HaloTag (Extended Data Fig. 1b). MTA2-HaloTag knock-in cell lines were generated in MBD3-inducible ES cells[18] (Extended Data Fig. 1d), in which MBD3 is fused to the estrogen receptor at both the N and C termini so that it initially localizes at the cytoplasm but then translocates to the nucleus when induced with 4-hydroxytamoxifen added directly to the culture medium to a final concentration of 0.4 nM. Western blots were carried out using nuclear lysates[18], to confirm the expression and assembly of the NuRD complex (Extended Data Fig. 1a). Immunoprecipitations were carried out using antibodies to CHD4 or MTA2, or Halo-Trap beads (ChromoTek) in the case of the CHD4-Halo line. For the antibodies used, see Supplementary Table 3.

### Live-cell 3D single-molecule imaging

ESCs expressing HaloTag-tagged CHD4, MBD3 and MTA2 were passaged 2 days before imaging onto 35 mm glass-bottom dishes No 1.0 (MatTek Corporation P35G-1.0-14-C Case) in serum/LIF imaging medium: Fluorobrite DMEM (Thermo Fisher Scientific, catalog no. A1896701) containing 100 mM 2-mercaptoethanol (Life Tech, catalog no. 21985023), 1× minimum essential medium nonessential amino acids (Sigma-Aldrich, catalog no. M7145), 2 mM L-glutamine (Life Tech, catalog no. 25030024), 1 mM sodium pyruvate (Sigma-Aldrich, catalog no. S8636-100ML), 10% fetal bovine serum (HyClone FBS, catalog no. (lot no.) SZB20006, GE Healthcare catalog no. SV30180.03) and 10 ng ml$^{-1}$ mLIF (provided by the Biochemistry Department, University of Cambridge). Glass-bottom dishes had been prepared by incubation in 0.01% poly-L-ornithine (Sigma-Aldrich catalog no. P4957) for 30 min, followed by two rinses in PBS and incubation in PBS containing 10 µg ml$^{-1}$ laminin (Sigma-Aldrich catalog no. L2020) for at least 4 h. Just before single-molecule imaging experiments, cells were labeled with 0.5 nM HaloTag-JF$_{549}$ ligand for 15 min, followed by two washes in PBS and a 30 min incubation at 37 °C in imaging medium, before imaging the cells in fresh serum/LIF imaging medium. Cells were underlabeled to prevent overlap of fluorophores during single-molecule tracking experiments. The HaloTag dyes were a kind gift from L.D. Lavis (Howard Hughes Medical Institute).

For the HaloTag-NLS control, the pEF-HaloTag-NLS vector was generated by replacing the HP1 sequence in a HaloTag-HP1 expression vector[78] with a SV40 NLS sequence. 1 µg of the pEF-HaloTag-NLS vector was transfected into ES cells using Lipofectamine 2000 Transfection Reagent (Thermo Fisher Scientific, catalog no. 11668019) during passaging onto 35 mm glass-bottom dishes No 1.0 (MatTek Corporation P35G-1.0-14-C Case). Media was changed after 24 h and samples were both labeled as above and imaged the following day.

Transcription elongation was inhibited using 100 µM 5,6-dichloro-1-β-D-ribofuranosylbenzimidazole (DRB) and deacetylase activity using 10 nM FK228 (TOCRIS Bioscience) both for 2 h before imaging[51,79].

A custom-made double-helix point spread function (DHPSF) microscope was then used for 3D single-molecule tracking[25]. The setup incorporates an index-matched 1.2 numerical aperture (NA) water immersion objective lens (Plan Apo VC ×60, Nikon) to facilitate imaging above the coverslip surface. The DHPSF transformation was achieved by the use of a 580 nm optimized double-helix phase mask (PM) (DoubleHelix) placed in the Fourier domain of the emission path of a fluorescence microscope (Eclipse Ti-U, Nikon). The objective lens was mounted onto a scanning piezo stage (P-726 PIFOC, PI) to calibrate the rotation rate of the DHPSF. A 4f system of lenses placed at the image plane relayed the image onto an EMCCD detector (Evolve Delta 512, Photometrics). Excitation and activation illumination was provided by 561 nm (200 mW, Cobolt Jive 100, Cobolt) and 405 nm (120 mW, iBeam smart-405-s, Toptica) lasers, respectively, that were circularly polarized, collimated and focused to the back focal plane of the objective lens. Oblique-angle illumination imaging was achieved by aligning the laser off axis such that the emergent beam at the sample interface was near-collimated and incident at an angle less than the critical angle ($\theta_c$ ~ 67°) for a glass–water interface. The fluorescence signal was then separated from the excitation beams into the emission path by a quad-band dichroic mirror (Di01-R405/488/561/635-25 ×36, Semrock) before being focused into the image plane by a tube lens. Finally, long-pass and band-pass filters (BLP02-561R-25 and FF01-580/14-25, respectively; Semrock) placed immediately before the camera isolated the fluorescence emission. Using 561 nm excitation, fluorescence images were collected as videos of 60,000 frames at 20 ms or 4,000 frames at 500 ms exposure. A continuous 561 nm excitation beam at ~1 kW cm$^{-2}$ was used for 20 ms exposure imaging and at ~40 W cm$^{-2}$ for 500 ms exposure imaging. Each experiment was carried out with at least three biological replicates (three fields of view, each containing around three cells).

## Residence time analysis from time-lapse 500 ms exposure imaging

Since photobleaching is related to the number of exposures, and the residence time is related to the time a molecule spends bound to chromatin, it is possible to change the time-lapse between exposures and use the data to extract both the residence time[1] and photobleaching rate. However, when we imaged at time intervals of 0.5 s, 2.5 s, 8 s and 32 s, we discovered that, at the longest time-lapse (32 s), we could see no decrease in the mean number of frames imaged before photobleaching, implying that the residence time had no impact on the measurement, which was thus dominated by photobleaching (Extended Data Fig. 4e). To estimate the residence time would probably require imaging at much longer time-lapses but, because chromosomes and the cell itself move during periods longer than this, it becomes unreliable to track individual chromatin-bound NuRD complex subunits.

## 3D single-molecule image processing, generation of trajectories and determination of experimental precision

Single molecules were localized from 3D videos using the easy-DHPSF software[80] with a relative localization threshold of 100 for all six angles for the 20 ms data and relative thresholds of 116, 127, 119, 99, 73 and 92 for the 500 ms data. The trajectories of individual molecules were then assembled using custom Python code for connecting localizations in subsequent frames if they were within 800 nm for 20 ms trajectories and within 500 nm for 500 ms trajectories (https://github.com/wb104/trajectory-analysis). This code also outputs average signal intensity per trajectory and trajectory lengths (OPTION -savePositionsFramesIntensities) and a summary of these data is reported in Extended Data Fig. 4b.

Our precision values (measured for fixed dye molecules on the coverslip and calculated using the approach described by Endesfelder et al.[81]) were 60 nm and 34 nm for the 20 and 500 ms tracking experiments, respectively. The lower limits of the effective diffusion coefficient ($D_{eff}$) one can measure are dependent on the precision values. The $D_{eff}$ is equal to the displacement squared over time. Thus, if the upper limit of the precision is 60 nm, then the lower limit of $D_{eff}$ that we can measure for 20 ms imaging is $0.06 \times 0.06/0.02 = 0.18$ m$^2$ s$^{-1}$. For 500 ms imaging on the other hand, the upper limit of our precision was 34 nm, corresponding to a lower limit of $D_{eff}$ that we can measure of $0.034 \times 0.034/0.5 = 0.002$ m$^2$ s$^{-1}$. Consistently, when we measured the $D_{app}$ values for dye molecules attached to a coverslip we determined values of $0.3 \pm 0.2$ μm$^2$ s$^{-1}$ and $0.004 \pm 0.003$ μm$^2$ s$^{-1}$, respectively (Extended Data Figs. 3c and 5a). Any measured diffusion coefficients below these values do not have a biophysical interpretation.

## Single-molecule trajectory analysis

The development of the algorithm to classify subtrajectories into confined and unconfined states based on four biophysical parameters using a GMM, and the use of this classification algorithm to analyze the single particle trajectories is described in the Supplementary Data and Methods.

## In vitro biochemical assays of the NuRD complex with and without nucleosomes

*Drosophila* PMMR and Human CHD4 were expressed in insect Sf21 cells and purified as described[24]. Sf21 cells expressing Human GAT-AD2A-MBP were resuspended in 50 mM Tris-HCl pH 7.5, 1 M NaCl, 5 mM DTT and 1× complete EDTA-free protease inhibitor cocktail (Roche), lysed by sonication and cleared by centrifugation at 50,000g for 1 h. The supernatant was applied to amylose resin pre-equilibrated with lysis buffer and incubated for 2 h with rotation at 4 °C. The resin was washed with 20 column volumes of lysis buffer and then eluted with 10 mM maltose in lysis buffer. Fractions containing hGATAD2A-MBP protein were concentrated and further purified by size exclusion chromatography using a Superose 6 Increase 3.2/300 column (GE Healthcare) equilibrated with 50 mM Tris-HCl pH 7.5, 150 mM NaCl, 5% glycerol and 1 mM DTT.

For pulldown experiments, purified protein was immobilized on MBP-Trap resin (ChromoTek) pre-equilibrated in pulldown buffer (50 mM HEPES pH 7.5, 300 mM NaCl, 1 mM DTT and 5% v/v glycerol) followed by incubation for 1 h with rotation at 4 °C. A sample of the 6% protein:bead mixture was retained as 'Input'. The resin was washed three times with pulldown buffer, then a washed 'beads' sample was retained for analysis on a 4–12% NuPAGE gel (Invitrogen).

Electrophoretic mobility shift assays were performed with n3-Widom-78bp DNA or recombinant nucleosomes made with this template[82,83] in 10 μl of binding buffer (20 mM Hepes pH 7.5, 2 mM MgCl$_2$, 5% glycerol and 1 mM TCEP) with varying concentrations of the indicated proteins. The reaction mixtures were incubated at 30 °C for 30 min followed by centrifugation at 1,000g. The resulting reaction mixtures were loaded onto 5% native polyacrylamide gels and run in 0.2× TBE. Gels were stained with SYBR Gold (Invitrogen) and imaged using a Typhoon FLA 9000 (GE Healthcare).

## dCas9-GFP imaging of enhancer loci

ESCs expressing dCas9 tagged with GFP were generated[3]. Briefly, MBD3-inducible ES cells were transfected with the PB-TRE3G-dCas9-eGFP-WPRE-ubcp-rtTA-IRES-puroR vector containing a dual promoter backbone, with a TRE3G (Tet-on) promoter expressing GFP-tagged inactive dCas9 and the ubiquitin C promoter expressing the reverse tetracycline-controlled transactivator, rtTA, and a puromycin cassette via an IRES sequence. Puromycin-resistant ES cells were then selected for 7 days and doxycycline was added for 24 h to induce expression of dCas9-GFP (through activation of the rtTA). Stable transfectants were then FACS sorted for low levels of GFP expression to select cells where only a few copies of the plasmid were integrated stably into the genome.

Before imaging, 1 μg ml$^{-1}$ of doxycycline was added to ES cells for 24 h to induce expression of low levels of dCas9-GFP. For imaging of *Tbx3* enhancer loci, three CARGO vectors in total expressing 36 gRNAs targeting the *Tbx3* enhancer were then transfected using lipofectamine 2000 (Invitrogen). The CARGO and dCas9-GFP expressing plasmids were gifts from the J. Wysocka laboratory. For imaging of *Nanog* enhancer loci, a custom designed gRNA was annealed with SygRNA Cas9 Synthetic Modified tracrRNA (Sigma-Aldrich catalog no. TRACRRNA05N). The gRNA was designed such that a single gRNA sequence could be used to uniquely target a repetitive sequence close to the relevant enhancer (Extended Data Fig. 6b). Cells were transfected with the tracr:crRNA complex using Lipofectamine 2000 (Invitrogen catalog no. 11668019). In all cases, cells were transfected during passaging straight onto imaging dishes in Fluorobrite imaging medium as described above. After 24 h, the medium was replaced with fresh medium and, for +MBD3 samples, 4-hydrooxytamoxifen was added to a final concentration of 0.4 nM. All samples were then imaged after a further 24 h.

2D tracking of genomic loci was carried out using oblique-angle illumination on a custom built 2D single-molecule tracking microscope[84]. Briefly, an IX73 Olympus inverted microscope was used with circularly polarized laser beams aligned and focused at the back aperture of an Olympus 1.40 NA ×100 oil objective (Universal Plan Super Apochromat, catalog no. UPLSAPO100XO/1.4). A 561 nm laser was used as a continuous wavelength diode laser light source. Oblique-angle illumination imaging was achieved by aligning the laser off axis such that the emergent beam at the sample interface was near-collimated and incident at an angle less than the critical angle ($\theta_c$ ~ 67°) for a glass/water interface. This generated a diameter excitation footprint of ~50 μm. The power of the collimated 488 nm beam at the back aperture of the microscope was 100 W cm$^{-2}$. The lasers were reflected by dichroic mirrors that also separated the collected fluorescence emission from the TIR beam (Semrock, Di01-R405/488/561/635). The fluorescence emission was collected through the same objective and then further filtered

using a combination of long-pass and band-pass filters (BLP01-561R and FF01-587/35). The emission signal was projected onto an EMCCD (Photometrics, Evolve 512 Delta) with an electron multiplication gain of 250 ADU per photon operating in a frame transfer mode. The instrument was automated using the open-source software micro-manager (https://www.micro-manager.org) and the data were displayed using the ImageJ software[85,86].

For image processing, PeakFit[87] was used to localize genomic loci from the images using the filter settings: 'shiftFactor':1.0, 'signalStrength':5.0, 'minPhotons':30.0, 'precisionThreshold':40.0, 'minWidthFactor':0.5, 'maxWidthFactor':0.5 and 'precisionMethod':'MORTENSEN'. Trajectories were then tracked in 2D using custom Python code for connecting foci in subsequent frames if they were within 500 nm (https://github.com/wb104/trajectory-analysis). Trajectories were classified as for single molecules using a GMM (see Single-molecule trajectory analysis in the Supplementary Data and Methods).

### Enhancer–promoter DNA-FISH
FISH probes were prepared from mouse BAC library clones (Source Biosciences)[88–101]. BAC vector DNA was purified using the Qiagen Large Construct Kit. BAC DNA was labeled using Cy3 and Alexa Fluor 647 Nick Translation Labeling Kits (Jena Bioscience, catalog nos. PP-305S-CY3, PP-305S-AF647) and purified[17]. The BAC probes generated are shown in Supplementary Table 7.

For two-color DNA-FISH, $2 \times 10^4$ cells were seeded per well on microscope slides with removable eight-well silicone chambers (Ibidi catalog no. 80841). Cells were fixed in 4% formaldehyde in PBS (Pierce catalog no. 28906, Thermo Fisher Scientific) for 10 min at room temperature, followed by permeabilization in 0.3% Triton X-100 (Sigma-Aldrich X100) in PBS for 15 min at room temperature. After three washes in PBS, cells were incubated in prewarmed 2× saline-sodium citrate (SSC) with 100 µg ml$^{-1}$ RNAse A (Qiagen, catalog no. 158922) for 1 h at 37 °C. Chambers were removed and slides washed in 2× SSC at room temperature. Slides were then dehydrated using 70% ethanol, 90% ethanol and 100% ethanol for 2 min each and left to air dry. Cells were denatured in 70% deionized formamide (Sigma-Aldrich catalog no. S4117) in 2× SSC at 80 °C for 15 min. Slides were then again dehydrated quickly through ice-cold 70% ethanol, 90% ethanol at room temperature and 100% ethanol at room temperature for 2 min each and again left to air dry.

For each sample slide, 150 ng of Cy3-labeled BAC probe and 150 ng of AF647-labeled BAC probe were precipitated with 5 µg of salmon sperm DNA (Invitrogen, catalog no. 15632011) using 0.1 volumes of 3 M sodium acetate (pH 5.2) and 2.5 volumes of 100% ethanol. Precipitated DNA was pelleted through centrifugation at 15,000g for 20 min and resuspended in 50 µl hybridization buffer (50% formamide, 10% dextran sulfate (Sigma-Aldrich, catalog no. 42867), 0.1% SDS, 2× SSC) through incubation for 1 h at 37 °C. Probes were denatured at 80 °C for 10 min and then transferred to 37 °C. Sample slides were overlaid with 50 µl hybridization solution and covered with Parafilm, after which hybridization was allowed to occur at 37 °C overnight in a humidity chamber. The following day, the coverslip and hybridization solution were removed, and the slides washed four times in 2× SSC at 40 °C for 3 min each, then four times in 0.1× SSC at 60 °C for 3 min. Slides were cooled by washing in 4× SSC at room temperature. After removing all the wash solution, cells were mounted in VECTASHIELD Antifade Mounting Medium with 4,6-diamidino-2-phenylindole (DAPI) (Vector Laboratories, catalog no. H-1200-10).

### Sequential immunofluorescence for dCas9-GFP and DNA-FISH
ESCs expressing dCas9 tagged with GFP were transfected with either CARGO plasmids or a gRNA as described above (dCas9-GFP imaging of enhancer loci). Fresh medium was added after 24 h and cells were fixed the following day in PBS containing 4% formaldehyde (Pierce, catalog no. 28908, Thermo Fisher Scientific) for 10 min at room temperature. Cells were permeabilized in PBS containing 0.5% Triton X-100

(Sigma-Aldrich X100) for 5 min, washed three times with PBS and then treated with blocking buffer (4% bovine serum albumin (Sigma-Aldrich, catalog no. A9418) in 0.1% Triton X-100 in PBS) for 30 min. Cells were incubated with GFP-Booster Alexa Fluor 488 nanobody (ChromoTek, catalog no. gb2AF488) in blocking buffer (1:1,000) through incubation for 1 h at room temperature. Samples were washed three times in PBS, each for 5 min.

Cells were postfixed in PBS containing 3% formaldehyde (Pierce, Thermo Fisher Scientific, catalog no. 28908) for 10 min at room temperature, followed by repermeabilization in 0.1 M HCl in 0.7% Triton X-100 in PBS for 10 min on ice. After two washes in 2× SSC for 5 min each, cells were incubated in prewarmed 2× SSC with 10 U ml$^{-1}$ RNAse A (Qiagen, catalog no. 158922) for 1 h at 37 °C. Slides were then equilibrated in 20% glycerol in PBS for 1 h, followed by three consecutive freeze–thaw cycles using liquid nitrogen. After incubation for 1 h in denaturing solution (50% formamide in 2× SSC) at room temperature, slides were denatured at 70 °C for 5 min and then washed several times in ice-cold 2× SSC. Probes were denatured at 70 °C for 10 min and then placed on ice to cool. Hybridization solution was prepared with 50 ng of BAC probe and 10 µg salmon sperm DNA (Invitrogen, catalog no. 15632011) per 100 µl of hybridization buffer (50% formamide, 10% dextran sulfate, 1 mg ml$^{-1}$ BSA and 2× SSC). Sample slides were overlaid with 25 µl hybridization solution per well and covered with Parafilm. After denaturation at 70 °C for 5 min on a heat block, the slide was gradually cooled to 37 °C and hybridization allowed to occur at 37 °C overnight in a humidity chamber. The following day, the coverslip and hybridization solution were removed, and the slides washed three times in 2× SSC at 40 °C for 5 min each, then three times in 2× SSC at room temperature for 5 min. After removing all the wash solution, cells were mounted in VECTASHIELD Antifade Mounting Medium with DAPI (Vector Laboratories).

### Immunofluorescence and DNA-FISH image acquisition and analysis
Imaging was carried out using a Zeiss LSM Airy Scan 2 super-resolution microscope set for imaging of DAPI (405 nm laser, 0.8%), Cy3 (514 nm laser, 10%) and AF647 (639 nm laser, 90%). Three stacks of horizontal plane images (38,04 × 3,804 pixels corresponding to 136.24 × 136.24 µm$^2$) with a z-step of 150 nm were acquired for each field of view. CZI image files were then imported into IMARIS v.9.6 (Bitplane) for 3D modeling. Quantitative analysis of interprobe distances within nuclei was carried out using the Surfaces and Spots modules of Imaris v.9.6.

### Software and code
The following tools were used for data collection: microscope image acquisition, Micro-manager (https://www.micro-manager.org); ImageJ software[85,86].

The following tools and methods were used: for Hi-C analysis - NucProcess (https://github.com/tjs23/nuc_processing), NucTools (https://github.com/tjs23/nuc_tools), Juicer (https://github.com/aidenlab/juicer), Cooler (https://cooler.readthedocs.io/en/latest/), CscoreTool (https://github.com/scoutzxb/CscoreTool) and Lavaburst (https://github.com/nvictus/lavaburst); for enhancer–promoter analysis (https://github.com/dhall1995/Acitivity-By-Contact_Enhancer-Promoter_Link_Prediction); for CUT&RUN analysis - Trim galore (https://www.bioinformatics.babraham.ac.uk/projects/trim_galore/), Bowtie2 (ref. 74), Deeptools v.2.5.0 (ref. 75) and MACS2 (ref. 76) (data were processed using the GRCm38.p6 mouse reference genome (https://www.ncbi.nlm.nih.gov/assembly/GCF_000001635.26/); for 2D single-molecule peak fitting - PeakFit[87]; for 3D single-molecule peak fitting - easy-DHPSF[80]; for trajectory analysis (https://github.com/wb104/trajectory-analysis); for trajectory overlay visualization - TrackMate; for GMM classification (https://zenodo.org/record/6497411#.YmlG-Fy8w3q0); for 3D DNA-FISH analysis - Imaris v.9.6.

## Biological materials

All constructs and cell lines are available upon request. The *Chd4*, *Klf4*, *Mbd3* and *Mta2* Eos-Halo targeting constructs have also been deposited with Addgene.

## Reporting summary

Further information on research design is available in the Nature Portfolio Reporting Summary linked to this article.

## Data availability

The single-molecule/locus imaging videos and XYZt single-molecule/locus trajectory data files are available at: https://zenodo.org/deposit/7985268 (https://doi.org/10.5281/zenodo.7985268). The Hi-C and Cut&Run datasets reported in this study are available from the Gene Expression Omnibus (GEO) repository under accession code GSE147789, and they were processed using the GRCm38.p6 mouse reference genome: (https://www.ncbi.nlm.nih.gov/assembly/GCF_000001635.26/). Source data are provided with this paper.

## Code availability

All the code developed in this project has been made freely available (for a description, see Methods and Supplementary Methods). The software repositories are at: https://github.com/wb104/trajectory-analysis, https://zenodo.org/record/6497411#.YmlGFy8w3q0 and https://github.com/dhall1995/Acitivity-By-Contact_Enhancer-Promoter_Link_Prediction.

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

## Acknowledgements

We thank T. Kretschmann for preparing the figures for publication, L. Lavis (Howard Hughes Medical Institute, Janelia Farm) for providing the JF549 dye, J. Wysocka (Stanford) for the *Tbx3* constructs used for 2D enhancer tracking, A. Riddell for flow cytometry and the CSCI imaging

(P. Humphreys and D. Clements) and DNA sequencing (M. Paramor and V. Murray) facilities. We thank K. Bowman, G. Brown and A. Crombie for preliminary computational analysis of NuRD-regulated genes and 2D enhancer tracking experiments, respectively. We thank the EU FP7 Integrated Project '4DCellFate' (277899 E.D.L., B.D.H., I.B., C.S. and L.D.C.), the Medical Research Council (MR/P019471/1 E.D.L.) and the Wellcome Trust (206291/Z/17/Z E.D.L.) for program funding. We also thank the MRC (MR/R009759/1 B.D.H., and MR/M010082/1 E.D.L.), the Wellcome Trust (106115/Z/14/Z I.B. and 210701/Z/18/Z C.S.) and the Isaac Newton Trust (17.24(aa) B.D.H.) for project grant funding, and we thank the Wellcome Trust/MRC for core funding (203151/Z/16/Z) to the Cambridge Stem Cell Institute (including a starter grant to S.B.).

## Author contributions

S.B. and O.S. contributed equally as first authors; D. Hall, P.P., A.P. and D. S. also contributed equally as co-second authors. S.B., B.D.H., D. Holcman and E.D.L. designed the experiments. D.L., S.B., W.B. and T.J.S. carried out the in-nucleus Hi-C data collection and processing. N.R., W.B., R.R., X.M. and B.D.H. carried out the CUT&RUN data collection and processing. D. Hall, R.R., X.M. and T.J.S. analyzed the ChIP-seq, CUT&RUN and Hi-C data. L.M., E.B. and L.D.C. carried out preliminary ChIP–seq experiments. B.D.H., N.R., J.C., R.F. and S.B. made and characterized the cell lines used for the live-cell single-molecule imaging. A.R.C., A.P., S.F.L. and D.K. designed and built the 3D DHPSF microscopes used for the live-cell single-molecule tracking experiments. L.-M.N., S.F.L. and D.K. designed and built the microscope for the 2D enhancer tracking. S.B., D.S. and L.H.S. carried out the live-cell 3D single-molecule imaging and analyzed the data together with O.S., P.P., D.K., B.D.H., E.D.L. and D. Holcman. S.B., D.S. and K.G. carried out the live-cell 2D tracking of enhancer movement. D.S., S.B., D.L. and M.S. carried out the immunofluorescence and DNA-FISH experiments. W.B. developed the code to generate 3D single-molecule tracks from the localization data and carry out dissociation time analysis. O.S., P.P., A.J. and D. Holcman developed the code for segmentation of the 3D single-molecule trajectories. The in vitro biochemical experiments were carried out by W.Z., J.B., T.A.D. and T.B., after purifying NuRD components and complexes expressed by A.A., G.C., I.B. and C.S. S.B., D. Holcman and E.D.L. wrote the manuscript with assistance from all the other authors.

## Competing interests

The authors declare no competing interests.

## Additional information

**Extended data** is available for this paper at https://doi.org/10.1038/s41594-023-01095-4.

**Correspondence and requests for materials** should be addressed to D. Klenerman, B. D. Hendrich, D. Holcman or E. D. Laue.

**a**

Mouse embryonic stem cell lines

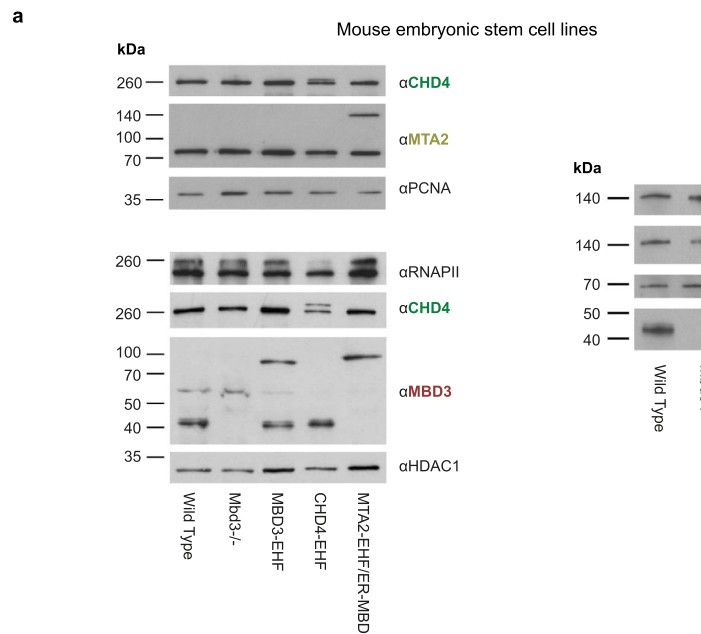

**b**

MBD3-mEos3.2-HaloTag-FLAG
/MBD3-FLAG

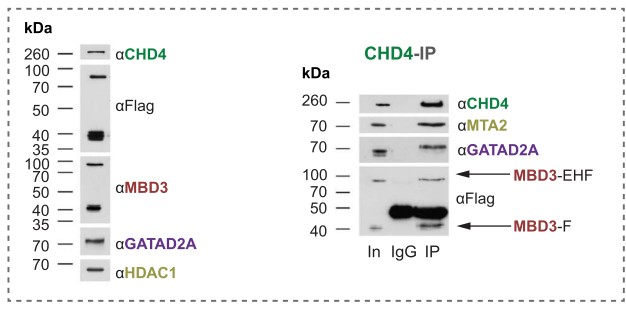

CHD4-IP

**c**

CHD4/
CHD4-mEos3.2-HaloTag-FLAG
Mbd3-flox/-

+ Cre
recombinase

CHD4/
CHD4-mEos3.2-HaloTag-FLAG
Mbd3-ko -/-

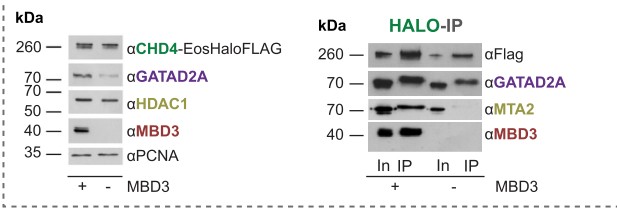

HALO-IP

**d**

MTA2-mEos3.2-HaloTag-FLAG
ER-MBD3-ER/-

+ Tamoxifen
(48 hr)

MTA2-mEos3.2-HaloTag-FLAG
ER-MBD3-ER/-

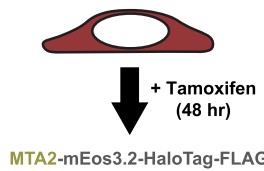

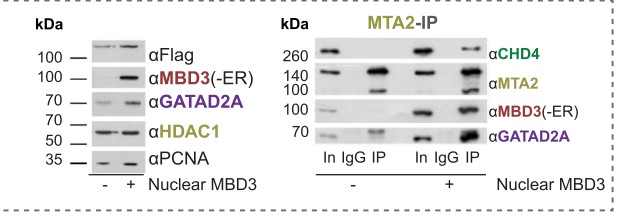

MTA2-IP

**Extended Data Fig. 1 | See next page for caption.**

**Extended Data Fig. 1 | Mouse embryonic stem cell lines expressing mEos3.2-HaloTag-FLAG tagged NuRD complex subunits. (a)** Western blot comparison of expression of (Left) NuRD components, and (Right) CTCF and Cohesin in the cell lines used. Detailed schematic of the **(b)** *Mbd3*, **(c)** *Chd4*[24] and **(d)** *Mta2* cell lines generated. MTA2 was tagged in ES cells expressing the ER-MBD3-ER (estrogen receptor-MBD3-estrogen receptor) fusion protein so that nuclear localisation of MBD3 is tamoxifen-inducible[18]. (Left) Expression of NuRD complex subunits was confirmed by western blot. Note that the stability of MTA2 and GATAD2A are both dependent upon MBD3, but that of CHD4 is not[68]. (Right) Immunoprecipitation of either CHD4 or MTA2 confirms that the Eos-Halo-FLAG tags do not prevent association with other NuRD components, and that NuRD complex integrity is dependent upon the presence of MBD3. Western blot images are representative of ≥3 independent replicates.

**a**   Maximum projection of 20ms CHD4 localisations    Maximum projection of 500ms CHD4 localisations

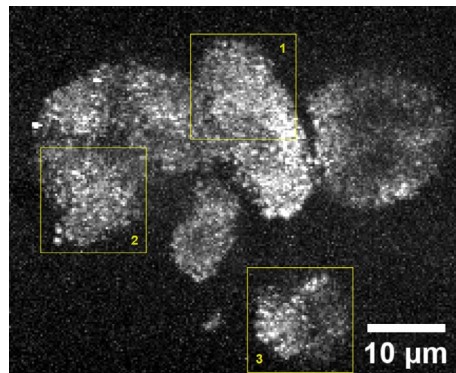 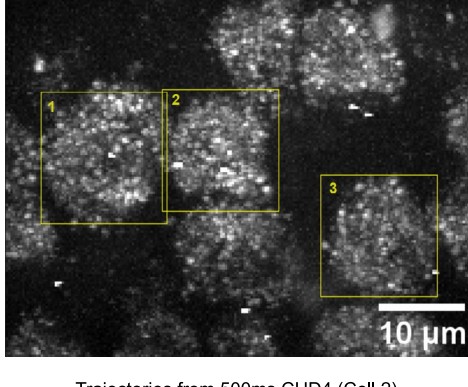

**b**   Trajectories from 20ms CHD4 (Cell 1)    Trajectories from 500ms CHD4 (Cell 2)

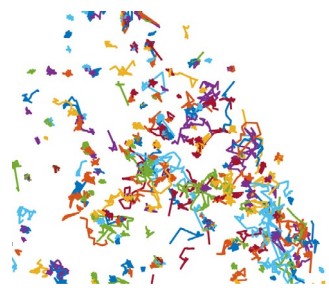 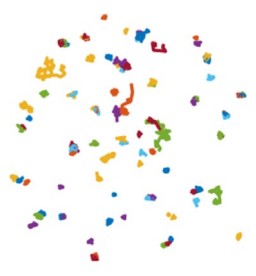

**c**   Classified Trajectories from 20ms CHD4 (Cell 1)    Classified Trajectories from 500ms CHD4 (Cell 2)

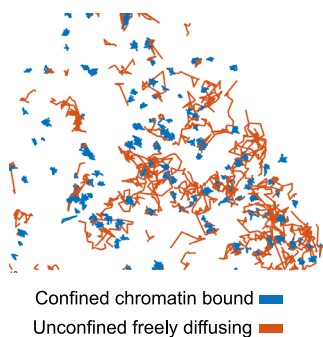 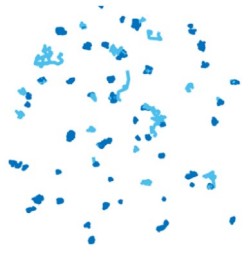

Confined chromatin bound ▬    Slow confined chromatin bound ▬
Unconfined freely diffusing ▬    Fast confined chromatin bound ▬

**Extended Data Fig. 2 | Representative single-molecule trajectories of CHD4-HaloTag-JF$_{549}$. (a)** Maximum projection images obtained when tracking CHD4 using either 20 ms (Left) or 500 ms (Right) exposures. Raw data for a single field of view (containing a single plane through several cells) can be found in Supplementary Videos 1 and 2 (obtained using 20 and 500 ms exposure imaging, respectively). The boxes in the Figures highlight cells for which smaller videos showing some of the raw data obtained from that cell after superimposition of the localisations and the resulting tracks following the data processing steps – see Supplementary Videos 3–5 and 6–8 (obtained using 20 and 500 ms exposure imaging, respectively). **(b)** Trajectories of the localisations obtained from the videos of the indicated cells, with different colours indicating different trajectories. **(c)** Trajectories classified using the Gaussian Mixture Model for the cells shown in (b) with the colours now representing the classification as indicated by the keys below.

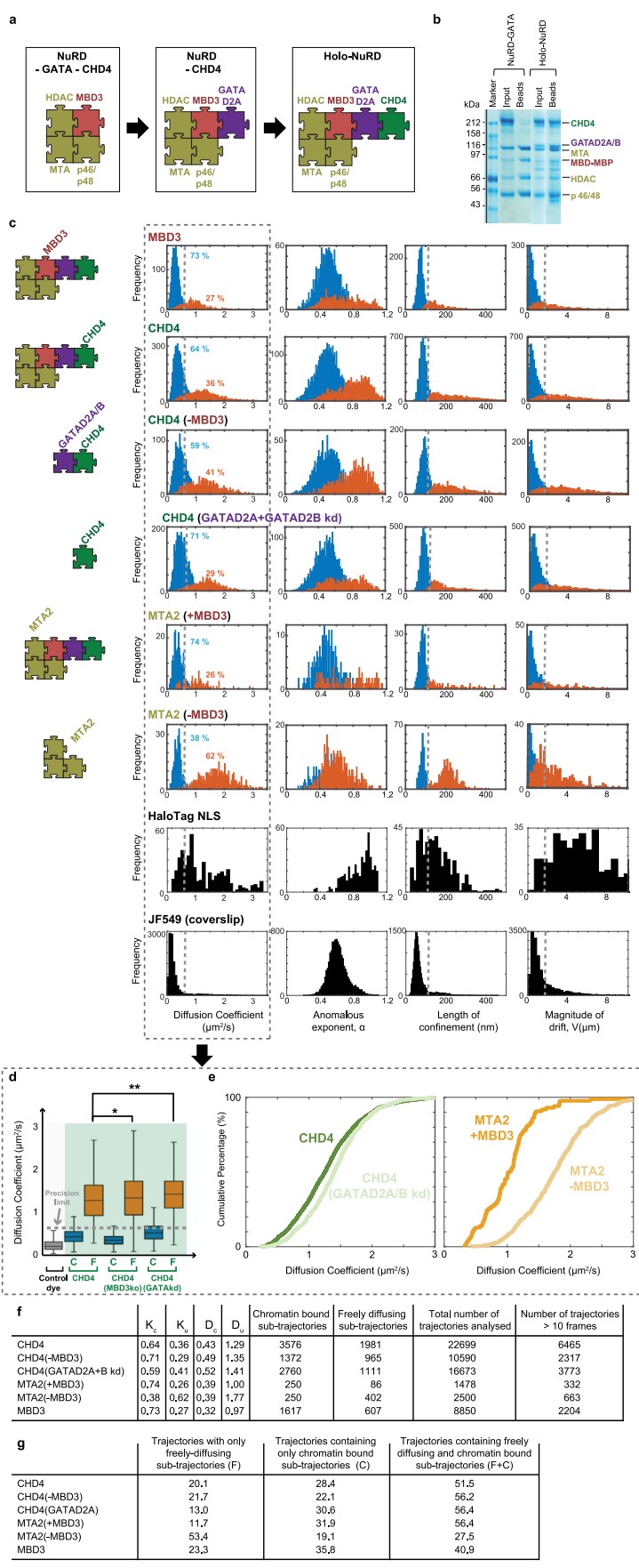

**Extended Data Fig. 3 | See next page for caption.**

**Extended Data Fig. 3 | *In vitro* and live cell single-molecule imaging experiments delineate holo-NuRD complex assembly. (a)** Schematic of holo-NuRD complex assembly with GATAD2A linking MBD3 to the CHD4 remodeler. **(b)** Pull-down experiments of MBP-tagged MBD with and without GATAD2A confirm that GATAD2A is required for CHD4 to interact with the deacetylase sub-complex. Pull-down images are representative of ≥3 independent replicates. **(c)** Distribution of the four biophysical parameters described in Fig. 1 for 20ms exposure tracking of MBD3 and CHD4 in wild-type ES cells, as well as CHD4 in the absence of either MBD3 or GATAD2A/B. The data for MTA2 in the presence and absence of nuclear localised MBD3 are also shown. HaloTag with a nuclear localisation sequence (HaloTag-NLS) is also shown as a control for a (mostly) freely diffusing molecule[42,43]. The grey dotted lines indicate the upper bound (at the 95 % confidence interval) of the different biophysical parameters determined for stationary $JF_{549}$ dye molecules. **(d)** Boxplot of apparent diffusion coefficients extracted from chromatin bound (C) and freely diffusing (F) CHD4 molecules in wild-type, *Mbd3* knockout and GATAD2A/B knock-down ES cells. The number of chromatin bound/freely

diffusing sub-trajectories used in the analysis were: 3576/1981 (CHD4), 1372/965 (CHD4-MBD3), 2760/1111 (CHD4-GATAD2A/B); *$p = 0.009$, **$p = 10^{-25}$, two sided Kolmogorov-Smirnov test. Center line, median; box limits, upper and lower quartiles; whiskers, 95 % confidence interval. The grey dotted line indicates the upper bound of the precision limit calculated at the 95 % confidence interval for an immobilised $JF_{549}$ dye control sample (11,313 sub-trajectories). **(e)** Cumulative distribution functions showing a higher diffusion coefficient for freely diffusing unconfined CHD4 upon removal of GATAD2A/B, and for freely diffusing MTA2 molecules upon removal of MBD3 from the nucleus. **(f)** Table showing the proportions (K) and estimated values of the apparent diffusion coefficients (D), as well as the number of chromatin bound and freely diffusing sub-trajectories obtained from the total number of trajectories analysed. (NB – many trajectories were discarded as they were either too short for analysis or because they had a low probability of being classified as confined or unconfined.) **(g)** Table showing the proportions of trajectories containing either freely diffusing, confined or both freely diffusing and confined sub-trajectories.

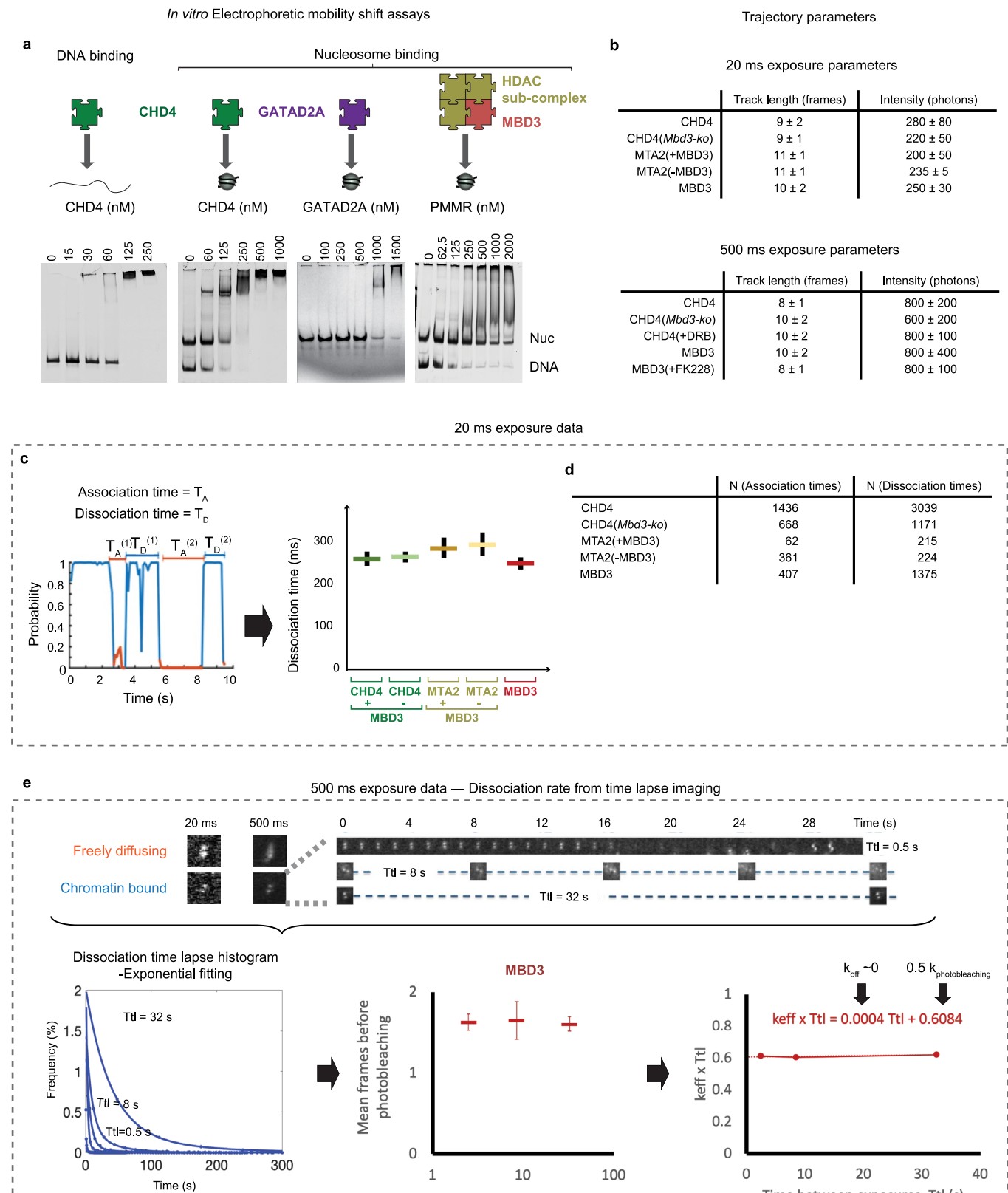

**Extended Data Fig. 4 | See next page for caption.**

**Extended Data Fig. 4 | The NuRD complex mainly interacts with both DNA and nucleosomes through the CHD4 remodeler. (a)** *In vitro* electrophoretic mobility shift assays confirm that CHD4 binds to both DNA and nucleosome core particles (NCPs) to form large complexes that only just enter the gel. GATAD2A alone shows low affinity binding to NCPs whilst the deacetylase complex interacts with DNA, but does not bind stably to NCPs. Electrophoretic mobility shift assay images are representative of ≥3 independent replicates. **(b)** Table showing the mean track length in frames, and mean photons detected per image frame, in 20 ms (Top) and 500 ms (Bottom) exposure trajectories. **(c)** (Left) Confinement probability allows collection of the association $T_A$ or dissociation $T_D$ times – defined respectively as the time a trajectory spends between periods of confined or unconfined motion. (Right) Dissociation times calculated using transitioning trajectories as periods of confined motion between two periods of unconfined motion (see also Fig. 3). The number of dissociation times used in the analysis were: 3039 (CHD4), 1171 (CHD4-MBD3), 215 (MTA2),

224 (MTA2-MBD3) and 1375 (MBD3). Data presented as mean values. Error bars show 95 % confidence intervals. **(d)** Table with the number of single molecule tracks that were used to determine the association and dissociation times. **(e)** (Top) Example images demonstrating how long 500 ms exposures motion blur freely diffusing molecules, but allow detection and tracking of those that are chromatin bound. Images are of single chromatin bound MBD3 molecules during time-lapse imaging with various dark times. (Bottom) Exponential fitting of time-lapse residence time histograms can be used to extract the photobleaching rate $k_b$ and the effective dissociation rate $k_{eff}$. However, examination of the mean number of frames before photobleaching for MBD3, where the time between exposures is varied, shows that the results are completely dominated by photobleaching. The number of tracked localisations used in the analysis were: 12922 (2 s), 6793 (8 s), 4215 (32 s). Data presented as mean values. Error bars show 95 % confidence intervals.

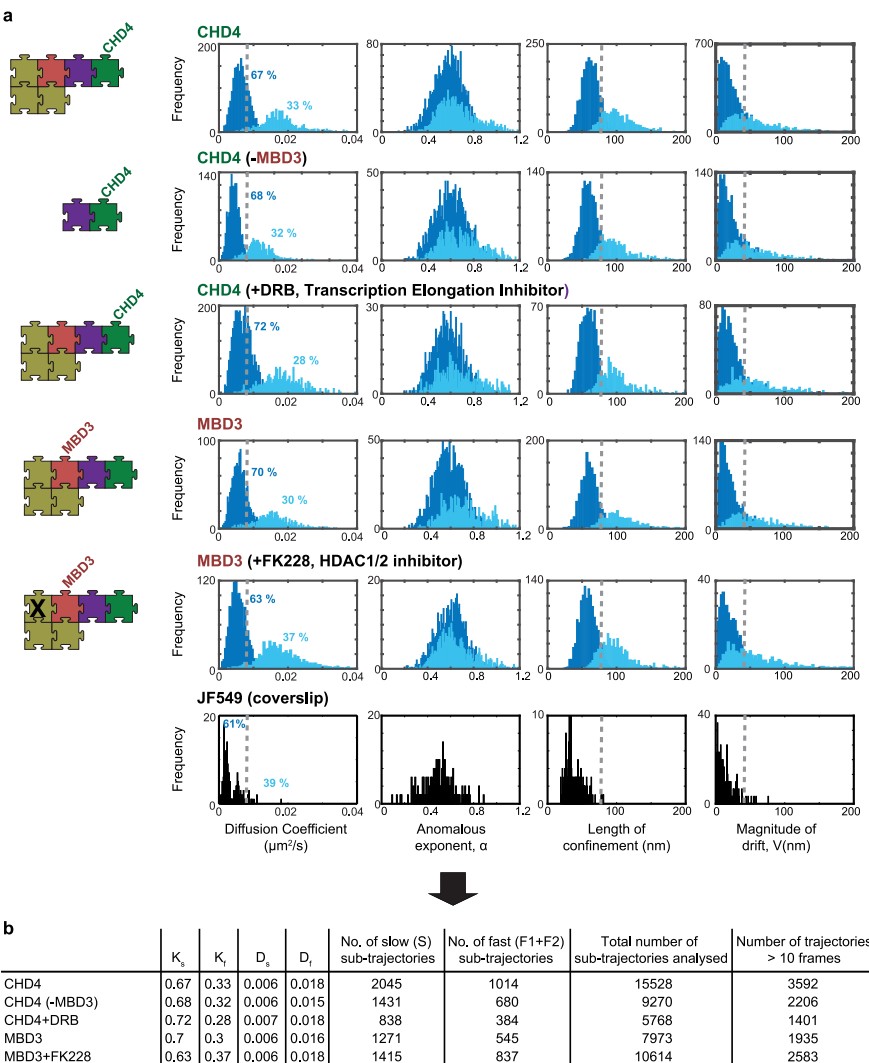

**a**

**CHD4**

**CHD4 (-MBD3)**

**CHD4 (+DRB, Transcription Elongation Inhibitor)**

**MBD3**

**MBD3 (+FK228, HDAC1/2 inhibitor)**

**JF549 (coverslip)**

Diffusion Coefficient (µm²/s) | Anomalous exponent, α | Length of confinement (nm) | Magnitude of drift, V(nm)

**b**

| | $K_s$ | $K_f$ | $D_s$ | $D_f$ | No. of slow (S) sub-trajectories | No. of fast (F1+F2) sub-trajectories | Total number of sub-trajectories analysed | Number of trajectories > 10 frames |
|---|---|---|---|---|---|---|---|---|
| CHD4 | 0.67 | 0.33 | 0.006 | 0.018 | 2045 | 1014 | 15528 | 3592 |
| CHD4 (-MBD3) | 0.68 | 0.32 | 0.006 | 0.015 | 1431 | 680 | 9270 | 2206 |
| CHD4+DRB | 0.72 | 0.28 | 0.007 | 0.018 | 838 | 384 | 5768 | 1401 |
| MBD3 | 0.7 | 0.3 | 0.006 | 0.016 | 1271 | 545 | 7973 | 1935 |
| MBD3+FK228 | 0.63 | 0.37 | 0.006 | 0.018 | 1415 | 837 | 10614 | 2583 |

**c**

**Anomalous exponents of and percentage of molecules in S, F1 and F2 states**

| | Slow confined (S) | | Fast confined (F1) | | Fast confined (F2) | | N |
|---|---|---|---|---|---|---|---|
| CHD4 | 0.593 ± 0.007 | (67 %) | 0.601 ± 0.014 | (25.8 ± 1.1 %) | 0.89 ± 0.02 | (7.4 ± 0.6 %) | 15528 |
| CHD4 (-MBD3) | 0.598 ± 0.006 | (68 %) | 0.559 ± 0.019 | (14 ± 6 %) | 0.77 ± 0.12 | (18 ± 5 %) | 9270 |
| CHD4+DRB | 0.598 ± 0.006 | (72 %) | 0.556 ± 0.015 | (19 ± 3 %) | 0.90 ± 0.04 | (8.3 ± 1.1 %) | 5768 |
| MBD3 | 0.580 ± 0.004 | (70 %) | 0.650 ± 0.016 | (22 ± 2 %) | 0.97 ± 0.03 | (8.0 ± 1.5 %) | 7973 |
| MBD3+FK228 | 0.580 ± 0.004 | (63%) | 0.572 ± 0.010 | (28 ± 5 %) | 0.82 ± 0.06 | (8.8 ± 1.8 %) | 10614 |

**d**

**Gaussian fitting of fast-diffusing anomalous exponents**

**Anomalous exponents of CHD4**

1 Gaussian R² = 0.9453 | 2 Gaussians R² = 0.9936 | 3 Gaussians R² = 0.9966

**Bayesian information criterion (BIC) from Gaussian fitting**

| | No. of Gaussians (n) | | |
|---|---|---|---|
| | n = 1 | n = 2 | n = 3 |
| CHD4 | -721 | -811 | -795 |
| CHD4 (-MBD3) | -436 | -471 | -457 |
| CHD4+DRB | -199 | -219 | -204 |
| MBD3 | -313 | -342 | -324 |
| MBD3+FK228 | -643 | -730 | -710 |
| Coverslip | -82 | -78 | -69 |

**e**

| | Slow-moving (S) | Fast-moving (F1) | Fast-moving (F2) | Transitioning (S+F1) | Transitioning (S+F2) | Transitioning (F1+F2) | Transitioning (S+F1+F2) |
|---|---|---|---|---|---|---|---|
| CHD4 | 28.6 | 2.7 | 0.8 | 10.5 | 0.4 | 8.9 | 48.1 |
| CHD4 (-MBD3) | 25.5 | 1.8 | 1.0 | 9.5 | 0.6 | 8.7 | 52.9 |
| CHD4+DRB | 24.6 | 2.0 | 0.5 | 15.0 | 0.0 | 7.8 | 50.1 |
| MBD3 | 27.3 | 1.4 | 0.8 | 9.8 | 0.3 | 9.0 | 51.5 |
| MBD3+FK228 | 27.9 | 4.2 | 0.7 | 10.0 | 0.4 | 12.0 | 44.9 |

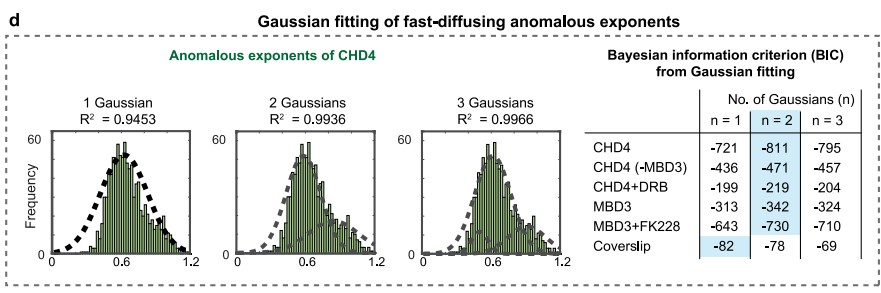

**Extended Data Fig. 5 | See next page for caption.**

**Extended Data Fig. 5 | Live cell single-molecule imaging experiments to study the chromatin-bound NuRD complex. (a)** Distribution of the four biophysical parameters from 500 ms exposure tracking of: (i) chromatin bound CHD4 in wild-type ES cells, in the absence of MBD3, and in the presence of DRB (an inhibitor of transcriptional elongation); (ii) chromatin bound MBD3 in wild-type ES cells, and in the presence of the HDAC1/2-specific inhibitor FK228; and (iii) JF549 dye bound to the coverslip. The grey dotted lines indicate the upper bound (at the 95% confidence interval) of the different biophysical parameters determined for stationary JF$_{549}$ dye molecules. **(b)** Table showing the proportions (K) and estimated values of the apparent diffusion coefficients (D), as well as the number of slow (S) and fast (F1+F2) diffusing sub-trajectories obtained from the total number of trajectories analysed. (NB – many trajectories were discarded as they were either too short for analysis or because they had a low probability of being classified as slow or fast.) **(c)** Table summarising the changes in anomalous exponent of the slow and fast chromatin bound NuRD complex subunits in the presence and absence of MBD3, or in the presence of specific inhibitors. Errors given are for 95% confidence intervals. **(d)** (Left) Fitting of 1, 2 or 3 Gaussians to the anomalous exponent distributions for fast moving chromatin bound CHD4 in wild-type ES cells – the $R^2$ values indicate the goodness of fit. (Right) The Bayesian information criterion (BIC) was calculated for all the datasets shown in (a) and shows that two populations (Gaussians) are the best model to account for the data – *that is* that model has the lowest BIC value (light blue box). **(e)** Table showing the proportions of trajectories containing either slow (S), fast (F1), or fast (F2) chromatin bound sub-trajectories (or combinations thereof).

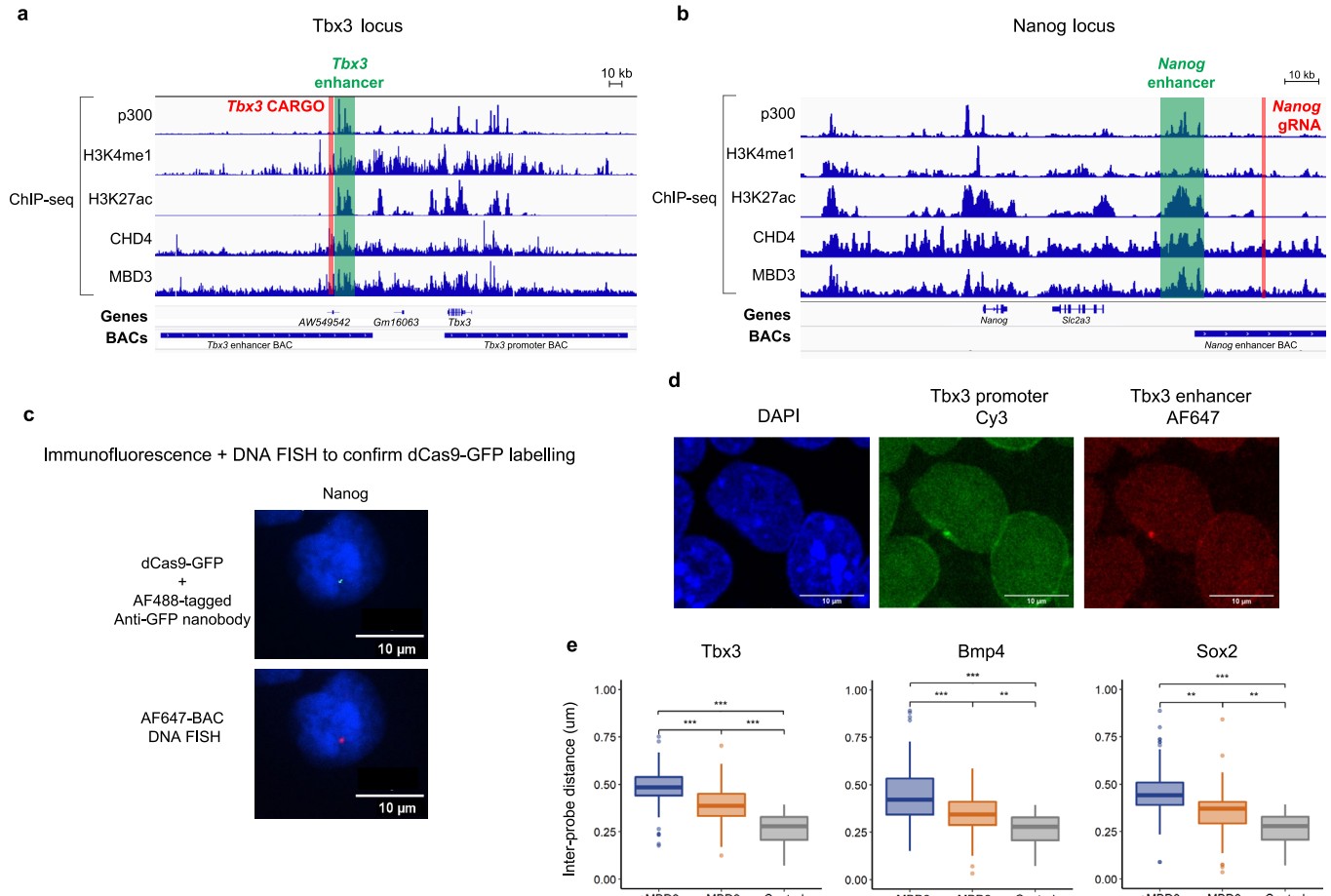

**Extended Data Fig. 6 | DNA FISH studies of NuRD-dependent changes in enhancer-promoter interactions.** Genomic locations of the **(a)** *Tbx3* and **(b)** *Nanog* genes annotated with the locations to which dCas9-GFP was targeted using either CARGO vectors[3] or a single gRNA that targets nearby genomic repeats (red lines). Locations of bacterial artificial chromosome (BAC) DNA FISH probes for the *Tbx3* and *Nanog* enhancers are also indicated as are the targeted enhancers themselves (green). The corresponding ChIP-seq profiles indicate the binding of the NuRD complex subunits CHD4 and MBD3 as well as the location of active enhancers (determined from the ChIP-seq profiles for H3K27ac, H3K4me1 and p300). **(c)** Representative confocal images showing co-localisation of dCas9-GFP (labelled using a AF488-tagged anti-GFP nanobody) and AF647-BAC DNA FISH probes targeting the *Nanog* enhancer. Images are representative of two independent replicates. **(d)** Representative confocal images of Cy3-labelled

BAC DNA FISH probes targeting the *Tbx3* promoter and AF647-labelled BAC DNA FISH probes targeting the *Tbx3* enhancer. Images are representative of two independent replicates. **(e)** Boxplots showing the enhancer-promoter distances in MBD3-inducible mESCs with and without tamoxifen: +MBD3 (blue) and -MBD3 (orange) respectively. There is a significant increase in enhancer-promoter distance in the presence of intact NuRD for *Tbx3* (+MBD3, n = 70; -MBD3, n = 101), *Bmp4* (+MBD3, n = 172; -MBD3, n = 71) and *Sox2* (+MBD3, n = 42; -MBD3, n = 50) (** $p < 0.01$, *** $p < 10^{-6}$, two-sided t-test). (Center line, median; box limits, upper and lower quartiles; whiskers, 95% confidence interval.) To estimate the precision limit of the experiment, control samples were generated in which the distance was measured for the *Sox2* enhancer labelled with both Cy3 and AF647 (grey, n = 32).

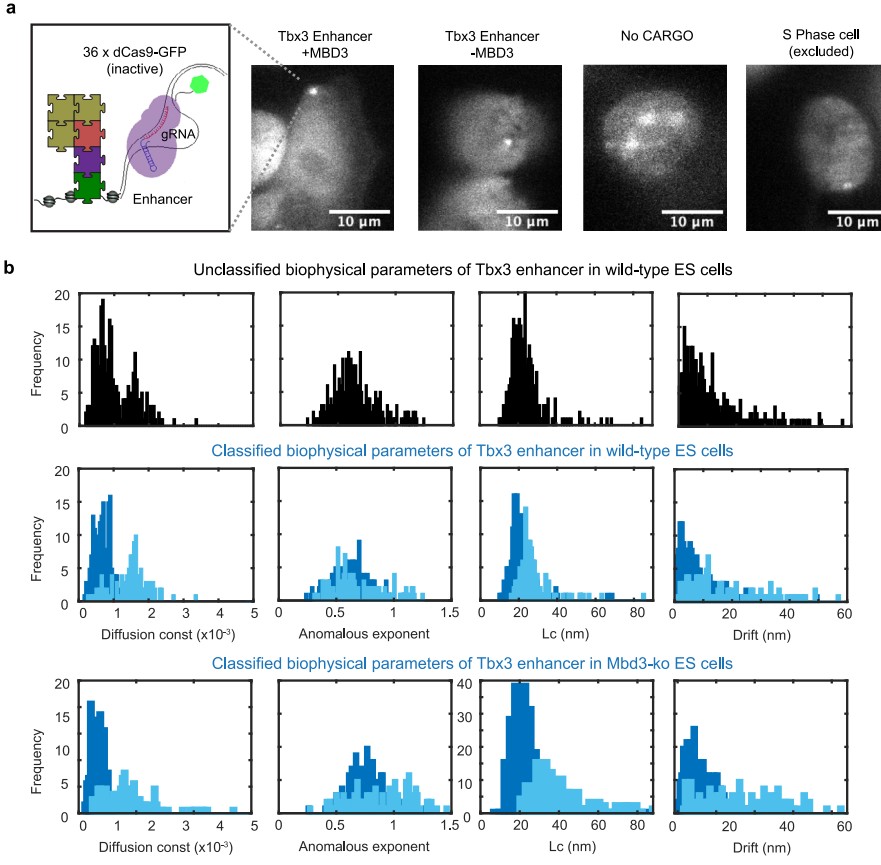

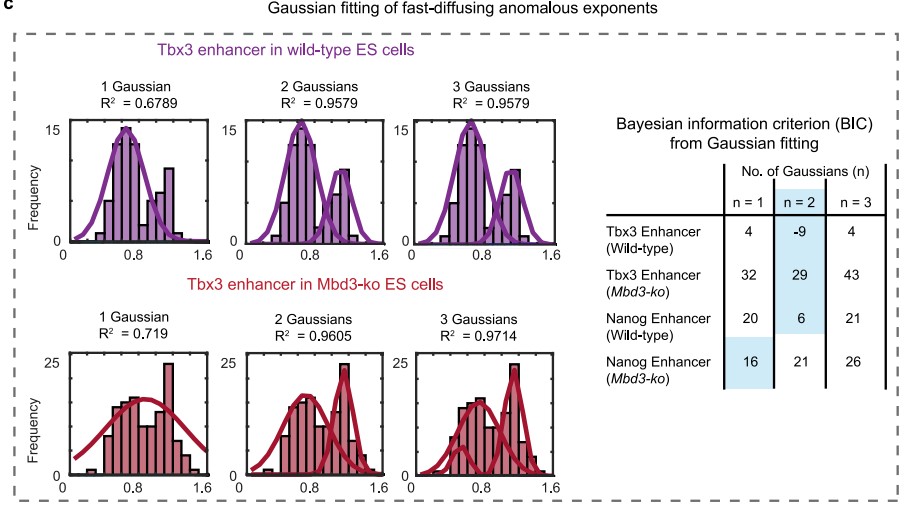

**d**    Anomalous exponents of and percentage of molecules in S, F1 and F2 states

| | Slow confined (S) | | Fast confined (F1) | | Fast confined (F2) | | N |
|---|---|---|---|---|---|---|---|
| Tbx3 Enhancer Wild-type | 0.675 ± 0.008 | (71.73%) | 0.635 ± 0.001 | (18.47%) | 1.071 ± 0.001 | (9.80%) | 237 |
| Tbx3 Enhancer *Mbd3-ko* | 0.754 ± 0.004 | (57.49%) | 0.621 ± 0.007 | (21.39%) | 1.075 ± 0.007 | (21.12%) | 287 |
| Nanog Enhancer Wild-type | 0.720± 0.027 | (61.17%) | 0.677 ± 0.025 | (21.48%) | 1.087 ± 0.013 | (17.35%) | 546 |
| Nanog Enhancer *Mbd3-ko* | 0.773 ± 0.029 | (58.08%) | 0.712 ± 0.028 | (20.38%) | 1.128 ± 0.018 | (21.54%) | 229 |

**Extended Data Fig. 7 | See next page for caption.**

**Extended Data Fig. 7 | 2D dCas9-GFP tracking of active enhancers.**
**(a)** Representative images of 36 gRNAs targeted to the *Tbx3* enhancer in the presence or absence of MBD3 with a negative control expressing no gRNAs. (Right) An example of a cell with a doublet indicating that it is in S phase (that was excluded from the analysis) is also shown. Images are representative of ≥3 independent replicates collected over ≥2 days. **(b)** Distribution of the four biophysical parameters extracted from sliding windows within the 2D single-molecule trajectories of dCas9-GFP bound at the *Tbx3* enhancer in wild-type ES cells imaged using 500 ms exposures – (Top) before and (Middle) after classification based on the anomalous exponent α, the apparent diffusion coefficient D, the length of confinement Lc, and the drift magnitude, norm‖V‖

of the mean velocity. (Bottom) Distribution of the four biophysical parameters after classification for the *Tbx3* enhancer in *Mbd3-ko* cells. **(c)** (Left) Fitting of 1, 2 or 3 Gaussians to the fast-moving anomalous exponent distributions of the *Tbx3* enhancer tracked in either wild-type ES (Top) or *Mbd3-ko* cells (Bottom) – the $R^2$ values indicate the goodness of fit. (Right) The Bayesian information criterion (BIC) was calculated for all the datasets to determine which number of populations (Gaussians) best modelled the data, with the lowest BIC value indicated by a light blue box. **(d)** Table showing the Gaussian fitted anomalous exponent values for slow- and the fast-moving chromatin bound dCas9-GFP at both enhancers tracked in the presence and absence of MBD3.

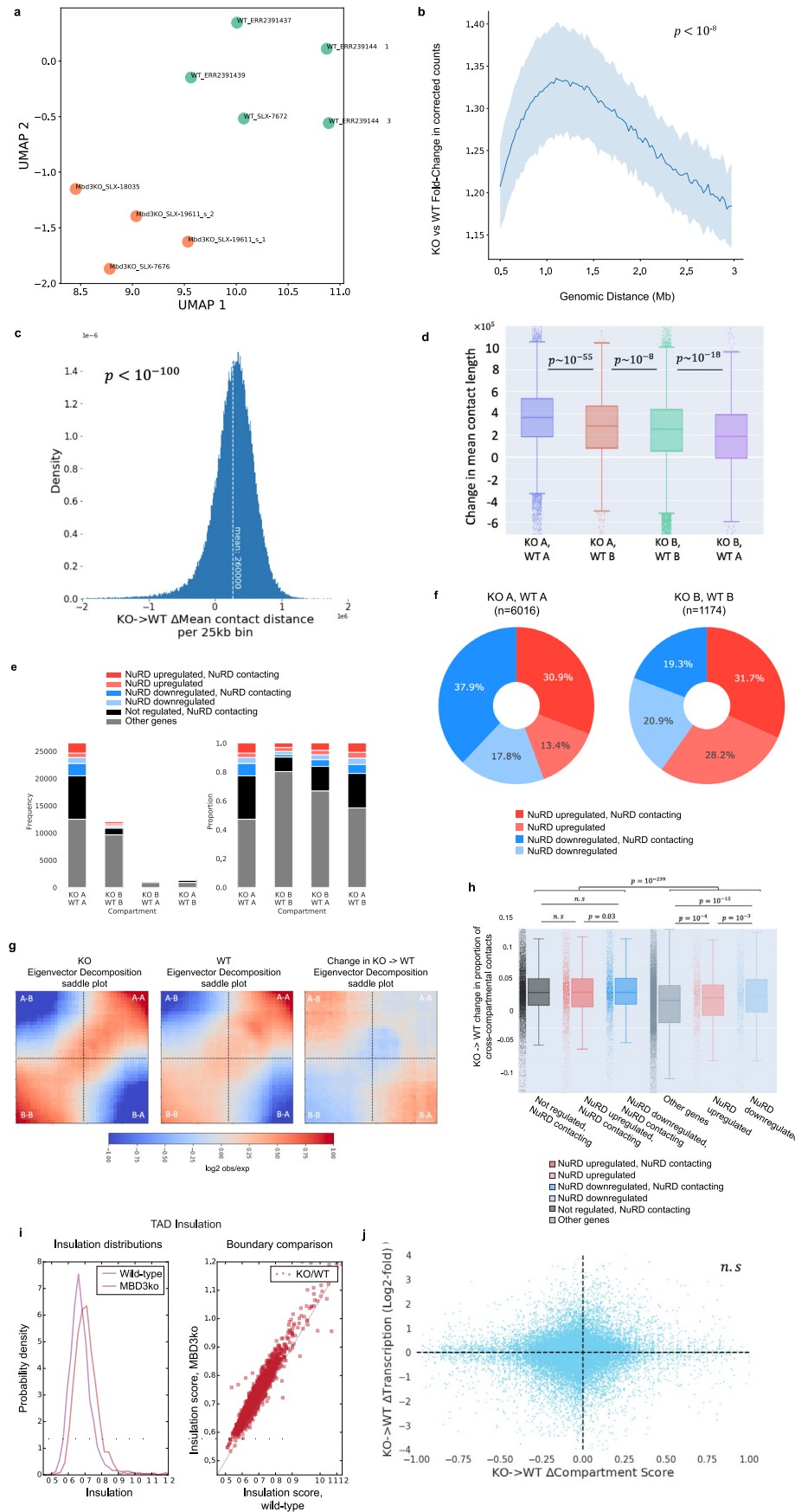

**Extended Data Fig. 8 | See next page for caption.**

**Extended Data Fig. 8 | MBD3-dependent assembly of the NuRD complex increases Mb-range chromosomal interactions. (a)** UMAP projection plot showing the reproducibility between our own and previously published (E-MTAB-6591) Hi-C datasets for wild-type (blue) mESC's grown in 2i/LIF conditions, and comparison with those from the *Mbd3-ko* (green). **(b)** The fold change in numbers of contacts at a range of genomic distances shows an increase in intermediate-range (~1 Mb) contacts in the presence of the intact NuRD complex ($p < 10^{-8}$, for all tested length scales, two sided Mann-Whitney U-test). Error bands depict the 95% confidence interval. **(c)** Histogram showing the difference in mean contact distance for 25 kb genomic regions between the *Mbd3-ko* and wild-type cells [$p < 10^{-100}$, the Bayesian version of the t-test (BEST) estimated a 95% probability of an effect size >= 652 kb]. **(d)** Boxplot showing the changes in mean contact length for genomic regions that are in the A compartment in both wild-type and *Mbd3-ko* cells (blue), are in the B compartment in both conditions (green), that switch from A to B compartment in the presence of the intact NuRD complex (red), or that switch from B to A compartment in the presence of the intact NuRD complex (purple). The number of 'WT A, KO A' contacts = 7911, 'WT A, KO B' contacts = 365, 'WT B KO A'

contacts = 2153, 'WT B, KO B' contacts = 15997; *p*-values (two sided t-test) were as indicated. **(e)** Bar and **(f)** pie charts showing the numbers of genes that remain or switch between compartments with the percentages of those genes that are up-/down-regulated. Saddle plots **(g)** show that there is an increase in inter-compartment contacts when going from the *Mbd3-ko* to wild-type cells, and thus A/B compartment mixing, and boxplots **(h)** show that this is more noticeable in regions where NuRD-bound enhancers may contact promoters. The number of 'NuRD contacting, NuRD upregulated' contacts = 2231; 'NuRD contacting, NuRD downregulated' contacts = 2506; 'NuRD contacting, no significant regulation' contacts = 8431; 'NuRD upregulated, no NuRD contact' contacts = 1137; NuRD downregulated, no NuRD contact' contacts = 1316; 'No NuRD association' contacts = 25293; *p*-values (two-sided t-test) were as indicated. In (d) and (h): center line, median; box limits, upper and lower quartiles; whiskers, 95% confidence interval. **(i)** Insulation scores derived from the contacts in the Hi-C data show a global decrease in TAD insulation in the presence of the intact NuRD complex. **(j)** A correlation plot shows that changes in A/B inter-compartment contacts do not correlate with changes in transcription.

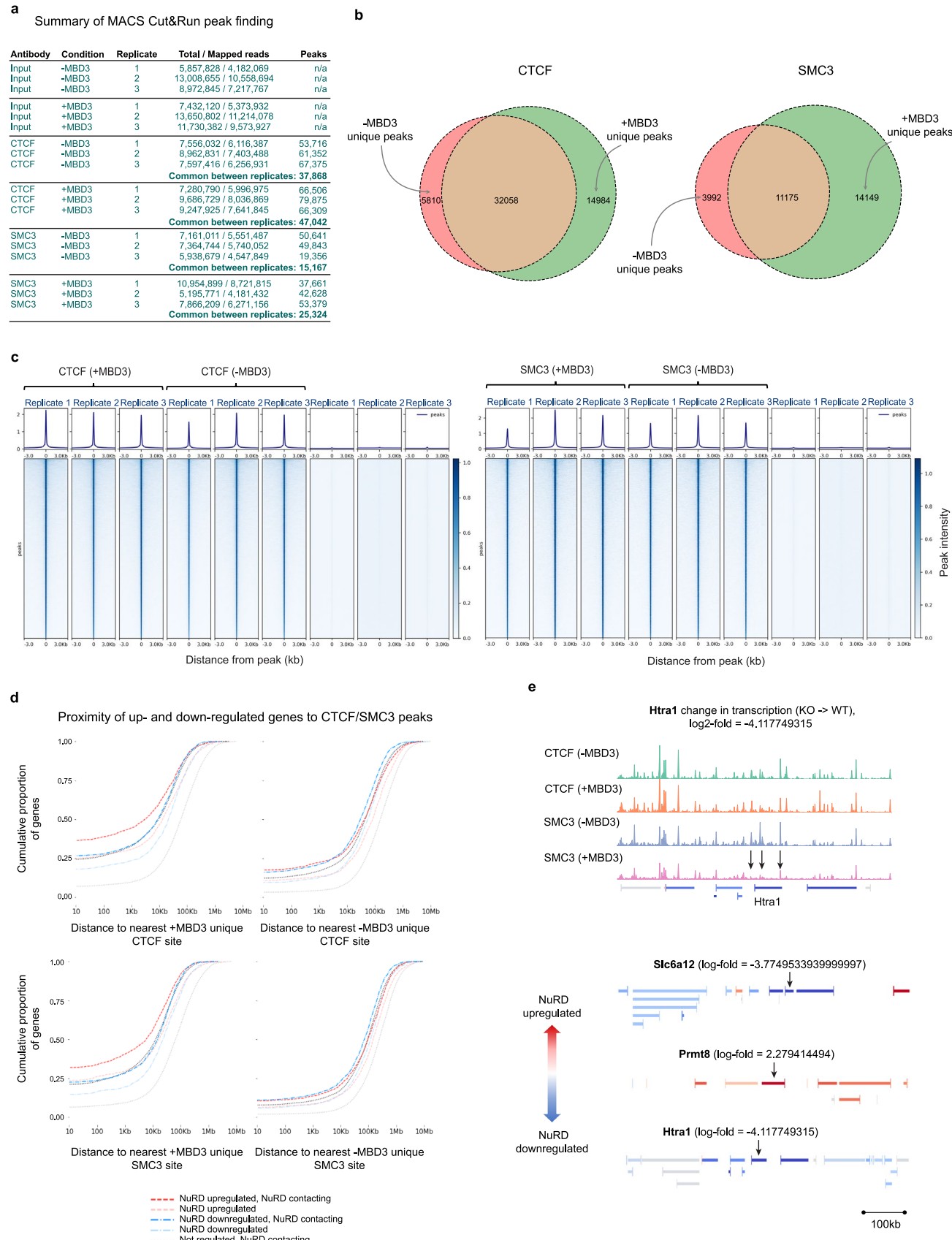

**a** Summary of MACS Cut&Run peak finding

| Antibody | Condition | Replicate | Total / Mapped reads | Peaks |
|---|---|---|---|---|
| Input | -MBD3 | 1 | 5,857,828 / 4,182,069 | n/a |
| Input | -MBD3 | 2 | 13,008,655 / 10,558,694 | n/a |
| Input | -MBD3 | 3 | 8,972,845 / 7,217,767 | n/a |
| Input | +MBD3 | 1 | 7,432,120 / 5,373,932 | n/a |
| Input | +MBD3 | 2 | 13,650,802 / 11,214,078 | n/a |
| Input | +MBD3 | 3 | 11,730,382 / 9,573,927 | n/a |
| CTCF | -MBD3 | 1 | 7,556,032 / 6,116,387 | 53,716 |
| CTCF | -MBD3 | 2 | 8,962,831 / 7,403,488 | 61,352 |
| CTCF | -MBD3 | 3 | 7,597,416 / 6,256,931 | 67,375 |
| | | | **Common between replicates: 37,868** | |
| CTCF | +MBD3 | 1 | 7,280,790 / 5,996,975 | 66,506 |
| CTCF | +MBD3 | 2 | 9,686,729 / 8,036,869 | 79,875 |
| CTCF | +MBD3 | 3 | 9,247,925 / 7,641,845 | 66,309 |
| | | | **Common between replicates: 47,042** | |
| SMC3 | -MBD3 | 1 | 7,161,011 / 5,551,487 | 50,641 |
| SMC3 | -MBD3 | 2 | 7,364,744 / 5,740,052 | 49,843 |
| SMC3 | -MBD3 | 3 | 5,938,679 / 4,547,849 | 19,356 |
| | | | **Common between replicates: 15,167** | |
| SMC3 | +MBD3 | 1 | 10,954,899 / 8,721,815 | 37,661 |
| SMC3 | +MBD3 | 2 | 5,195,771 / 4,181,432 | 42,628 |
| SMC3 | +MBD3 | 3 | 7,866,209 / 6,271,156 | 53,379 |
| | | | **Common between replicates: 25,324** | |

**Extended Data Fig. 9 | See next page for caption.**

**Extended Data Fig. 9 | CUT&RUN experiments reveal that assembly of the intact NuRD complex leads to a redistribution of both CTCF and SMC3 (a Cohesin subunit) near NuRD-regulated genes. (a)** Sequencing statistics and identification of peaks that are shared in the CUT&RUN experiment replicates. **(b)** Venn diagrams showing the overlap in CTCF and SMC3 peaks in the absence and presence of MBD3. **(c)** (Top) Average peak profile and (Bottom) heatmap of CTCF and SMC3 signals +/-3 kb either side of identified peaks shows no significant changes in the overall levels of CTCF and Cohesin (SMC3) in the absence and presence of intact NuRD. **(d)** Cumulative probability plots of the distance from different categories of promoter to the nearest CTCF or Cohesin (SMC3) binding site found uniquely in either the presence (left) or absence (right) of MBD3. These plots are compared to genes with no transcriptional change (dotted orange lines) and they all have $p$-values of $< 1 \times 10^{-30}$ (Mann-Whitney U test). **(e)** (Top) Comparison of the CTCF and Cohesin (SMC3) Cut&Run data around the *Htra1* gene. Positions where Cohesin is lost in the presence of intact NuRD within the body of the *Htra1* gene and upstream of its promoter are indicated with black arrows. (Bottom) Genome browser views showing three representative examples of regions containing genes – *Slc6a12*, *Prmt8* and *Htra1* – that are highly regulated by NuRD. The genes are coloured according to their log-fold-change in levels of expression (red = upregulated, blue = downregulated, intensity = absolute log fold change).

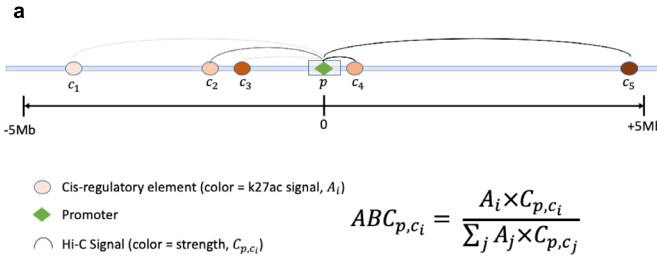

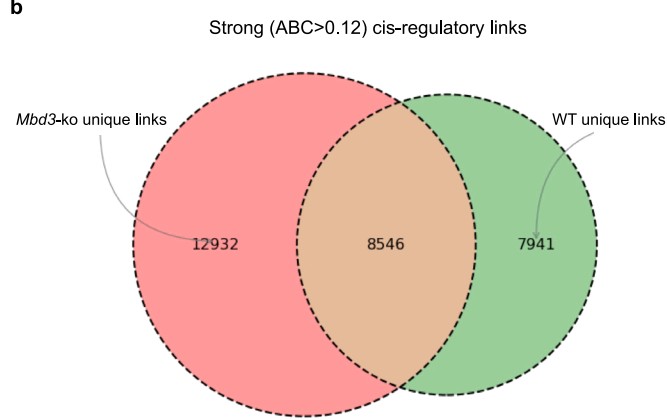

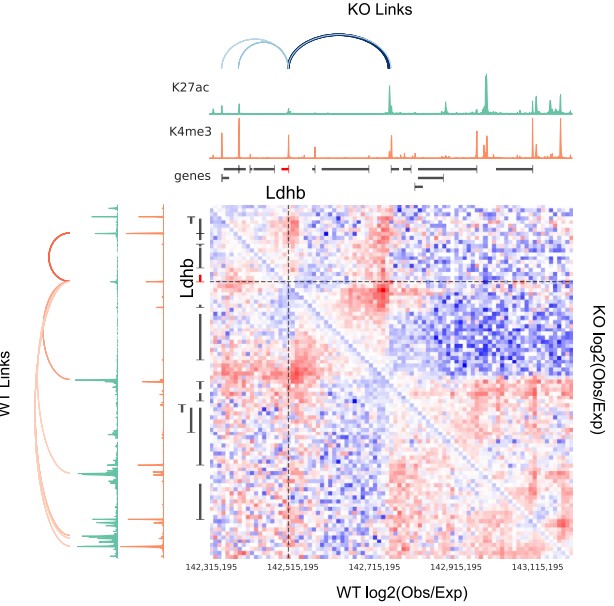

**Extended Data Fig. 10 | Activity-by-contact model analysis of enhancer-promoter interaction strength in *Mbd3* knockout and wild-type ES cells.** **(a)** Schematic of the activity-by-contact (ABC) analysis illustrating how the enhancer-promoter (E-P) interaction strength (ABC score) is calculated from the H3K4me3 signal at promoters and H3K27ac signal at nearby enhancers (defined using ChIP-seq data from wild-type and *Mbd3*-depleted ES cells)[18] as well as the Hi-C signal strength. **(b)** Venn diagram showing the overlap between strong enhancer-promoter contacts identified in *Mbd3* knockout and wild-type ES cells.

**(c)** Contact maps for a region around the *Ldhb* gene with changes in enhancer-promoter interactions and loops/TADs indicated. The maps for the *Mbd3-ko* and wild-type cells are shown above/below the diagonal, respectively, and they are coloured according to their log-fold-change in contact frequency relative to that expected theoretically at that particular distance (red = increased, blue = decreased, intensity = absolute log fold change). The black dotted lines mark the position of the *Ldhb* promoter.

# nature research

|---|---|

# Reporting Summary

Nature Research wishes to improve the reproducibility of the work that we publish. This form provides structure for consistency and transparency in reporting. For further information on Nature Research policies, see Authors & Referees and the Editorial Policy Checklist.

## Statistics

For all statistical analyses, confirm that the following items are present in the figure legend, table legend, main text, or Methods section.

| n/a | Confirmed | |
|---|---|---|
| ☐ | ☒ | The exact sample size (*n*) for each experimental group/condition, given as a discrete number and unit of measurement |
| ☐ | ☒ | A statement on whether measurements were taken from distinct samples or whether the same sample was measured repeatedly |
| ☐ | ☒ | The statistical test(s) used AND whether they are one- or two-sided<br>*Only common tests should be described solely by name; describe more complex techniques in the Methods section.* |
| ☒ | ☐ | A description of all covariates tested |
| ☐ | ☒ | A description of any assumptions or corrections, such as tests of normality and adjustment for multiple comparisons |
| ☐ | ☒ | A full description of the statistical parameters including central tendency (e.g. means) or other basic estimates (e.g. regression coefficient) AND variation (e.g. standard deviation) or associated estimates of uncertainty (e.g. confidence intervals) |
| ☐ | ☒ | For null hypothesis testing, the test statistic (e.g. *F*, *t*, *r*) with confidence intervals, effect sizes, degrees of freedom and *P* value noted<br>*Give P values as exact values whenever suitable.* |
| ☒ | ☐ | For Bayesian analysis, information on the choice of priors and Markov chain Monte Carlo settings |
| ☒ | ☐ | For hierarchical and complex designs, identification of the appropriate level for tests and full reporting of outcomes |
| ☒ | ☐ | Estimates of effect sizes (e.g. Cohen's *d*, Pearson's *r*), indicating how they were calculated |

*Our web collection on statistics for biologists contains articles on many of the points above.*

## Software and code

Policy information about availability of computer code

| Data collection | Imaging data collection: Micro-manager (https://www.micro-manager.org) and ImageJ (https://imagej.nih.gov/ij/) |
|---|---|
| Data analysis | See Methods for references:<br><br>Hi-C analysis: NucProcess (https://github.com/tjs23/nuc_processing); NucTools (https://github.com/tjs23/nuc_tools); Juicer (https://github.com/aidenlab/juicer); Cooler (https://cooler.readthedocs.io/en/latest/); CscoreTool (https://github.com/scoutzxb/CscoreTool); lavaburst (https://github.com/nvictus/lavaburst); enhancer-promoter analysis (https://github.com/dhall1995/Acitivity-By-Contact_Enhancer-Promoter_Link_Prediction);<br><br>Cut&Run analysis: Trim galore (https://www.bioinformatics.babraham.ac.uk/projects/trim_galore/); Bowtie2 (REF. 74); Deeptools v2.5.0 (https://github.com/deeptools/deepTools).<br><br>Single molecule/foci image analysis: [PeakFit (REF. 99), easy-DHPSF software (REF. 79)]; Trajectory analysis (https://github.com/wb104/trajectory-analysis); Gaussian mixture model classification (https://zenodo.org/record/6497411#.YmlGFy8w3q0);<br><br>3D DNA FISH analysis: Imaris 9.6. |

For manuscripts utilizing custom algorithms or software that are central to the research but not yet described in published literature, software must be made available to editors/reviewers. We strongly encourage code deposition in a community repository (e.g. GitHub). See the Nature Research guidelines for submitting code & software for further information.

## Data

Policy information about availability of data

All manuscripts must include a data availability statement. This statement should provide the following information, where applicable:

- Accession codes, unique identifiers, or web links for publicly available datasets
- A list of figures that have associated raw data
- A description of any restrictions on data availability

The single-molecule/locus imaging movies and XYZt single molecule/locus trajectory data files are available at: https://zenodo.org/deposit/7985268. (DOI: 10.5281/zenodo.7985268)

Figures 1, 2, 3 and 4, and Extended Data Figures 2, 3, 4, 5 and 7 - XYZt data

Cut&Run and Hi-C datasets are available from the Gene Expression Omnibus (GEO) repository under accession code GSE179007. These data were processed using the GRCm38.p6 mouse reference genome (https://www.ncbi.nlm.nih.gov/assembly/GCF_000001635.26/).

Extended Data Figure 9 - Cut&Run data

Figure 5 and Extended Data Figures 8 and 10 - Hi-C data

# Field-specific reporting

Please select the one below that is the best fit for your research. If you are not sure, read the appropriate sections before making your selection.

☒ Life sciences  ☐ Behavioural & social sciences  ☐ Ecological, evolutionary & environmental sciences

For a reference copy of the document with all sections, see nature.com/documents/nr-reporting-summary-flat.pdf

# Life sciences study design

All studies must disclose on these points even when the disclosure is negative.

| | |
|---|---|
| Sample size | In Figure 2c and Extended Data Figure 3c, the apparent diffusion coefficient of freely diffusing CHD4 and MTA2 molecules increase upon removal of MBD3. We ensured a minimum of 80 trajectories to estimate the diffusion coefficient for MTA2 molecules since our data indicate most MTA2 molecules are associated with the remodeller (shift in diffusion coefficient histogram observed as opposed to change in variance upon removal of MBD3). In contrast, because CHD4 exists on its own as well as in the NuRD (~36 %) and ChAHP (~7 %) complexes, we recorded > 900 trajectories per CHD4 sample to ensure detection of changes in diffusion coefficient for CHD4 in complexes that may be as little as ~36 % of the total CHD4 molecules. In Figure 2c and Extended Data Figure 3c, the percentage of freely diffusing CHD4 and MTA2 molecules (as opposed to chromatin bound) increased in the absence of MBD3 by 5 % and 36 % respectively. We ensured these percentages were represented by at least 100 trajectories. For example, to determine the small 5 % increase for CHD4, > 2000 trajectories per sample is sufficient (5 % is >100 trajectories). To determine the 36 % increase of MTA2, >300 trajectories is sufficient (36 % is >100 trajectories). In Figure 2d, our data indicated a single exponential fit and so we ensured > 60 association times were sufficient to accurately fit association times. In Figure 3 and Extended Data Figure 5, we analyse changes in the movement of chromatin bound CHD4. Since the F2 state accounted for as low as 7 % of the total trajectories and since as little as 36 % of CHD4 could be in NuRD as opposed to other complexes, we ensured >4500 trajectories to ensure more than 100 trajectories of the bound NuRD complex were used to estimate parameters in the F2 state. |
| Data exclusions | No data were excluded. |
| Replication | Replicates were collected for all experiments, and all were successful. Biochemical experiments are representative of ≥3 independent replicates. Overlap of Hi-C replicates and comparison with published replicates is shown in Extended Data Figure 8a. Overlap of CTCF/SMC3 Cut&Run replicates is shown in Extended Data Figures 9a and 9b. Replicates were collected for all single-molecule tracking datasets and are as described in "Live-cell 3D single-molecule imaging" in the Methods, or in "Gaussian fitting of 500 ms exposure anomalous exponents" in the Supplementary Methods. "Data was collated from 3 fields of view with either around 3 or 6 cells in each field of view imaged, leading to a total of around 9 or 18 cells per condition." In addition, "to assess the reproducibility of these results an additional 3 fields of view containing around 18 cells were collected for chromatin bound CHD4 on a different day and shown to have a similar anomalous exponent distribution". |
| Randomization | N/A - sample allocation was not randomized because comparative experiments were designed, carried out and analysed by the same person. |
| Blinding | N/A - see above. |

# Reporting for specific materials, systems and methods

We require information from authors about some types of materials, experimental systems and methods used in many studies. Here, indicate whether each material, system or method listed is relevant to your study. If you are not sure if a list item applies to your research, read the appropriate section before selecting a response.

## Materials & experimental systems

| n/a | Involved in the study |
|---|---|
| ☐ | ☒ Antibodies |
| ☐ | ☒ Eukaryotic cell lines |
| ☒ | ☐ Palaeontology |
| ☒ | ☐ Animals and other organisms |
| ☒ | ☐ Human research participants |
| ☒ | ☐ Clinical data |

## Methods

| n/a | Involved in the study |
|---|---|
| ☐ | ☒ ChIP-seq |
| ☒ | ☐ Flow cytometry |
| ☒ | ☐ MRI-based neuroimaging |

# Antibodies

| | |
|---|---|
| Antibodies used | Antibodies used were:<br><br>1. Rabbit anti-CTCF, Millipore, 07-729, Polyclonal<br>2. Rabbit anti-SMC3, Abcam, ab9263, Polyclonal<br>3. Mouse anti-CHD4, Abcam, ab70469, Monoclonal [3F2/4]<br>4. Mouse anti-FLAG, Sigma, F1804, Monoclonal [M2]<br>5. Rabbit anti-GATAD2A, Abcam, ab87663, Polyclonal<br>6. Rabbit anti-HDAC1, Abcam, ab7028, Polyclonal<br>7. Rabbit anti-MBD3, Abcam, ab157464, Monoclonal [EPR9913]<br>8. Mouse anti-MTA2, Abcam, ab50209, Monoclonal [MTA2-276]<br>9. Mouse anti-PCNA, Santa Cruz, Sc56, Monoclonal [PC10]<br>10. Recombinant GFP-Booster Alexa Fluor® 488 nanobody, ChromoTek, gb2AF488 |
| Validation | Primary antibodies validated in multiple previous studies. Knockout validation examples are provided below:<br><br>1. Rabbit anti-CTCF - (Ren et al, Mol Cell. 2017 Sep 21;67(6):1049-1058.e6. doi: 10.1016/j.molcel.2017.08.026)<br>2. Rabbit anti-SMC3 - (Wang et al, Exp Hematol 2019 70:70-84.e6)<br>3. Mouse anti-CHD4 - (O'Shaughnessy-Kirwan et al, Development. 2015 Aug 1; 142(15): 2586–2597)<br>4. Mouse anti-FLAG - validated in manuscript as negative using untagged cell lines<br>5. Rabbit anti-GATAD2A - validated using knock-down cells and purified GATAD2A.<br>6. Rabbit anti-HDAC1 - (Gonneaud et al, Sci Rep. 2019 Mar 29;9(1):5363. doi: 10.1038/s41598-019-41842-6)<br>7. Rabbit anti-MBD3 - validated in manuscript as negative in Mbd3-knockout cell lines and in (Bornelov et al, Mol Cell. 2018 Jul 5; 71(1): 56–72.e4)<br>8. Mouse anti-MTA2 - (Burgold et al, EMBO J. 2019 Jun 17; 38(12): e100788)<br>9. Mouse anti-PCNA - (Dietsch et al, Biotechniques 2017 Feb 1;62(2):80-82. doi: 10.2144/000114518) |

# Eukaryotic cell lines

Policy information about cell lines

| | |
|---|---|
| Cell line source(s) | Sf21 insect cells were used for the expression of recombinant proteins.<br><br>The background mouse E14tg2a ES cells are available from Sigma Aldrich (08021401). Cell lines generated in this study are described in "Mouse embryonic stem cell line generation" and are being deposited at Addgene. |
| Authentication | The background mouse E14tg2a ES cell lines and those generated in this study were characterized by qPCR, RNA-seq, ChIP-seq, and potency assays. In addition, Western blots and immunoprecipitation studies of the NuRD complex are shown in Extended Data Figure 1. |
| Mycoplasma contamination | Mouse ES cell lines used in this study were routinely screened for mycoplasma contamination and tested negative. |
| Commonly misidentified lines (See ICLAC register) | None were used in this study. |

# ChIP-seq

## Data deposition

☒ Confirm that both raw and final processed data have been deposited in a public database such as GEO.

☒ Confirm that you have deposited or provided access to graph files (e.g. BED files) for the called peaks.

| | |
|---|---|
| Data access links<br>*May remain private before publication.* | https://www.ncbi.nlm.nih.gov/geo/query/acc.cgi?acc=GSE179007 |
| Files in database submission | Processed data files:<br>SLX-20518.A1.HCCJGDRXY.Q30.srt.nodup.noChrM.bam_peaks.narrowPeak.txt.gz |

SLX-20518.A4.HCCJGDRXY.Q30.srt.nodup.noChrM.bam_peaks.narrowPeak.txt.gz
SLX-20518.A6.HCCJGDRXY.Q30.srt.nodup.noChrM.bam_peaks.narrowPeak.txt.gz
SLX-20518.B2.HCCJGDRXY.Q30.srt.nodup.noChrM.bam_peaks.narrowPeak.txt.gz
SLX-20518.B4.HCCJGDRXY.Q30.srt.nodup.noChrM.bam_peaks.narrowPeak.txt.gz
SLX-20518.B7.HCCJGDRXY.Q30.srt.nodup.noChrM.bam_peaks.narrowPeak.txt.gz
SLX-20518.E1.HCCJGDRXY.Q30.srt.nodup.noChrM.bam_peaks.narrowPeak.txt.gz
SLX-20518.E3.HCCJGDRXY.Q30.srt.nodup.noChrM.bam_peaks.narrowPeak.txt.gz
SLX-20518.E6.HCCJGDRXY.Q30.srt.nodup.noChrM.bam_peaks.narrowPeak.txt.gz
SLX-20518.F1.HCCJGDRXY.Q30.srt.nodup.noChrM.bam_peaks.narrowPeak.txt.gz
SLX-20518.F4.HCCJGDRXY.Q30.srt.nodup.noChrM.bam_peaks.narrowPeak.txt.gz
SLX-20518.F6.HCCJGDRXY.Q30.srt.nodup.noChrM.bam_peaks.narrowPeak.txt.gz
SLX-20518.A1.HCCJGDRXY.Q30.srt.nodup.noChrM.bam_treat_pileup_filter_norm.bw.txt.gz
SLX-20518.A4.HCCJGDRXY.Q30.srt.nodup.noChrM.bam_treat_pileup_filter_norm.bw.txt.gz
SLX-20518.A6.HCCJGDRXY.Q30.srt.nodup.noChrM.bam_treat_pileup_filter_norm.bw.txt.gz
SLX-20518.B2.HCCJGDRXY.Q30.srt.nodup.noChrM.bam_treat_pileup_filter_norm.bw.txt.gz
SLX-20518.B4.HCCJGDRXY.Q30.srt.nodup.noChrM.bam_treat_pileup_filter_norm.bw.txt.gz
SLX-20518.B7.HCCJGDRXY.Q30.srt.nodup.noChrM.bam_treat_pileup_filter_norm.bw.txt.gz
SLX-20518.E1.HCCJGDRXY.Q30.srt.nodup.noChrM.bam_treat_pileup_filter_norm.bw.txt.gz
SLX-20518.E3.HCCJGDRXY.Q30.srt.nodup.noChrM.bam_treat_pileup_filter_norm.bw.txt.gz
SLX-20518.E6.HCCJGDRXY.Q30.srt.nodup.noChrM.bam_treat_pileup_filter_norm.bw.txt.gz
SLX-20518.F1.HCCJGDRXY.Q30.srt.nodup.noChrM.bam_treat_pileup_filter_norm.bw.txt.gz
SLX-20518.F4.HCCJGDRXY.Q30.srt.nodup.noChrM.bam_treat_pileup_filter_norm.bw.txt.gz
SLX-20518.F6.HCCJGDRXY.Q30.srt.nodup.noChrM.bam_treat_pileup_filter_norm.bw.txt.gz

Raw data files:
SLX-20518.A1.HCCJGDRXY.s_2.r_1.fq.gz
SLX-20518.A1.HCCJGDRXY.s_2.r_2.fq.gz
SLX-20518.A4.HCCJGDRXY.s_2.r_1.fq.gz
SLX-20518.A4.HCCJGDRXY.s_2.r_2.fq.gz
SLX-20518.A6.HCCJGDRXY.s_2.r_1.fq.gz
SLX-20518.A6.HCCJGDRXY.s_2.r_2.fq.gz
SLX-20518.B2.HCCJGDRXY.s_2.r_1.fq.gz
SLX-20518.B2.HCCJGDRXY.s_2.r_2.fq.gz
SLX-20518.B4.HCCJGDRXY.s_2.r_1.fq.gz
SLX-20518.B4.HCCJGDRXY.s_2.r_2.fq.gz
SLX-20518.B7.HCCJGDRXY.s_2.r_1.fq.gz
SLX-20518.B7.HCCJGDRXY.s_2.r_2.fq.gz
SLX-20518.D3.HCCJGDRXY.s_2.r_1.fq.gz
SLX-20518.D3.HCCJGDRXY.s_2.r_2.fq.gz
SLX-20518.D5.HCCJGDRXY.s_2.r_1.fq.gz
SLX-20518.D5.HCCJGDRXY.s_2.r_2.fq.gz
SLX-20518.D8.HCCJGDRXY.s_2.r_1.fq.gz
SLX-20518.D8.HCCJGDRXY.s_2.r_2.fq.gz
SLX-20518.E1.HCCJGDRXY.s_2.r_1.fq.gz
SLX-20518.E1.HCCJGDRXY.s_2.r_2.fq.gz
SLX-20518.E3.HCCJGDRXY.s_2.r_1.fq.gz
SLX-20518.E3.HCCJGDRXY.s_2.r_2.fq.gz
SLX-20518.E6.HCCJGDRXY.s_2.r_1.fq.gz
SLX-20518.E6.HCCJGDRXY.s_2.r_2.fq.gz
SLX-20518.F1.HCCJGDRXY.s_2.r_1.fq.gz
SLX-20518.F1.HCCJGDRXY.s_2.r_2.fq.gz
SLX-20518.F4.HCCJGDRXY.s_2.r_1.fq.gz
SLX-20518.F4.HCCJGDRXY.s_2.r_2.fq.gz
SLX-20518.F6.HCCJGDRXY.s_2.r_1.fq.gz
SLX-20518.F6.HCCJGDRXY.s_2.r_2.fq.gz
SLX-20518.H2.HCCJGDRXY.s_2.r_1.fq.gz
SLX-20518.H2.HCCJGDRXY.s_2.r_2.fq.gz
SLX-20518.H5.HCCJGDRXY.s_2.r_1.fq.gz
SLX-20518.H5.HCCJGDRXY.s_2.r_2.fq.gz
SLX-20518.H7.HCCJGDRXY.s_2.r_1.fq.gz
SLX-20518.H7.HCCJGDRXY.s_2.r_2.fq.gz

**Genome browser session**
(e.g. UCSC)

No longer applicable.

## Methodology

**Replicates**

Three biological replicates were obtained per Cut&Run sample. They showed good agreement in peaks called as indicated in Extended Data Figures 9a and 9b.

**Sequencing depth**

See "Cut&Run" in the Methods and Extended Data Figure 9a. 50 bp paired-end sequencing was carried out:

Samples: Total number of uniquely mapped reads (8-16 million reads/replicate)
Inputs: Total number of uniquely mapped reads (8-23 million reads/replicate)

| | |
|---|---|
| Antibodies | Antibodies were:<br>1. Rabbit anti-CTCF, Millipore, 07-729, Polyclonal<br>2. Rabbit anti-SMC3, Abcam, ab9263, Polyclonal |
| Peak calling parameters | Read mapping:<br>All Cut&Run data was trimmed using trim_galore (https://www.bioinformatics.babraham.ac.uk/projects/trim_galore/) and then aligned (with standard parameters) using Bowtie2 (REF. 74) to the Mus Musculus reference genome GRCm38.p6 (https://www.ncbi.nlm.nih.gov/assembly/GCF_000001635.26/)<br><br>Peak calling:<br>Peaks were called using MACS2 (REF. 76) to be at FDR 1 % and above 5-fold enrichment. |
| Data quality | Peaks at FDR 1% and above 5-fold enrichment:<br><br>CTCF 0h 1 = 53716 peaks<br>CTCF 0h 2 = 61352 peaks<br>CTCF 0h 3 = 67375 peaks<br> Shared between replicates: 37868 peaks<br><br>CTCF 48h 1 = 66506 peaks<br>CTCF 48h 2 = 79875 peaks<br>CTCF 48h 3 = 66309 peaks<br> Shared between replicates: 47042 peaks<br><br>Smc3 0h 1 = 50641 peaks<br>Smc3 0h 2 = 49843 peaks<br>Smc3 0h 3 = 19356 peaks<br> Shared between replicates: 15167 peaks<br><br>Smc3 48h 1 = 37661 peaks<br>Smc3 48h 2 = 42628 peaks<br>Smc3 48h 3 = 53379 peaks<br> Shared between replicates: 25324 peaks<br><br>Data quality ensured by comparison of peaks called between replicates but also by comparing peaks to published datasets. |
| Software | See "Cut&Run" in the Methods section. Software used:<br><br>1) Trim galore (https://www.bioinformatics.babraham.ac.uk/projects/trim_galore/)<br>2) Bowtie2 (REF. 74)<br>3) Deeptools v2.5.0 (REF. 75)<br>4) MACS2 (REF. 76) |

