## [Peer Review File · Nature Structural & Molecular Biology]

Peer Review Information

Manuscript Title: Live-cell 3D single-molecule tracking reveals modulation of enhancer dynamics by NuRD

Corresponding author name(s): David Holcman, Brian Hendrich, David Klenerman, Ernest Laue

Editorial Notes:

Transferred manuscripts This manuscript has been previously reviewed at another journal that is not operating a transparent peer review scheme. This document only contains reviewer comments, rebuttal and decision letters for versions considered at Nature Structural & Molecular Biology.

Reviewer Comments & Decisions:

Decision Letter, initial version:
--

Message: Dear Dr. Laue,

Thank you again for submitting your manuscript "Live-cell 3D single-molecule tracking reveals how NuRD modulates enhancer dynamics". I apologize for the very long and unusual delay in responding, which resulted from the difficulty in obtaining suitable referee reports. While we received a report from the original [Redacted] Reviewer #1, Reviewers #2 and #3 were unfortunately not able to provide us with reports. Therefore we recruited new referee #4 with overlapping expertise.

We now have comments (below) from the 2 reviewers who evaluated your paper. In light of those reports, we remain interested in your study and would like to see your response to the comments of the referees, in the form of a revised manuscript.

You will see that while both referees find that the revised manuscript has addressed most of the previous comments, some important concerns remain. We agree with reviewer #1's suggestion to revise the manuscript without directed motion claims, plus minor revisions, would be an adequate revision avenue. Please be sure to address/respond to all concerns of the referees in full in a point-by-point response and highlight all changes in the revised manuscript text file. If you have comments that are intended for editors only, please include those in a separate cover letter.

We expect to see your revised manuscript within 6 weeks. If you cannot send it within this time, please contact us to discuss an extension; we would still consider your revision, provided that no similar work has been accepted for publication at NSMB or published elsewhere.

Reporting Summary:

Please note that all key data shown in the main figures as cropped gels or blots should be presented in uncropped form, with molecular weight markers. These data can be aggregated into a single supplementary figure item. While these data can be displayed in a relatively informal style, they must refer back to the relevant figures. These data should be submitted with the final revision, as source data, prior to acceptance, but you may want to start putting it together at this point.

SOURCE DATA: we urge authors to provide, in tabular form, the data underlying the graphical representations used in figures. This is to further increase transparency in data reporting, as detailed in this editorial

(<http://www.nature.com/nsmb/journal/v22/n10/full/nsmb.3110.html>). Spreadsheets can be submitted in excel format. Only one (1) file per figure is permitted; thus, for multi-paneled figures, the source data for each panel should be clearly labeled in the Excel file; alternately the data can be provided as multiple, clearly labeled sheets in an Excel file. When submitting files, the title field should indicate which figure the source data pertains to. We encourage our authors to provide source data at the revision stage, so that they are part of the peer-review process.

Data availability: this journal strongly supports public availability of data. All data used in accepted papers should be available via a public data repository, or alternatively, as Supplementary Information. If data can only be shared on request, please explain why in your Data Availability Statement, and also in the correspondence with your editor. Please note that for some data types, deposition in a public repository is mandatory - more information on our data deposition policies and available repositories can be found below: <https://www.nature.com/nature-research/editorial-policies/reporting-standards#availability-of-data>

Nature Structural & Molecular Biology is committed to improving transparency in authorship. As part of our efforts in this direction, we are now requesting that all authors identified as 'corresponding author' on published papers create and link their Open Researcher and Contributor Identifier (ORCID) with their account on the Manuscript Tracking System (MTS), prior to acceptance. This applies to primary research papers only. ORCID helps the scientific community achieve unambiguous attribution of all scholarly contributions. You can create and link your ORCID from the home page of the MTS by clicking on 'Modify my Springer Nature account'. For more information please visit please visit www.springernature.com/orcid.

[redacted]

We look forward to seeing the revised manuscript and thank you for the opportunity to

review your work.

Sincerely,
Carolina

Carolina Perdigoto, PhD
Chief Editor
Nature Structural & Molecular Biology
orcid.org/0000-0002-5783-7106

Reviewers' Comments:

Reviewer #1:

Remarks to the Author:

I reviewed an earlier version of the manuscript for Nature and am now reviewing a revision. The authors have undertaken significant revisions. The methods are cutting-edge. The questions are very interesting. The biological importance of the topic is high. The revised manuscript is substantially strengthened. However, reviewing the revised manuscript was frustrating. The rebuttal is extremely long (20,000 words), and for many of my concerns, the authors have responded to a related point, but not my specific concern. So now I am in the position of either 1) giving up or 2) writing a response of similar length to explain for each point, explaining what my original concern was, why the very long response did not accurately address my concern, and start another cycle.

I still believe several of the claims are unsupported and that many of my concerns have not been addressed, and I do not feel like I can support publication in the present form.

I would suggest that the authors either fully demonstrate directed motion and that a version of the manuscript without directed motion is approved after the revisions below.

DIRECTED MOTION

It's really challenging to be a reviewer when the authors make clearly incorrect claims like "Chromatin inside the cell moves significantly faster than the cell itself – this has been demonstrated in a range of papers including those referred to by the reviewer (Gu et al., Science, 2018; Germier PMID 28978433; Nagashima PMID 30824489)". Cell movement tends to be faster than chromatin movement. However, the more salient point is that cell movement tends to be directional, making any protein or DNA locus inside the cell also look at least partially directional in its movement.

In fact, the classic newbie mistake in the live-cell nuclear imaging field is to mistake directional cell movement for directed motion of chromatin bound molecules.

So let me outline what I would need to be persuaded of the directed motion claim:

1. Clearly define your criteria for directed motion. The authors are vague about this, but seem to suggest that it is not just $\alpha > 1$. Is it the drift vector?
2. Show us positive and negative controls. Many other studies have shown many other

chromatin loci that are not directed. Thus, repeat the dCas9-locus tracking for a handful of other loci and run the same analysis and show that these show no directed motion.

3. For positive controls, find some examples of directed motion and demonstrate this.

4. Same for protein tracking. HaloTag-NLS is a nice control for some things, but we need a largely DNA bound protein. Do SMT of both a positive and negative directed motion control protein, and run the same analysis, and show that only NuRD and NuRD-bound enhancers show directed motion. Perhaps a histone protein would work as a negative control here.

5. Do 2-color imaging. In the first color, use an orthogonal dCas9-RFP (or similar) bound to telomeres. This will give you many spots to calculate cell movement. Then simultaneously do dCas9-GFP imaging with and without correction for cell movement (e.g. use center of mass of the telomeres to estimate cell movement), run the same analysis and show that the directed motion claim is not an artifact of cell movement.

6. Show us at least 5 independent large-field-of-view movies ($\sim 100\mu\text{m} \times 100\mu\text{m}$) of confluent mESC colonies with the nucleus labeled (e.g. Histone GFP) imaged for at least 1 hour and at least 1 z-stack per minute (ideally faster). Show us these 5 movies and show how none of one the cells noticeably move.

7. In addition to the speed of mESC movement, the point is that at the minute time-scale, cell movement tends to be directional such that even if cell movement is comparable to single-molecule movement, when summed, it will make single-molecule movement look directional.

8. Systematically benchmark the accuracy of the algorithm for inferring alpha on simulations similar to the experimental data with worse and better localization noise. Similarly for the magnitude of the drift vector, V .

9. Find a good protein and DNA negative control for directed movement, and show us that the magnitude of the drift vector is 0, such that cell movement does not artefactually make it look like directed motion.

10. Ideally, find a mutant that disrupts the directed motion (e.g. if you could KO a factor, which causes NuRD/enhancers to no longer be directed).

11. Based on the telomere-dCas9 image, they will get "cell movement trajectories". Now perform simulations of undirected motion (e.g. Brownian and confined) and add the cell movement trajectories to the simulated data as well as realistic noise. Then apply the same analysis algorithm and show us that directed motion is not accidentally inferred.

That KLF4 also shows directed movement in their analysis does not strengthen the claim, since if it is a cell movement artifact, essentially all chromatin-bound proteins would similarly show this, as would also loci.

Frankly, if the authors could demonstrate directed enhancer movement this would in my opinion be a landmark paper on its own, but as mentioned in my first review, extraordinary claims require extraordinary evidence, and many controls are missing here in my opinion.

SPECIFIC COMMENTS:

Hi-C concern: the authors did not really address my Hi-C analysis concern. Something got lost in communication and they have addressed a different concern. To be clear, I was not worried about experimental variability between replicates, though this is always good to estimate. I was concerned about Hi-C analysis with few contacts as I wrote ($>500\text{ M}$ needed). If they have indeed gotten sufficiently resolution now, I'd like to see a supplementary figure demonstrating this. To avoid any further misunderstanding, I'll try

to be very specific. I'd like to see a new supplementary figure showing 4 separate 1Mb regions at 5kb resolution where one triangle is WT-data from this paper and the other triangle is data from Pekowska 2018 (since the author say that only Pekowska used same cell culture conditions). I'd like to see 4 separate 1Mb regions where you can clearly see loops/dots/cornerpeaks in the Pekowska data, and see that directly compared to the present paper.

To support the claim that the authors study enhancer-promoter interactions in Hi-C, I would also like to see some example plots side-by-side with Pekowska, where there is a clear loop/dot between an enhancer and a promoter. And the same for the Mbd3-KO Hi-C data side by side.

LOCALIZATION UNCERTAINTY:

The authors state that "As we discuss in the response to Q2 above, computing the MSD and then taking the average over many realizations shows that the anomalous exponent alpha is not affected by the localization noise – thus we do not need to worry about the impact of the localization error on alpha." I am sorry but do not believe this is correct. Try simulating Brownian motion ($\alpha=1$) at 100 Hz with $\sigma = [0,10,20,40,60,80,100 \text{ nm}]$ and a $D=1\mu\text{m}^2/\text{s}$ and then chop up the data in short trajectories of 5 frames and fit an MSD, both at the single trajectory data and time-and-ensemble averaged MSD. If the authors claim is true, a fit of $\text{MSD}=4Dt^\alpha$ should give the same estimates of D and α as well as the same distribution of D and α estimates from single trajectories independent of noise. When I do this, I see an effect of noise on the alpha estimate. Also, the authors reference yeast data from Shukron, but what they need to do is to systematically test the accuracy of their estimate of alpha using simulations that are realistic for the present paper. They do something along these lines on page 53 for other parameters, but they need to do and show this for alpha as well. Also, it would be best to try localization noise both at 0, below, similar to, and well-above the experimental data estimates.

Also, can you report the full distribution of localization errors/uncertainties instead of just the median. If the noise is heavy-tailed, the median can be misleading.

Basically, I have concerns about the accuracy of the alpha estimation and the lack of validation of their method.

CODE AND DATA:

The authors write: "The code was available to the reviewers upon request (as outlined in: <https://www.nature.com/nature-research/editorial-policies/reporting-standards>). However, we did not receive any such request during the review process.

For the resubmission, we have made the data and code available to the Editor and Reviewers using the following link to our group Dropbox:

https://www.dropbox.com/sh/t5dm7fqb3oxdq2b/AABwXmVhOo1QL5XyPn5Fec1_a?dl=0

The password is: 5422801006"

I suppose it would be helpful if the editor would chime in here. In my opinion, asking supposedly anonymous reviewers to email the corresponding author to request access to code during the review process is not appropriate. It would be great to hear what the editor thinks?

In my opinion, code and data should be on a repository like Figshare, Zenodo, Dryad, or Github, where it is time and date stamped. This is necessary to ensure that the code that is currently available was the same as what was used to make the figures in the paper.

When I click on the link to the Nature journal policy, I see: "This minimum dataset may be provided through deposition in public community/discipline-specific repositories, custom proprietary repositories for certain types of datasets, or general repositories like Figshare, Zenodo and Dryad." I see "public depositories", I don't see anywhere where it says that the reviewer must email the corresponding author. Could the authors point to the specific sentences in the Nature policy <https://www.nature.com/nature-research/editorial-policies/reporting-standards> where it says that asking reviewers to request the data and code is recommended?

The authors write that "We would also be happy to make the data available on request or to deposit it if a suitable site can be identified." I must confess that I find it difficult to believe that anyone towards the end of 2021 would not be aware of appropriate public repositories, especially when the data policy that the authors link to suggest 3 (Figshare, Zenodo, Dryad). Sharing trajectory data (e.g. CSV files) would likely only be a few megabytes, and can be done freely on some of repositories outlined above.

Like, what are we doing here? Arguing to the reviewer that the original submission did not violate the data sharing policy or that public repositories for data sharing are not well-known, when this is clearly not true seems like a waste of everyone's time. Just make the code and data available and we can all move on. Now I see GitHub links on page 64 which is great.

MOTION BLURRING

The authors compare their HaloTag-NLS SMT experiment to Zhen eLife 2016 (which also suffered from extensive motion-blurring), and estimate a wide range of diffusion coefficients. In their response, they do not give numbers for their own data, but say it is similar to 0.24 and 2.5 $\mu\text{m}^2/\text{s}$. I believe I have seen other studies measure $>10 \mu\text{m}^2/\text{s}$ for HaloTag alone, and e.g. Izeddin 2014 measure 13.5 $\mu\text{m}^2/\text{sec}$ for Myc. The concern is that motion-blurring causes you to overestimate the slower moving populations and overestimate the faster moving populations and incorrectly estimate the various subpopulation sizes. This issue needs to be explicitly stated in the main text, with a clear statement that they cannot exclude faster moving states and that the subpopulation sizes may be incorrect.

RESPONSE:

"Line 210-220: So the association time of 134 ms in Fig. 2d implies a similar dissociation rate to get similar bound vs. free subpopulations, since $\text{percent-bound} \sim k_{\text{ON}} / (k_{\text{OFF}} + k_{\text{ON}})$. So why is the dissociation time $>100 \text{ sec}$, i.e. 3 orders of magnitude bigger than the association time. This does not seem right?

The binding of a protein to a substrate with multiple binding sites, some specific and some non-specific, is probably very complex so we wouldn't expect it to obey the simple rules for 1:1 binding in solution. In particular, multivalency prolongs chromatin retention and accelerates the association rate, see e.g. Kilic et al. (Nat Commun., 2015). There are

many ways in which the NuRD complex can interact with chromatin – all of the CHD4, HDAC1/2, MBD2/3, MTA1/2/3 and RbAp46/48 subunits are expected to interact with chromatin in different ways – so we don't expect simple kinetics.

With regards to the percentage of bound versus free molecules, although the actual percentages may be biased because long trajectories of freely diffusing molecules may be harder to track – we can demonstrate that the changes in percentage bound CHD4 or MTA2 are meaningful. For example, the loss in MTA2 chromatin binding in the absence of MBD3 is supported by our in vitro chromatin binding experiments (Supplementary Figure 4a) and by unpublished ChIP-seq data for MTA2 in wild-type and Mbd3-ko ES cells (see Rebuttal Figure 2, below). "

The authors did not address my comment. If the k_{ON} and k_{OFF} rates are correct, then $\text{percent-bound} \sim k_{ON}/(k_{OFF}+k_{ON})$. If not, are the assumptions of a single dissociation rate used in the manuscript not violated?

The authors mention "With regards to the percentage of bound versus free molecules, although the actual percentages may be biased because long trajectories of freely diffusing molecules may be harder to track". What it looks like the authors is saying is that their data are not quantitatively interpretable and not in internal quantitative agreement. This would need to be stated in the main text, since this is currently not clear and explicit.

FRAP/FDAP

The experiment the authors did on FDAP in Rebuttal Figure 3a is very nice and strongly supports the long residence time albeit with a slight deviation (~ 100 sec vs. ~ 51 sec). To me this is strong support for a long residence time of NuRD and strengthens and supports the case of the authors. It is obviously their own decision, but I think this would make a nice supplementary figure.

DNA-FISH: the addition of the DNA-FISH experiments is a nice addition that in my opinion was necessary to support the claim of compaction.

RAW MOVIES:

Please show us 1) raw movies of the raw NuRD single-molecule tracking data; 2) with localizations and tracking overlaid and 3) show us several raw movies of the dCas9 locus tracking. These could be attached as supplementary movies so we can get a sense of the raw data quality and be referred to in the main text.

Authors should cite Ref 4 and 5 on page 17 in the context of enhancer dynamics and transcription.

Page 24, does not seem accurate to refer to cross-linking of chromatin in the last paragraph.

P.S. It would have been wonderful if the authors would have used line numbers in both the manuscript and rebuttal.

Reviewer #4:

Remarks to the Author:

The work by Basu and co-workers introduces a live-cell 3D single-molecule tracking method to identify motion of molecules in the nucleus. They focus the application of the method to track the NuRD complex (and perturbation of the same) with the goal of identifying enhancer dynamics. The imaging analysis is further complemented by NGS approaches including Hi-C and Cut&Run. Overall, the work seems well constructed and designed and the results solid. However, to this reviewer the claims may need to be softened or more data provided. That NuRD decompacts and increases enhancer-dynamics at longer range via modulating CTCF/Cohesin is a strong assumption given the data. It is also important to mention that the manuscript is very difficult to follow. Many figures are misplaced or not even referenced. Supplementary Figures are difficult to follow. No clear step-by-step conclusions provided nor a final conclusion/s in the text. Altogether, the reading of the manuscript is tedious and demerits the work. Here are some additional specifics:

1. Why the authors really see 3 states (slow, fast1 and fast2). Not clear for the data and difficult to accept without further explanation. To this reviewer there are 2 states (slow and fast) and does not understand why the slow is split into slow + fast1.
2. Many of the statistical significance are because the genome is really large and one have tons of points in the plots. But, one wonders what it really means (biologically) a "significant" fold change of 1.05?
3. I could not see clear evidence to conclude that "NuRD-bound enhancers contact promoters to down-regulates genes in the A (euchromatic) compartment".
4. I could not see clear evidence to suggest that "NuRD may alter genome structure/dynamics to affect expression of whole groups of nearby genes".
5. The provided molecular data clearly shows to this reviewer that compartmentalization is affected in Nbd3-ko cells. However, no very strong and clear data on TAD or loop alterations. Despite that, the manuscripts conclusions are mostly driven to the effect in enhancer-promoter interactions, which are likely modulated by TAD boundaries and loop extrusion.
6. No Hi-C maps are shown to the exception of a local region. I would suggest to provide the reader with visualization of data that sometimes is stronger than any smaller genome-wide differences. This can be done via JuiceBox, HiGlass or the Nucleome Browser.
7. A/B compartmentalization does not correlate with more or less genes.
8. Figure2 a/b/e never refereed. Figure 2d misplaced in the text.

Author Rebuttal to Initial comments

Reviewer #1:

I would suggest that the authors either fully demonstrate directed motion and that a version of the manuscript without directed motion is approved after the revisions below.

As we said in the previous 'Response to reviewers', we find it hard to believe that individual molecules would move such long distances in straight lines unless the motion were directed in some way. We are happy to not describe the motion's we see as '*directed*', although it does

make it hard to describe the fast F2 motion because it is characterised by high drift (i.e. movement in a defined direction).

DIRECTED MOTION

It's really challenging to be a reviewer when the authors make clearly incorrect claims like "Chromatin inside the cell moves significantly faster than the cell itself – this has been demonstrated in a range of papers including those referred to by the reviewer (Gu et al., Science, 2018; Germier PMID 28978433; Nagashima PMID 30824489)."

Cell movement tends to faster than chromatin movement. However, the more salient point is that cell movement tend to be directional, making any protein or DNA locus inside the cell also look at least partially directional in its movement.

In fact, the classic newbie mistake in the live-cell nuclear imaging field is to mistake directional cell movement for directed motion of chromatin bound molecules.

We have now removed "directed motion" from the manuscript (except for a speculative statement in the discussion) to satisfy the reviewer and, quite frankly, to ensure our authors are not subject to further derogatory comments. There is therefore no longer a need to respond to this point.

However, as we explain in the cover letter, we are confident that we are not looking at cell movement. Single molecule tracking data of chromatin bound proteins automatically provides the sorts of controls requested by the reviewer because in most images we see one or more stationary and one or more moving molecule, at the same time, inside the same cell. Clearly, we are not looking at cell movement because if we were we would see all the molecules moving in the same way. We have now added Supplementary video's showing some of our data to demonstrate this directly so that the field can make their own judgements about single molecule tracking data.

We kindly ask the reviewer for references that "*Cell movement tends to faster than chromatin movement*" at the timescale of our imaging (sliding windows of 5 s) and we will add these references to the discussion section of our manuscript alongside recent papers that support our findings of tethered (low alpha) and untethered (high alpha) chromatin motion (e.g. Shaban *et al.* Hi-D: nanoscale mapping of nuclear dynamics in single living cells. *Genome Biol* **21**, 95 (2020). <https://doi.org/10.1186/s13059-020-02002-6>).

SPECIFIC COMMENTS:

Hi-C concern: the authors did not really address my Hi-C analysis concern. Something got lost in communication and they have addressed a different concern. To be clear, I was not worried about experimental variability between replicates, though this is always good to estimate. I was concerned about Hi-C analysis with few contacts as I wrote (>500 M needed). If they have indeed gotten sufficiently resolution now, I'd like to see a Extended Data Figure demonstrating this. To avoid any further misunderstanding, I'll try to be very specific. I'd like to see a new Extended Data Figure showing 4 separate 1Mb regions at 5kb resolution where one triangle is WT-data from this paper and the other triangle is data from Pekowska 2018 (since the author say that only Pekowska used same cell culture conditions). I'd like to see 4 separate 1Mb regions where you can clearly see loops/dots/cornerpeaks in the Pekowska data, and see that directly compared to the present paper.

As we discuss in the cover letter, the Hi-C experiments were included solely to explain to the reader why we embarked on our single molecule imaging study. We showed in our previous 'Response to reviewers' (and now also provide further figures showing example loops below) that our data for WT cells was consistent with the WT data previously deposited by Pekowska et al., and we therefore combined it with our own. We then collected extra data for the Mbd3-ko so that we could generate large datasets for both the WT and Mbd3-ko cells and carry out the analyses suggested by the three reviewers. We can see no sense in wasting a lot more money to carry out further sequencing so that we can make a high-resolution comparison between our own and the Pekowska et al. WT data. We think the data we provide in the revised manuscript is quite sufficient to show that our WT data is comparable to the Pekowska et al. Hi-C experiments.

To support the claim that the authors study enhancer-promoter interactions in Hi-C, I would also like to see some example plots side-by-side with Pekowska, where there is a clear loop/dot between an enhancer and a promoter. And the same for the Mbd3-KO Hi-C data side by side.

See above – the Hi-C experiments were included solely to introduce our single molecule imaging study. We are not interested in making the claim that we can “study E-P interactions using Hi-C”. The E-P analysis (ABC contact analysis) was carried out to satisfy the requests of another reviewer, who asked us to see if E-P contact changes were consistent with transcriptional changes observed and indeed they are. Furthermore, the change in E-P proximity we observe has also been validated using the more sensitive and accurate 3D DNA FISH approach.

LOCALIZATION UNCERTAINTY:

The authors state that “As we discuss in the response to Q2 above, computing the MSD and then taking the average over many realizations shows that the anomalous exponent alpha is

not affected by the localization noise – thus we do not need to worry about the impact of the localization error on alpha.” I am sorry but do not believe this is correct. Try simulating Brownian motion (alpha=1) at 100 Hz with sigma = [0,10,20,40,60,80,100 nm] and a D=1um²/s and then chop up the data in short trajectories of 5 frames and fit an MSD, both at the single trajectory data and time-and-ensemble averaged MSD. If the authors claim is true, a fit of MSD=4Dt^{alpha} should give the same estimates of D and alpha as well as the same distribution of D and alpha estimates from single trajectories independent of noise. When I do this, I see an effect of noise on the alpha estimate. Also, the authors reference yeast data from Shukron, but what they need to do is to systematically test the accuracy of their estimate of alpha using simulations that are realistic for the present paper. They do something along these lines on page 53 for other parameters, but they need to do and show this for alpha as well.

As we explained in our previous ‘Response to reviewers’ localisation uncertainty has only a small effect on the estimation of alpha – indeed this can be seen in the simulations carried out by Shukron et al. which we showed. As we also explained, the absolute values of the parameters we measure are not really of interest – we are interested in (and can only measure) changes between apparent values determined from different samples under the same labelling and imaging conditions (i.e. datasets that have the same localisation errors and sampling rates).

Regarding the simulations of the reviewer, it is an impossible task to comment on the coding of a reviewer who has not sent us his/her code, methods or results. However, qualitatively, for the 500 ms data where we report changes in α , we compare below the simulations we carry out vs those the reviewer has carried out:

	Actual parameters	Simulations we show	Simulations of reviewer
Δt	500 ms	300 ms	10 ms (100 Hz)
D	0.006 – 0.018 $\mu\text{m}^2/\text{s}$	0.003 $\mu\text{m}^2/\text{s}$	1 $\mu\text{m}^2/\text{s}$
Lc	0-0.2 μm	0.16 μm	Unknown
Sigma	34 nm	0-50 nm	0-100 nm

As one can see, the reviewer’s simulations are far away from our datasets. Note that there may be alpha estimation errors in our 20 ms datasets collected at 50 Hz with a Sigma of 60 nm. We therefore make no conclusions regarding changes in alpha for these datasets: we only use relative differences in our alpha estimate as one of several parameters to inform our classification. The fact that our classification shows that slow-diffusing molecules (chromatin bound) have an α of 0.51 ± 0.02 and our fast-diffusing molecules have an α of 0.94 ± 0.12 provides evidence that our approach has allowed classification.

Also, can you report the full distribution of localization errors/uncertainties instead of just the median. If the noise is heavy-tailed, the median can be misleading. Basically, I have concerns about the accuracy of the alpha estimation and the lack of validation of their method.

We already show the distributions of the biophysical parameters that we estimate from samples of the dye fixed to the coverslip – see Extended Data Figures 3c and 5b. Because the dye is immobile, these experiments very clearly demonstrate the effect of localisation uncertainty on the parameters we measure. As we explain in the ‘Response to reviewers’, we make no conclusions in the paper about measurements below our precision limit (the 95 % confidence interval in those distributions) to avoid a false interpretation of the data. We have provided extensive validation of the method using simulations and comparison with other methods in the previous ‘Response to reviewers’.

MOTIOR BLURRING

The authors compare their HaloTag-NLS SMT experiment to Zhen eLife 2016 (which also suffered from extensive motion-blurring), and estimate a wide range of diffusion coefficients. In their response, they do not give numbers for their own data, but say it is similar to 0.24 and 2.5 $\mu\text{m}^2/\text{s}$. I believe I have seen other studies measure $>10 \mu\text{m}^2/\text{s}$ for HaloTag alone, and e.g. Izeddin 2014 measure 13.5 $\mu\text{m}^2/\text{sec}$ for Myc.

The concern is that motion-blurring causes you to overestimate the slower moving populations and overestimate the faster moving populations and incorrectly estimate the various subpopulation sizes. This issue needs to be explicitly stated in the main text, with a clear statement that they cannot exclude faster moving states and that the subpopulation sizes may be incorrect.

The reviewer is correct that our imaging conditions were selected to monitor large NuRD complexes and that very fast-moving molecules cannot be detected by our method. Our imaging conditions will therefore affect the tracking of small proteins such as HaloTag and transcription factors – although we can still track them, we will underestimate their diffusion coefficients. We have therefore added a statement to the text as requested: “We also note that, although imaging at 20 ms time resolution allows us to distinguish between large freely diffusing and chromatin bound NuRD complexes, motion blurring would reduce the detection of faster moving molecules (e.g. transcription factors) unless shorter exposures were recorded. On our instrument we have been able to successfully image proteins using 5 ms exposures, but absolute subpopulation sizes need to be interpreted with caution.”

RESPONSE:

“Line 210-220: So the association time of 134 ms in Fig. 2d implies a similar dissociation rate to get similar bound vs. free subpopulations, since percent-bound $\sim k_{ON}/(k_{OFF}+k_{ON})$. So why is the dissociation time >100 sec, i.e. 3 orders of magnitude bigger than the association time. This does not seem right?

The binding of a protein to a substrate with multiple binding sites, some specific and some non-specific, is probably very complex so we wouldn’t expect it to obey the simple rules for 1:1 binding in solution. In particular, multivalency prolongs chromatin retention and accelerates the association rate, see e.g. Kilic et al. (Nat Commun., 2015). There are many ways in which the NuRD complex can interact with chromatin – all of the CHD4, HDAC1/2, MBD2/3, MTA1/2/3 and RbAp46/48 subunits are expected to interact with chromatin in different ways – so we don’t expect simple kinetics.

With regards to the percentage of bound versus free molecules, although the actual percentages may be biased because long trajectories of freely diffusing molecules may be harder to track – we can demonstrate that the changes in percentage bound CHD4 or MTA2 are meaningful. For example, the loss in MTA2 chromatin binding in the absence of MBD3 is supported by our in vitro chromatin binding experiments (Extended Data Figure 4a) and by unpublished CHIP-seq data for MTA2 in wild-type and Mbd3-ko ES cells (see Rebuttal Figure 2, below).”

The authors did not address my comment. If the k_{ON} and k_{OFF} rates are correct, then percent-bound $\sim k_{ON}/(k_{OFF}+k_{ON})$. If not, are the assumptions of a single dissociation rate used in the manuscript not violated?

The authors mention “With regards to the percentage of bound versus free molecules, although the actual percentages may be biased because long trajectories of freely diffusing molecules may be harder to track”. What it looks like the authors is saying is that their data are not quantitatively interpretable and not in internal quantitative agreement. This would need to be stated in the main text, since this is currently not clear and explicit.

I think we make it very clear in the paper that we couldn’t determine dissociation rates due to the unexpectedly long residence times, and therefore that we cannot make any conclusions regarding K_{OFF} (and thus percent-bound from $k_{ON}/(k_{OFF}+k_{ON})$). We have also made another statement now to make it even clearer that there are possible limitations with our k_{OFF} measurements: “[We should note that very long association times are difficult to monitor with our approach given the limit on trajectory length imposed by photobleaching.]”

As we have also explained in the previous ‘Response to reviewers’ the binding of different CHD4 and NuRD complexes in vivo is likely to be complex with quite likely different binding kinetics for the different complexes, which may in addition vary in different regions of eu-/hetero-chromatin within the cell etc. Clearly, a lot more work is needed to characterise association/disassociation rates of NuRD and other CHD4 containing complexes in vivo, but the important point is that our approach now provides a means to do this.

FRAP/FDAP

The experiment the authors did on FDAP in Rebuttal Figure 3a is very nice and strongly supports the long residence time albeit with a slight deviation (~100 sec vs. ~51 sec). To me this is strong support for a long residence time of NuRD and strengthens and supports the case of the authors. It is obviously their own decision, but I think this would make a nice Extended Data Figure.

As the reviewer says, the FDAP experiments do support the conclusions we make using the single molecule tracking (by showing that the K_{off} is very low), and we have now mentioned this experiment in the manuscript.

RAW MOVIES:

Please show us 1) raw movies of the raw NuRD single-molecule tracking data; 2) with localizations and tracking overlaid and 3) show us several raw movies of the dCas9 locus tracking. These could be attached as supplementary movies so we can get a sense of the raw data quality and be referred to in the main text.

We have now added these.

Authors should cite Ref 4 and 5 on page 17 in the context of enhancer dynamics and transcription.

We didn’t understand this comment. The work described in References 4 and 5 doesn’t look at enhancer dynamics whereas that in Reference 3 specifically does.

Page 24, does not seem accurate to refer to cross-linking of chromatin in the last paragraph.

We also didn’t understand this comment but have removed it. [It is well known that Cohesin cross-links different CTCF sites in chromatin together – e.g. TADs and loops disappear in cells when NIPBL, which loads Cohesin, is knocked out (Schwarzer et al., Nature, 2017).]

Reviewer #4:

Overall, the work seems well constructed and designed and the results solid. However, to this reviewer the claims may need to be softened or more data provided. That NuRD decompacts and increases enhancer-dynamics at longer range via modulating CTCF/Cohesin is a strong assumption given the data.

We have modified the abstract to remove any implication of causality.

It is also important to mention that the manuscript is very difficult to follow. Many figures are misplaced or not even referenced. Extended Data Figures are difficult to follow. No clear step-by-step conclusions provided nor a final conclusion/s in the text. Altogether, the reading of the manuscript is tedious and demerits the work. Here are some additional specifics:

It's true that to address comments from the referees the manuscript is now full of further validation studies to confirm the results we see and more explanation. Most of that could probably usefully be deleted, which would make the manuscript clearer and more readable. For the moment we have made sure that there is a short concluding sentence to each section and have added a more comprehensive summary in the Discussion. We have also moved one of the panels of the old Figure 4 (e) to create a new Figure 6 that summarises the main conclusions.

- 1. Why the authors really see 3 states (slow, fast1 and fast2). Not clear for the data and difficult to accept without further explanation. To this reviewer there are 2 states (slow and fast) and does not understand why the slow is split into slow + fast1.***

We agree with the reviewer that we see two states (slow-diffusing and fast-diffusing). We only break the fast-diffusing into two fast-diffusing sub-states because BIC analysis clearly demonstrates that there are molecules/loci with two different anomalous exponents within the fast state. We have hopefully improved the explanation of this.

- 2. Many of the statistical significance are because the genome is really large and one has tons of points in the plots. But, one wonders what it really means (biologically) a "significant" fold change of 1.05?***

As we explain in the paper, we would have expected a larger increase in the apparent diffusion coefficient of freely diffusing CHD4 (1.05-fold, $p = 0.009$ in e.g. the absence of MBD3), but it is consistent with the fact that CHD4 is part of many complexes within ES cells. Since we are not

making any biological conclusion from this we have removed “significant” from the manuscript. Our biological conclusions are made from the much larger change in apparent diffusion coefficient of freely diffusing MTA2 in the absence of MBD3 (1.7-fold, $p < 10^{-5}$).

- 3. *I could not see clear evidence to conclude that “NuRD-bound enhancers contact promoters to down-regulates genes in the A (euchromatic) compartment”.***

We have rewritten this part to hopefully make it clearer.

- 4. *I could not see clear evidence to suggest that “NuRD may alter genome structure/dynamics to affect expression of whole groups of nearby genes”.***

We have modified the text to say that “NuRD alters genome structure/dynamics **and** affects the expression of whole groups of nearby genes in a similar way” – i.e. removing any implication of causality.

- 5. *The provided molecular data clearly shows to this reviewer that compartmentalization is affected in Nbd3-ko cells. However, no very strong and clear data on TAD or loop alterations. Despite that, the manuscripts conclusions are mostly driven to the effect in enhancer-promoter interactions, which are likely modulated by TAD boundaries and loop extrusion.***

We think there is a very clear genome wide trend of decreased TAD insulation in WT cells vs the *Mbd3*-ko (see Extended Data Figure 8i). This is also evident in Figure 5b. We could add more similar plots to show this, but we don’t think it is needed.

- 6. *No Hi-C maps are shown to the exception of a local region. I would suggest to provide the reader with visualization of data that sometimes is stronger than any smaller genome-wide differences. This can be done via JuiceBox, HiGlass or the Nucleome Browser.***

We are not clear what else we could show here. We could show more examples of comparisons of regions of the WT and *Mbd3*-ko Hi-C maps, but they all show similar small differences to that seen in Figure 5b.

- 7. *A/B compartmentalization does not correlate with “more or less” genes.***

The reviewer is correct. We have changed this to say “higher or lower density of” genes.

8. Figure 2 a/b/e never refereed. Figure 2d misplaced in the text.

We thank the referee for pointing this out. [We had rearranged the paper to move the Hi-C analysis from the beginning to the end (following the requests from the referees to extend the analysis of this data) and some changes to the referencing of the Figures were missed.]

Decision Letter, first revision:

Message: 25th Aug 2022

Dear Dr. Laue,

Thank you again for submitting your manuscript "Live-cell 3D single-molecule tracking reveals how NuRD modulates enhancer dynamics". I apologize for the delay in getting back to you, I am afraid that in light of reviewer's comments, this required careful discussion with the editorial team.

We have received comments from the original reviewers #1 and #4. While Reviewer #4 has no further comments, Reviewer #1 remains unconvinced that their points have been fully addressed. At this point, we all agree that it is best for this referee to be withdrawn for further reviewing - it is unfortunate when this happens. Nevertheless, the referee has unique expertise in single molecule tracking experiments, and therefore we need to recruit an arbitrating referee to sign off on these.

In order for us to consult an arbitrating referees, we would kindly request a point-by-point response that addresses in full all the points raised by the Reviewer #1, even those points that relate to issues of toning down or removed data, or data. Please also include the single-molecule trajectory data, all code used and any other data/information that would be helpful for the arbitrating referee to assess your study. Please feel free to reach out if you would like to discuss this further.

We expect to see your revised manuscript within 6 weeks. If you cannot send it within this time, please contact us to discuss an extension; we would still consider your revision, provided that no similar work has been accepted for publication at NSMB or published elsewhere.

Reporting Summary:

Please note that all key data shown in the main figures as cropped gels or blots should be presented in uncropped form, with molecular weight markers. These data can be aggregated into a single supplementary figure item. While these data can be displayed in a relatively informal style, they must refer back to the relevant figures. These data should be submitted with the final revision, as source data, prior to acceptance, but you may want to start putting it together at this point.

Data availability: this journal strongly supports public availability of data. All data used in accepted papers should be available via a public data repository, or alternatively, as Supplementary Information. If data can only be shared on request, please explain why in your Data Availability Statement, and also in the correspondence with your editor. Please note that for some data types, deposition in a public repository is mandatory - more information on our data deposition policies and available repositories can be found below: <https://www.nature.com/nature-research/editorial-policies/reporting-standards#availability-of-data>

We require deposition of coordinates (and, in the case of crystal structures, structure factors) into the Protein Data Bank with the designation of immediate release upon

publication (HPUB). Electron microscopy-derived density maps and coordinate data must be deposited in EMDB and released upon publication. Deposition and immediate release of NMR chemical shift assignments are highly encouraged. Deposition of deep sequencing and microarray data is mandatory, and the datasets must be released prior to or upon publication. To avoid delays in publication, dataset accession numbers must be supplied with the final accepted manuscript and appropriate release dates must be indicated at the galley proof stage.

[redacted]

Sincerely,

Carolina

Carolina Perdigoto, PhD
Chief Editor
Nature Structural & Molecular Biology
orcid.org/0000-0002-5783-7106

Reviewer #4:

Remarks to the Author:

I thank the effort made by the authors to address my concerns specifically for readability of the work as well as the tune down of some claims.

Author Rebuttal, first revision:

Reviewer #1:

I would suggest that the authors either fully demonstrate directed motion and that a version of the manuscript without directed motion is approved after the revisions below.

As we said in the previous 'Response to reviewers', we find it hard to believe that individual molecules would move such long distances in straight lines (and in some cases back again) unless the motion were directed in some way. However, we don't know the mechanism and we are happy to not describe the motion's we see as '*directed*', although it does make it hard to describe the fast F2 motion because it is characterised by high drift (i.e. movement in a defined direction).

DIRECTED MOTION

It's really challenging to be a reviewer when the authors make clearly incorrect claims like "Chromatin inside the cell moves significantly faster than the cell itself – this has been demonstrated in a range of papers including those referred to by the reviewer (Gu et al., Science, 2018; Germier PMID 28978433; Nagashima PMID 30824489)."

Cell movement tends to be faster than chromatin movement. However, the more salient point is that cell movement tends to be directional, making any protein or DNA locus inside the cell also look at least partially directional in its movement.

In fact, the classic newbie mistake in the live-cell nuclear imaging field is to mistake directional cell movement for directed motion of chromatin bound molecules.

We have now removed 'directed motion' from the manuscript to satisfy the reviewer and, quite frankly, to ensure our authors are not subject to further derogatory comments. There is, therefore, no longer a need to respond to this point.

However, as we explain above, we are confident that we are not looking at cell movement. Single molecule tracking data of chromatin bound proteins automatically provides the sorts of controls requested by the reviewer because in most images we see one or more stationary and one or more moving molecule, *at the same time, inside the same cell*. Clearly, we are not looking at cell movement because if we were we would see all the molecules moving in a similar way. We have now added Supplementary videos showing some of our data to demonstrate this directly so that the field can make their own judgements about single molecule tracking data. We also refer to our previous paper describing a single molecule FRET approach (Basu *et al.*, *Nature Commun*, 2018) where we are able to track CHD4 molecules for much longer periods of time and observe chromatin bound CHD4 molecules moving large distances across the cell and back again whilst other molecules remain stationary (see the attached video: CHD4-mEos-JF646_example_500ms_5fps.avi).

We kindly ask the reviewer for references that “*Cell movement tends to faster than chromatin movement*” at the timescale of our imaging (sliding windows of 5 s). There are several old papers in the literature showing the movement of chromosome sites from the periphery to the interior and back to the periphery [see e.g. Tumber & Belmont, Interphase movements of a DNA chromosome region modulated by VP16 transcriptional activator, *Nature Cell Biology*, **3**, 134–139 (2001)]. There are also recent papers that support our findings of tethered (low α) and untethered (high α) chromatin motion (e.g. Shaban *et al.* Hi-D: nanoscale mapping of nuclear dynamics in single living cells. *Genome Biol* **21**, 95 (2020). <https://doi.org/10.1186/s13059-020-02002-6>).

So let me outline what I would need to be persuaded of the directed motion claim:

- 1. Clearly define your criteria for directed motion. The authors are vague about this, but seem to suggest that it is not just $\alpha > 1$. Is it the drift vector?***

This was clearly stated in the text on p9 of the previous draft. “In addition, values of α that are higher than 1 can represent energy-dependent directed motion.” and “Finally, by computing the magnitude V_i (k) of the drift vector V_i in three dimensions we can characterise directed movement of a molecule.”

- 2. Show us positive and negative controls. Many other studies have shown many other chromatin loci that are not directed. Thus, repeat the dCas9-locus tracking for a handful of other loci and run the same analysis and show that these show no directed motion.***
- 3. For positive controls, find some examples of directed motion and demonstrate this.***

- 4. Same for protein tracking. HaloTag-NLS is a nice control for some things, but we need a largely DNA bound protein. Do SMT of both a positive and negative directed motion control protein, and run the same analysis, and show that only NuRD and NuRD-bound enhancers show directed motion. Perhaps a histone protein would work as a negative control here.**

We have carried out all the controls and further experiments previously requested by the referees, including the HaloTag control suggested by Referee 1. We don't feel it is reasonable for he/she to ask for further experiments now.

Moreover, because much of chromatin may well be actively moved in the cell – e.g. during transcription or replication – it is not clear what regions of chromatin or which chromatin bound proteins (including histones) would make good controls. Studying the movement of many different regions/proteins might reveal directed motions.

Our aim has been to study NuRD assembly and function, not to provide a comprehensive study of directed motion of chromatin or chromatin-bound proteins. The important point, however, is that our approach now allows one to do this.

- 5. Do 2-color imaging. In the first color, use an orthogonal dCas9-RFP (or similar) bound to telomeres. This will give you many spots to calculate cell movement. Then simultaneously do dCas9-GFP imaging with and without correction for cell movement (e.g. use center of mass of the telomeres to estimate cell movement), run the same analysis and show that the directed motion claim is not an artifact of cell movement.**

Our single molecule tracking data of chromatin bound proteins already provides exactly these sorts of controls because in most images we see one or more stationary and one or more moving molecule, at the same time, inside the same cell. Clearly, we are not looking at cell movement because if we were we would see all the molecules moving in a similar way. We have now added Supplementary videos showing some of our data to demonstrate this directly.

- 6. Show us at least 5 independent large-field-of-view movies (~100um x 100 um) of confluent mESC colonies with the nucleus labeled (e.g. Histone GFP) imaged for at least 1 hour and at least 1 z-stack per minute (ideally faster). Show us these 5 movies and show how none of one the cells noticeably move.**

Clearly, live cells do move on the timescale of hours. However, we are looking at large scale movements of chromatin bound proteins and enhancers on the timescale of a few minutes in locus tracking and only seconds in single-molecule tracking experiments. Our imaging clearly shows that cells are not moving appreciably on these timescales.

- 7. *In addition to the speed of mESC movement, the point is that at the minute time-scale, cell movement tends to be directional such that even if cell movement is comparable to single-molecule movement, when summed, it will make single-molecule movement look directional. [...] That KLF4 also shows directed movement in their analysis does not strengthen the claim, since if it is a cell movement artifact, essentially all chromatin-bound proteins would similarly show this, as would also loci.***

Clearly, cell movement is not comparable to single molecule movement – see response to 5) above.

- 8. *Systematically benchmark the accuracy of the algorithm for inferring alpha on simulations similar to the experimental data with worse and better localization noise.***

We provided simulations similar to the experimental data to assess the effect of localisation noise on the estimation of apparent diffusion constants in the previous ‘Response to referees’ because we cannot remove the localization noise from the diffusion coefficient. As we also explained, however, we have shown in previous work (Shukron *et al.*, *Trends in Genetics*, 2019) that alpha is not particularly sensitive to localisation noise, so we do not think we need to carry out further simulations to address this.

Similarly for the magnitude of the drift vector, V.

- 9. *Find a good protein and DNA negative control for directed movement, and show us that the magnitude of the drift vector is 0, such that cell movement does not artefactually make it look like directed motion.***

See the response to 5) – the stationary molecules that we detect at the same time as moving molecules already provide this control – they have a drift vector that is close to 0. Clearly, we are not looking at cell movement that artefactually looks like directed motion.

10. Ideally, find a mutant that disrupts the directed motion (e.g. if you could KO a factor, which causes NuRD/enhancers to no longer be directed).

Our paper already provides a mutant like this – but which works in the opposite sense. By knocking out *Mbd3* we greatly increase the occurrence of untethered F2 motion.

11. Based on the telomere-dCas9 image, they will get “cell movement trajectories”. Now perform simulations of undirected motion (e.g. Brownian and confined) and add the cell movement trajectories to the simulated data as well as realistic noise. Then apply the same analysis algorithm and show us that directed motion is not accidentally inferred.

We believe that the differences in movement between stationary and moving molecules is sufficiently clear from looking at the raw videos that this sort of simulation is not needed. We refer to our previous paper describing a single molecule FRET approach (Basu *et al.*, *Nature Commun*, 2018) where we are able to track single CHD4 molecules for long periods of time and observe chromatin bound CHD4 moving large distances across the cell and back again whilst other molecules remain stationary (see the attached video taken from Supplementary video 3 from that paper).

Frankly, if the authors could demonstrate directed enhancer movement this would in my opinion be a landmark paper on its own, but as mentioned in my first review, extraordinary claims require extraordinary evidence, and many controls are missing here in my opinion.

Because our paper is based on so many different single molecule experiments, where we collect data from stationary and moving chromatin bound molecules at the same time inside the same cell, with data provided in pretty much every figure, we are quite frankly astounded that the referee is even questioning this.

SPECIFIC COMMENTS:

Hi-C concern: the authors did not really address my Hi-C analysis concern. Something got lost in communication and they have addressed a different concern. To be clear, I was not worried about

experimental variability between replicates, though this is always good to estimate. I was concerned about Hi-C analysis with few contacts as I wrote (>500 M needed). If they have indeed gotten sufficiently resolution now, I'd like to see a Extended Data Figure demonstrating this. To avoid any further misunderstanding, I'll try to be very specific. I'd like to see a new Extended Data Figure showing 4 separate 1Mb regions at 5kb resolution where one triangle is WT-data from this paper and the other triangle is data from Pekowska 2018 (since the author say that only Pekowska used same cell culture conditions). I'd like to see 4 separate 1Mb regions where you can clearly see loops/dots/cornerpeaks in the Pekowska data, and see that directly compared to the present paper.

As we discuss above, the Hi-C experiments were included solely to explain to the reader why we embarked on our single molecule imaging study. We showed in our previous 'Response to reviewers' (and now also provide further figures showing example loops below) that our data for WT cells was consistent with the WT data previously deposited by Pekowska et al., and we therefore combined it with our own. We then collected extra data for the Mbd3-ko so that we could generate large datasets for both the WT and Mbd3-ko cells and carry out the analyses suggested by the three reviewers. We can see no sense in wasting a lot more money to carry out further sequencing so that we can make a high-resolution comparison between our own and the Pekowska et al. WT data. We think the data we provide in the revised manuscript is quite sufficient to show that our WT data is comparable to the Pekowska et al. Hi-C experiments.

To support the claim that the authors study enhancer-promoter interactions in Hi-C, I would also like to see some example plots side-by-side with Pekowska, where there is a clear loop/dot between an enhancer and a promoter. And the same for the Mbd3-KO Hi-C data side by side.

See above – the Hi-C experiments were included solely to introduce our single molecule imaging study. It was not our purpose to claim that we can ‘study E-P interactions using Hi-C’. The E-P analysis (ABC contact analysis) was carried out to satisfy the requests of another reviewer, who asked us to see if E-P contact changes were consistent with transcriptional changes observed and indeed they are. Furthermore, the change in E-P proximity we observe has also been validated using the more sensitive and accurate 3D DNA FISH approach.

LOCALIZATION UNCERTAINTY:

The authors state that “As we discuss in the response to Q2 above, computing the MSD and then taking the average over many realizations shows that the anomalous exponent alpha is not affected by the localization noise – thus we do not need to worry about the impact of the localization error on alpha.” I am sorry but do not believe this is correct. Try simulating Brownian motion (alpha=1) at 100 Hz with sigma = [0,10,20,40,60,80,100 nm] and a D=1um²/s and then chop up the data in short trajectories of 5 frames and fit an MSD, both at the single trajectory data and time-and-ensemble averaged MSD. If the authors claim is true, a fit of MSD=4Dt^{alpha} should give the same estimates of D and alpha as well as the same distribution of D and alpha estimates from single trajectories independent of noise. When I do this, I see an effect of noise on the alpha estimate. Also, the authors reference yeast data from Shukron, but what they need to do is to systematically test the accuracy of their estimate of alpha using simulations that are realistic for the present paper. They do something along these lines on page 53 for other parameters, but they need to do and show this for alpha as well.

As we explained in our previous ‘Response to reviewers’ localisation uncertainty has only a small effect on the estimation of alpha – indeed this can be seen in the simulations carried out by Shukron et al. which we showed. As we also explained, the absolute values of the parameters we measure are not really of interest – we are interested in (and indeed can only measure) changes between apparent values determined at the same time from different samples under the same labelling and imaging conditions (i.e. datasets that have the same localisation error).

Regarding the simulations of the reviewer, it is an impossible task to comment on the coding of a reviewer who has not sent us his/her code, methods or results. However, qualitatively, for the 500 ms data where we report changes in α , we compare below the simulations we carry out vs those the reviewer has carried out:

	Actual parameters	Simulations we show	Simulations of reviewer
Δt	500 ms	300 ms	10 ms
D	0.006 – 0.018 $\mu\text{m}^2/\text{s}$	0.003 $\mu\text{m}^2/\text{s}$	1.0 $\mu\text{m}^2/\text{s}$
Lc	0-0.2 μm	0.16 μm	Unknown
Sigma	34 nm	0-50 nm	0-100 nm

As one can see, the reviewer’s simulations are far away from our datasets. Note that there may be alpha estimation errors in our 20 ms datasets collected at 50 Hz with a Sigma of 60 nm. We therefore make no conclusions regarding changes in alpha for these datasets: we only use relative differences in our alpha

estimate as one of several parameters to inform our classification. The fact that our classification shows that slow-diffusing molecules (chromatin bound) have an α of 0.51 ± 0.02 and our fast-diffusing molecules have an α of 0.94 ± 0.12 provides evidence that our approach has allowed classification.

Also, can you report the full distribution of localization errors/uncertainties instead of just the median. If the noise is heavy-tailed, the median can be misleading. Basically, I have concerns about the accuracy of the alpha estimation and the lack of validation of their method.

We do show the distributions of the biophysical parameters that we estimate from samples of the dye fixed to the coverslip – see Extended Data Figures 3c and 5b. Because the dye is immobile, these experiments very clearly demonstrate the effect of localisation uncertainty on the parameters we measure. As we explained in the previous ‘Response to reviewers’, we make no conclusions in the paper about measurements below our precision limit (the 95 % confidence interval in those distributions) to avoid a false interpretation of the data. We have provided extensive validation of the method using simulations and comparison with other methods in the previous ‘Response to reviewers’.

MOTION BLURRING

The authors compare their HaloTag-NLS SMT experiment to Zhen eLife 2016 (which also suffered from extensive motion-blurring), and estimate a wide range of diffusion coefficients. In their response, they do not give numbers for their own data, but say it is similar to 0.24 and 2.5 $\mu\text{m}^2/\text{s}$. I believe I have seen other studies measure $>10 \mu\text{m}^2/\text{s}$ for HaloTag alone, and e.g. Izeddin 2014 measure 13.5 $\mu\text{m}^2/\text{sec}$ for Myc.

The concern is that motion-blurring causes you to overestimate the slower moving populations and overestimate the faster moving populations and incorrectly estimate the various subpopulation sizes. This issue needs to be explicitly stated in the main text, with a clear statement that they cannot exclude faster moving states and that the subpopulation sizes may be incorrect.

The reviewer is correct that our imaging conditions were selected to monitor large NuRD complexes and that very fast-moving molecules cannot be detected by our method. Our imaging conditions will therefore affect the tracking of small proteins such as HaloTag and transcription factors – although we can still track them, we will underestimate their diffusion coefficients. We have therefore added a statement to the text as requested: “We also note that, although motion blurring would significantly reduce the detection of faster moving molecules (e.g. transcription factors) unless shorter exposures were recorded, imaging at

20 ms time resolution allows us to distinguish between freely diffusing and chromatin bound NuRD complexes due to the large size of the complex.””

RESPONSE:

“Line 210-220: So the association time of 134 ms in Fig. 2d implies a similar dissociation rate to get similar bound vs. free subpopulations, since $\text{percent-bound} \sim k_{\text{ON}} / (k_{\text{OFF}} + k_{\text{ON}})$. So why is the dissociation time >100 sec, i.e. 3 orders of magnitude bigger than the association time. This does not seem right?”

The binding of a protein to a substrate with multiple binding sites, some specific and some non-specific, is probably very complex so we wouldn’t expect it to obey the simple rules for 1:1 binding in solution. In particular, multivalency prolongs chromatin retention and accelerates the association rate, see e.g. Kilic et al. (Nat Commun., 2015). There are many ways in which the NuRD complex can interact with chromatin – all of the CHD4, HDAC1/2, MBD2/3, MTA1/2/3 and RbAp46/48 subunits are expected to interact with chromatin in different ways – so we don’t expect simple kinetics.

With regards to the percentage of bound versus free molecules, although the actual percentages may be biased because long trajectories of freely diffusing molecules may be harder to track – we can demonstrate that the changes in percentage bound CHD4 or MTA2 are meaningful. For example, the loss in MTA2 chromatin binding in the absence of MBD3 is supported by our in vitro chromatin binding experiments (Extended Data Figure 4a) and by unpublished ChIP-seq data for MTA2 in wild-type and Mbd3-ko ES cells (see Rebuttal Figure 2, below). ”

The authors did not address my comment. If the k_{ON} and k_{OFF} rates are correct, then $\text{percent-bound} \sim k_{\text{ON}} / (k_{\text{OFF}} + k_{\text{ON}})$. If not, are the assumptions of a single dissociation rate used in the manuscript not violated?

The authors mention “With regards to the percentage of bound versus free molecules, although the actual percentages may be biased because long trajectories of freely diffusing molecules may be harder to track”. What it looks like the authors is saying is that their data are not quantitatively interpretable and not in internal quantitative agreement. This would need to be stated in the main text, since this is currently not clear and explicit.

I think we make it very clear in the paper that we couldn't determine dissociation rates due to the unexpectedly long residence times, and therefore that we cannot make any conclusions regarding K_{OFF} (and thus percent-bound from $k_{ON}/(k_{OFF}+k_{ON})$). We have also made another statement now to make it even clearer that there are possible limitations with our k_{OFF} measurements: “[We should note that very long association times are difficult to monitor with our approach given the limit on trajectory length imposed by photobleaching.]”

As we have also explained in the previous ‘Response to reviewers’ the binding of different CHD4 and NuRD complexes *in vivo* is likely to be complex, with quite likely different binding kinetics for NuRD complexes assembled from different components, which may in addition vary in different regions of eu-/hetero-chromatin within the cell etc. Clearly, a lot more work is needed to characterise association/disassociation rates of NuRD and other CHD4 containing complexes *in vivo*, but the important point is that our approach now provides a means to do this.

FRAP/FDAP

The experiment the authors did on FDAP in Rebuttal Figure 3a is very nice and strongly supports the long residence time albeit with a slight deviation (~100 sec vs. ~51 sec). To me this is strong support for a long residence time of NuRD and strengthens and supports the case of the authors. It is obviously their own decision, but I think this would make a nice Extended Data Figure.

As the reviewer says, the FDAP experiments do support the conclusions we make using the single molecule tracking (by showing that the K_{off} is very low), and we have now mentioned this experiment in the manuscript.

RAW MOVIES:

Please show us 1) raw movies of the raw NuRD single-molecule tracking data; 2) with localizations and tracking overlaid and 3) show us several raw movies of the dCas9 locus tracking. These could be attached as supplementary movies so we can get a sense of the raw data quality and be referred to in the main text.

We have now added these.

Authors should cite Ref 4 and 5 on page 17 in the context of enhancer dynamics and transcription.

We didn't understand this comment. The work described in References 4 and 5 doesn't look at enhancer dynamics whereas that in Reference 3 specifically does.

Page 24, does not seem accurate to refer to cross-linking of chromatin in the last paragraph.

We also didn't understand this comment but have removed it. [It is well known that Cohesin cross-links different CTCF sites in chromatin together – e.g. TADs and loops disappear in cells when NIPBL, which loads Cohesin, is knocked out (Schwarzer et al., Nature, 2017).]

Reviewer #4:

Overall, the work seems well constructed and designed and the results solid. However, to this reviewer the claims may need to be soften or more data provided. That NuRD decompacts and increases enhancer-dynamics at longer range via modulating CTCF/Cohesin is a strong assumption given the data.

We have modified the abstract to remove any implication of causality.

It is also important to mention that the manuscript is very difficult to follow. Many figures are misplaced or not even referenced. Extended Data Figures are difficult to follow. No clear step-by-step conclusions provided nor a final conclusion/s in the text. Altogether, the reading of the manuscript is tedious and demerits the work. Here are some additional specifics:

It's true that to address comments from the referees the manuscript is now full of further validation studies to confirm the results we see and more explanation. Most of these could probably usefully be

deleted, which would make the manuscript clearer and more readable. For the moment we have made sure that there is a short concluding sentence to each section and have added a more comprehensive summary in the Discussion. We have also moved one of the panels of the old Figure 4 (e) to create a new Figure 6 that summarises the main conclusions.

- 9. *Why the authors really see 3 states (slow, fast1 and fast2). Not clear for the data and difficult to accept without further explanation. To this reviewer there are 2 states (slow and fast) and does not understand why the slow is split into slow + fast1.***

We agree with the reviewer that we see two states (slow-diffusing and fast-diffusing). We only break the fast-diffusing into two fast-diffusing sub-states because BIC analysis clearly demonstrates that there are molecules/loci with two different anomalous exponents within the fast state. We have hopefully improved the explanation of this.

- 10. *Many of the statistical significance are because the genome is really large and one have tons of points in the plots. But, one wonders what it really means (biologically) a “significant” fold change of 1.05?***

As we explain in the paper, we would have expected a larger increase in the apparent diffusion coefficient of freely diffusing CHD4 (1.05-fold, $p = 0.009$ in e.g. the absence of MBD3), but it is consistent with the fact that CHD4 is part of many complexes within ES cells. Since we are not making any biological conclusion from this we have removed “significant” from the manuscript. Our biological conclusions are made from the much larger change in apparent diffusion coefficient of freely diffusing MTA2 in the absence of MBD3 (1.7-fold, $p < 10^{-5}$).

- 11. *I could not see clear evidence to conclude that “NuRD-bound enhancers contact promoters to down-regulates genes in the A (euchromatic) compartment”.***

We have rewritten this part to hopefully make it clearer.

- 12. *I could not see clear evidence to suggest that “NuRD may alter genome structure/dynamics to affect expression of whole groups of nearby genes”.***

We have modified the text to say that “NuRD alters genome structure/dynamics **and** affects the expression of whole groups of nearby genes in a similar way” – i.e. removing any implication of causality.

13. The provided molecular data clearly shows to this reviewer that compartmentalization is affected in *Nbd3*-ko cells. However, no very strong and clear data on TAD or loop alterations. Despite that, the manuscripts conclusions are mostly driven to the effect in enhancer-promoter interactions, which are likely modulated by TAD boundaries and loop extrusion.

We think there is a very clear genome wide trend of decreased TAD insulation in WT cells vs the *Mbd3*-ko (see Extended Data Figure 8i). This is also evident in Figure 5b. We could add more similar plots to show this, but we don't think it is needed.

14. No Hi-C maps are shown to the exception of a local region. I would suggest to provide the reader with visualization of data that sometimes is stronger than any smaller genome-wide differences. This can be done via JuiceBox, HiGlass or the Nucleome Browser.

We are not clear what else we could show here. We could show more examples of comparisons of regions of the WT and *Mbd3*-ko Hi-C maps, but they all show similar small differences to that seen in Figure 5b.

15. A/B compartmentalization does not correlate with “more or less” genes.

The reviewer is correct. We have changed this to say ‘higher or lower density of’ genes.

16. Figure 2 a/b/e never refereed. Figure 2d misplaced in the text.

We thank the referee for pointing this out. [We had rearranged the paper to move the Hi-C analysis from the beginning to the end (following the requests from the referees to extend the analysis of this data) and some changes to the referencing of the Figures were missed.]

Decision Letter, second revision:

Message: Our ref: NSMB-A45588B

7th Feb 2023

Dear Dr. Laue,

Thank you for submitting your revised manuscript "Live-cell 3D single-molecule tracking reveals how NuRD modulates enhancer dynamics" (NSMB-A45588B). Please accept my sincere apologies for the very long delay reaching a final decision on your study.

It has now been seen by the arbitrating referee who we invited to assess the single molecule experiments - their comments are below. The reviewers has no significant technical concerns, and therefore we'll be happy in principle to publish it in Nature Structural & Molecular Biology, pending minor revisions to address the referees' minor requests and to comply with our editorial and formatting guidelines.

Sincerely,

Carolina

Carolina Perdigoto, PhD
Chief Editor
Nature Structural & Molecular Biology
orcid.org/0000-0002-5783-7106

Reviewer #5 (Remarks to the Author):

This manuscript presents intriguing new insights into the role of NuRD in the regulation of chromatin dynamics at enhancers and enhancer-promoter interactions. The data, obtained using state-of-the-art methods, are of high quality and support the conclusions of the manuscript. The manuscript substantially adds to the current understanding of chromatin dynamics and suggests a number of exciting avenues for future studies, including intriguing evidence for a potential interplay between the activities of NuRD and cohesin/CTCF. Overall, the manuscript is well-written and would appeal to a broad readership at NSMB.

The skepticism regarding the directional motion in chromatin seems somewhat surprising given that a number of processes, including loop extrusion by cohesin, are expected to produce such a movement. Importantly, the 500-ms exposure videos that show stationary and mobile molecules at the same time in the same cell, in my opinion, provide sufficient evidence to exclude the possibility that the observed directed motion might be caused by an artifact that arises from cell movement.

Some minor points:

Since the association times estimated based on 20-ms data (between 100-200 ms) are comparable to the mean track length (around 10 frames, according to EDF 4b, or 200 ms), it seems that these values were substantially limited by photobleaching. This does not affect the conclusions, since photobleaching rates are similar among the conditions that are being compared. If anything, it is likely that the difference between +/-MBD3 conditions for MTA2 is underestimated due to photobleaching. Nevertheless, it would seem appropriate to discuss this limitation in the text not only for the dissociation rates, but also for the association rates, since otherwise it would create the impression that association rates are not affected by photobleaching. Again, this does not seem to be the case, even though they are affected to a lesser extent.

End of page 15: "Importantly, visualisation of individual trajectories identified molecules that switch between the three states; S, F1 and F2 (see e.g. the trajectory in Figure 3a)"

Since the existence of such trajectories is important for demonstrating that different states do not simply arise from several coexisting populations of CHD4-containing complexes, it would seem appropriate for the authors to provide a bit more information about these trajectories beyond referring to an example trajectory (e.g., a total number and/or fraction of such traces).

Page 28: "protein subunit that links the chromatin remodelling and histone deacetylase activities of NuRD together"

Since histone modification activity (e.g., deacetylation) is usually also classified as chromatin remodeling, it would seem more appropriate to use the term "nucleosome sliding" here instead of "chromatin remodeling"

Author Rebuttal, second revision:

Response to Remaining Reviewer Comments (NSMB-A45588B)

Reviewer #5

Since the association times estimated based on 20-ms data (between 100-200 ms) are comparable to the mean track length (around 10 frames, according to EDF 4b, or 200 ms), it seems that these values were substantially limited by photobleaching. This does not affect the conclusions, since photobleaching rates are similar among the conditions that are being compared. If anything, it is likely that the difference between +/-MBD3 conditions for MTA2 is underestimated due to

photobleaching. Nevertheless, it would seem appropriate to discuss this limitation in the text not only for the dissociation rates, but also for the association rates, since otherwise it would create the impression that association rates are not affected by photobleaching. Again, this does not seem to be the case, even though they are affected to a lesser extent.

The referee is correct. We have added a note to say that measurements of association times were also affected by photobleaching, albeit less so because the times are shorter. (In more recent work we are now only using tracks where we see the molecule come off chromatin and then rebind, which removes this problem.)

End of page 15: “Importantly, visualisation of individual trajectories identified molecules that switch between the three states; S, F1 and F2 (see e.g. the trajectory in Figure 3a)”

Since the existence of such trajectories is important for demonstrating that different states do not simply arise from several coexisting populations of CHD4-containing complexes, it would seem appropriate for the authors to provide a bit more information about these trajectories beyond referring to an example trajectory (e.g., a total number and/or fraction of such traces).

We have now provided a Table showing the proportions of trajectories containing either freely diffusing, confined or both freely diffusing and confined sub-trajectories in EDF 3g, and a further Table showing the proportions of trajectories containing either slow (S), fast (F1), or fast (F2) chromatin bound sub-trajectories (or combinations thereof) in EDF 5g.

“Page 28: “protein subunit that links the chromatin remodelling and histone deacetylase activities of NuRD together”

Since histone modification activity (e.g., deacetylation) is usually also classified as chromatin remodeling, it would seem more appropriate to use the term “nucleosome sliding” here instead of “chromatin remodeling”

I think we would prefer to leave this as it is. NuRD certainly does lead to nucleosome sliding, but this could involve the partial dismantling of nucleosomes (e.g. via the transient loss of H2A/H2B dimers) or indeed their complete eviction followed by reassembly. For this reason, we prefer to refer to CHD4 activities as nucleosome remodelling. Histone deacetylation, on the other hand, mainly appears to lead to changes in compaction resulting from the modification of interactions *between* nucleosomes or through facilitating the binding of proteins such as BRD4.

Final Decision Letter:

Message Dear Dr. Laue,
:

We are now happy to accept your revised paper "Live-cell 3D single-molecule tracking reveals modulation of enhancer dynamics by NuRD" for publication as a Article in Nature Structural & Molecular Biology.

Your paper will be published online soon after we receive proof corrections and will appear in print in the next available issue. You can find out your date of online publication by contacting the production team shortly after sending your proof corrections. Content is published online weekly on Mondays and Thursdays, and the embargo is set at 16:00 London time (GMT)/11:00 am US Eastern time (EST) on the day of publication. Now is the time to inform your Public Relations or Press Office about your paper, as they might be interested in promoting its publication. This will allow them time to prepare an accurate and satisfactory press release. Include your manuscript tracking number (NSMB-A45588C) and our journal name, which they will need when they contact our press office.

About one week before your paper is published online, we shall be distributing a press release to news organizations worldwide, which may very well include details of your work. We are happy for your institution or funding agency to prepare its own press release, but it must mention the embargo date and Nature Structural & Molecular Biology. If you or your Press Office have any enquiries in the meantime, please contact press@nature.com.

Please note that *Nature Structural & Molecular Biology* is a Transformative Journal (TJ). Authors may publish their research with us through the traditional subscription access route or make their paper immediately open access through payment of an article-processing charge (APC). Authors will not be required to make a final decision about access to their article until it has been accepted. Find out more about Transformative Journals <https://www.springernature.com/gp/open-research/transformative-journals>

Authors may need to take specific actions to achieve [compliance with funder and institutional open access mandates](https://www.springernature.com/gp/open-research/funding/policy-compliance-faqs). If your research is supported by a funder that requires immediate open access (e.g. according to [Plan S principles](https://www.springernature.com/gp/open-research/plan-s-compliance)) then you should select the gold OA route, and we will direct you to the compliant route where possible. For authors selecting the subscription publication route, the journal's standard licensing terms will need to be accepted, including [self-archiving policies](https://www.springernature.com/gp/open-research/policies/journal-policies). Those licensing terms will supersede any other terms that the author or any third party may assert apply to any version of the manuscript.

In approximately 10 business days you will receive an email with a link to choose the appropriate publishing options for your paper and our Author Services team will be in

touch regarding any additional information that may be required.

Sincerely,

Carolina Perdigoto, PhD
Chief Editor
Nature Structural & Molecular Biology
orcid.org/0000-0002-5783-7106
